# Insights from elastic thermobarometry into exhumation of high-pressure metamorphic rocks from Syros, Greece

Miguel Cisneros[1,2*], Jaime D. Barnes[1], Whitney M. Behr[1,2], Alissa J. Kotowski[1,3*], Daniel F. Stockli[1], and Konstantinos Soukis[4]

[1]Department of Geological Sciences, Jackson School of Geosciences, University of Texas at Austin, Austin, TX, USA
[2*]Current address: Geological Institute, ETH Zürich, Zürich, Switzerland
[3*]Current address: Department of Earth and Planetary Sciences, McGill, Montreal, Canada
[4]Faculty of Geology and Geoenvironment, NKUA, Athens, Greece

*Correspondence to*: Miguel Cisneros (miguel.cisneros@erdw.ethz.ch)

**Abstract.** Retrograde metamorphic rocks provide key insights into the pressure-temperature (P-T) evolution of exhumed material, and resultant P-T constraints have direct implications for the mechanical and thermal conditions of subduction interfaces. However, constraining P-T conditions of retrograde metamorphic rocks has historically been challenging and has resulted in debate about the conditions experienced by these rocks. In this work, we combine elastic thermobarometry with oxygen isotope thermometry to quantify the P-T evolution of retrograde metamorphic rocks of the Cycladic Blueschist Unit (CBU), an exhumed subduction complex exposed on Syros, Greece. We employ quartz-in-garnet and quartz-in-epidote barometry to constrain pressures of garnet and epidote growth near peak subduction conditions and during exhumation, respectively. Oxygen isotope thermometry of quartz and calcite within boudin necks was used to estimate temperatures during exhumation and to refine pressure estimates. Three distinct pressure groups are related to different metamorphic events and fabrics: high-pressure garnet growth at ~1.4 - 1.7 GPa between 500 - 550 °C, retrograde epidote growth at ~1.3 – 1.5 GPa between 400 - 500 °C, and a second stage of retrograde epidote growth at ~1.0 GPa and 400 °C. These results are consistent with different stages of deformation inferred from field and microstructural observations, recording prograde subduction to blueschist-eclogite facies and subsequent retrogression under blueschist-greenschist facies conditions. Our new results indicate that the CBU experienced cooling during decompression after reaching maximum high-pressure/low-temperature conditions. These P-T conditions and structural observations are consistent with exhumation and cooling within the subduction channel in proximity to the refrigerating subducting plate, prior to Miocene core-complex formation. This study also illustrates the potential of using elastic thermobarometry in combination with structural and microstructural constraints, to better understand the P-T-deformation conditions of retrograde mineral growth in HP/LT metamorphic terranes.

## 1 Introduction

Constraining the pressure-temperature (P-T) evolution of metamorphic rocks is fundamental for understanding the mechanics, timescales, and thermal conditions of plate tectonic processes operating on Earth. Historically, one of the most challenging aspects of thermobarometry has been deciphering the P-T evolution of rocks during their exhumation from peak depths back to the surface (e.g., Essene, 1989; Kohn and Spear, 2000; Pattison et al., 2003; Schliestedt and Matthews, 1987; Spear and Pattison, 2017; Spear and Selverstone, 1983). Exhumation P-T paths are particularly challenging to reconstruct because during retrogression rocks are cooled, fluids are consumed by metamorphic reactions, and strain is progressively localized, all of which result in more sluggish reaction kinetics and lesser degrees of chemical equilibrium (e.g., Baxter, 2003; Carlson, 2002; Jamtveit et al., 2016; Rubie, 1998). These issues are especially pronounced in high-pressure/low-temperature (HP/LT) environments characteristic of subduction zones.

Elastic thermobarometry offers an alternative to conventional thermobarometry. Rather than relying on equilibrium metamorphic reactions, this approach constrains the P-T conditions at which a host crystal entraps an inclusion (e.g., Adams et al., 1975a, 1975b; Rosenfeld, 1969; Rosenfeld and Chase, 1961). Because inclusion-host-pair bulk moduli and thermal expansivities commonly differ, upon ascent, an inclusion develops residual strain(s) that can be determined from measurements of Raman shifts. A residual inclusion pressure can be calculated from strain(s) by using Grüneisen tensors (Angel et al., 2019; Murri et al., 2018, 2019) or experimental hydrostatic calibrations (e.g., Ashley et al., 2014; Enami et al., 2007; Thomas and Spear, 2018). Elastic modeling is then used to calculate the initial entrapment conditions of when the host grew around the inclusion, and thus can be used to determine the conditions at which individual host minerals grew during metamorphism (e.g., Alvaro et al., 2020; Ashley et al., 2014; Enami et al., 2007).

The purpose of this study is to illustrate the potential of using elastic thermobarometry in combination with structural and microstructural observations, to better understand the P-T-deformation (D) conditions of prograde-to-peak and retrograde mineral growth in subduction-related HP/LT metamorphic rocks. We focus on a subduction complex exposed on Syros Island, Cyclades, Greece, where despite several decades of petrological study, the early exhumation history remains enigmatic. We combine the recently tested quartz-in-epidote (qtz-in-ep) barometer (Cisneros et al., 2020), quartz-in-garnet (qtz-in-grt) barometry (e.g., Ashley et al., 2014; Bonazzi et al., 2019; Thomas and Spear, 2018), and oxygen isotope thermometry (e.g., Javoy, 1977; Urey, 1947), to constrain metamorphic growth pressures and temperatures near peak subduction depths and during early exhumation. The results demonstrate that combining qtz-in-ep barometry with careful structural and microstructural observations allows us to delineate a retrograde P-T-D path that is contextually constrained, and provide new insights into the exhumation history of the CBU on Syros, Greece.

## 2. Geologic Setting

Syros Island in the Cyclades of Greece consists of metamorphosed tectonic slices of oceanic and continental affinity that belong to the Cycladic Blueschist Unit (CBU), structurally below the Pelagonian Upper Unit (Fig. 1). CBU rocks on Syros

record Eocene subduction (~52 - 49 Ma) to peak blueschist-eclogite facies conditions (Bröcker et al., 2013; Cliff et al., 2017; Lagos et al., 2007; Laurent et al., 2017; Lister and Forster, 2016; Putlitz et al., 2005; Tomaschek et al., 2003; Uunk et al., 2018), followed by exhumation during Oligo-Miocene (~25 Ma) back-arc extension (e.g., Jolivet and Brun, 2010; Ring et al., 2010). A retrograde regional metamorphic event occurred between 25-18 Ma and caused greenschist- to amphibolite facies metamorphism in the Cycladic islands, but was most pervasive in the footwall adjacent to the large-scale extensional North and West Cycladic Detachment Systems (e.g., Bröcker et al., 1993; Bröcker and Franz, 2006; Gautier et al., 1993; Grasemann et al., 2012; Jolivet et al., 2010; Pe-Piper and Piper, 2002; Schneider et al., 2018). Despite these documented metamorphic events, the exhumation history of the CBU between ~52 and ~25 Ma remains enigmatic and poorly constrained; yet, this period spans exhumation of the CBU from maximum subduction to middle crust pressures (~0.3 - 0.7 GPa). Previous work has constrained some aspects of the early exhumation history of the CBU on Syros, including: the timing of peak and retrograde metamorphism (e.g., Bröcker et al., 2013; Cliff et al., 2017; Lagos et al., 2007; Laurent et al., 2017; Skelton et al., 2018; Tomaschek et al., 2003), prograde and exhumation-related kinematics (e.g., Behr et al., 2018; Keiter et al., 2011; Kotowski and Behr, 2019; Laurent et al., 2016; Philippon et al., 2011; Rosenbaum et al., 2002), and the retrograde P-T path (e.g., Laurent et al., 2018; Ring et al., 2020; Schumacher et al., 2008; Skelton et al., 2018; Trotet et al., 2001a, 2001b); however, debate remains about the relationship between deformation events and retrograde metamorphism, the maximum pressure reached by different CBU rock types, the retrograde P-T path, and the mechanisms and kinematics of CBU exhumation.

In this work, we focus on rocks within the CBU, which consist of intercalated metavolcanic and metasedimentary rocks, metabasites, and serpentinites (e.g., Keiter et al., 2011). The CBU has been separated into the "Upper Cycladic Blueschist Nappe" and the "Lower Cycladic Blueschist Nappe" on Milos Island; the Upper Nappe records peak pressure conditions above ~0.8 GPa (~2.0 GPa and 550 °C; Grasemann et al., 2018). Previous studies have reported a wide range of maximum P-T conditions for rocks from the Upper Cycladic Blueschist Nappe on different Cycladic islands [Sifnos: ~1.4 – 2.2 GPa and 450 – 550 °C (e.g., Schmädicke and Will, 2003; Groppo et al., 2009; Dragovic et al., 2012, 2015; Schliestedt and Matthews, 1987; Matthews and Schliestedt, 1984; Ashley et al., 2014; Spear et al., 2006); Tinos: ~1.4 – 2.6 GPa and ~450 – 550 °C (e.g., Bröcker et al., 1993; Lamont et al., 2020; Parra et al., 2002); Naxos: ~1.2 – 2.0 GPa and ~450 – 600 °C (e.g., Avigad, 1998; Peillod et al., 2017, 2021); Sikinos: ~1.1 – 1.7 GPa and ~ 500 °C (e.g., Augier et al., 2015; Gupta and Bickle, 2004)]. Some conventional thermobarometry (i.e., thermobarometry techniques that rely on chemical equilibrium) suggests that the CBU on Syros reached peak P-T conditions of ~1.5 GPa and ~500 °C (Ridley, 1984). Trotet et al. (2001a) and Laurent et al. (2018) suggest higher peak P-T conditions of ~2.0 - 2.4 GPa and ~500 - 550 °C; however, multi-mineral phase equilibria of marbles (Schumacher et al., 2008) and elastic thermobarometry of metabasites from Kini beach (Behr et al., 2018) support the original P-T estimates of ~1.5 GPa and 500 °C. Published exhumation P-T paths for the CBU on Syros are also highly variable, ranging from cooling during decompression, near-isothermal decompression, to cooling during decompression followed by reheating at moderate pressures (Laurent et al., 2018; Schumacher et al., 2008; Skelton et al., 2018; Trotet et al., 2001a). Because of these conflicting P-T paths, several models have been proposed to explain the exhumation history of the CBU, including coaxial vertical thinning (Rosenbaum et al., 2002), extrusion wedge tectonics (Keiter et al., 2011; Ring et al.,

2020), multiple cycles of thrusting and extension (Lister and Forster, 2016), continuous accretion and syn-orogenic extension

(Trotet et al., 2001a, b), and subduction channel exhumation (Laurent et al., 2016).

## 3. Field and Microstructural Observations

We studied four localities on Syros (Kalamisia, Delfini, Lotos, Megas Gialos; Fig. 1). Each locality exhibits multiple stages of mineral growth, and the same deformation and P-T progression. Kalamisia records blueschist facies metamorphism, and Delfini, Lotos, and Megas Gialos record blueschist-greenschist facies metamorphism. GPS coordinates of collected samples and their associated mineralogy are provided in the supplementary material (Supplementary Table S1). 1 – 4 samples from each locality were examined petrographically.

### 3.1 Kalamisia

Mafic rocks from Kalamisia preserve retrograde blueschist facies metamorphism (Fig. 1). Protoliths of Kalamisia rocks are fine-grained basalts. They exhibit an early foliation ($S_s$) characterized by relict blueschist and eclogite facies minerals. The early $S_s$ fabric is re-folded by upright folds ($F_{t1}$) with steeply dipping axial planes, NE-SW-oriented fold hinge lines, and NE-SW-oriented stretching lineations primarily defined by white mica, glaucophane, and epidote; this indicates syn-blueschist facies folding ($D_{t1}$).

Garnets in Kalamisia mafic samples occur as ~1 - 4 mm porphyroblasts (KCS70A, Supplementary Fig. S1), lack a well-defined internal foliation, and the $S_s$ foliation deflect around garnets. Glaucophane typically grows within pressure shadows and brittle fractures of garnet, and omphacite displays breakdown and alteration to glaucophane; this indicates retrograde glaucophane growth. Glaucophane inclusions within epidote are commonly oriented parallel to $S_s$, and no omphacite is observed as inclusions within epidote; these observations support epidote (ep1) growth during retrograde metamorphism.

### 3.2 Delfini Beach

Metasedimentary rocks (quartz-rich lenses intermixed with metavolcanics) at Delfini Beach show retrogression from eclogite- and blueschist- to greenschist facies (Fig. 1). Protoliths of Delfini rocks remain enigmatic, but may be graywackes or sandstones variably intermixed with mafic tuffitic intercalations. The rocks at Delfini exhibit an early foliation (also considered $S_s$) characterized by relict blueschist and eclogite facies minerals (garnet porphyroblasts, and foliation-parallel white mica, blue amphibole, and epidote) aligned in tight isoclinal folds ($F_s$) with shallow axial planes. This early fabric was locally retrogressed and re-folded by upright folds (considered $F_{t2}$) with steeply dipping axial planes, E-W-oriented fold hinge lines, and E-W-oriented stretching lineations primarily defined by white mica, chlorite, and actinolite (considered $D_{t2}$, Fig. 2a,b); this indicates folding under greenschist facies conditions. $D_{t2}$ folding was associated with boudinage of earlier-generation epidote parallel to the fold hinge lines, and simultaneous precipitation of new coarse-grained epidote (ep2), along

with quartz, calcite and iron oxides in boudin necks (Fig. 3). In some areas of tight $D_{t2}$ folding, a new generation of fine-grained epidote (also interpreted as ep2) grows within a newly developed crenulation cleavage ($S_{t2}$, Fig. 2c,d,e).

Garnets in Delfini metasedimentary samples occur as ~1 - 4 mm, partially chloritized porphyroblasts (KCS34, Fig. 2c), and as <1 mm garnets that are commonly found as inclusions within epidote (KCS1621, Supplementary Fig. S3). Foliation parallel epidotes (ep1) found within early blueschist-greenschist facies outcrops (KCS1621) range in size from ~0.5 – 5 mm (b-axis length), are strongly poikiloblastic, lack late greenschist facies inclusions such as chlorite, and commonly contain an internal foliation that is oblique to the external matrix $S_s$ foliation (Fig. 2f,g; Supplementary Fig. S3). Late epidote (ep2) crystals are found within sample KCS34 from the core of an upright fold ($F_{t2}$). During upright folding, a predominant portion of the rock is recrystallized to late-stage greenschist facies minerals, and contains new epidote (ep2) that is oriented parallel to the $S_{t2}$ crenulation cleavage. Ep2 crystals range from ~50 - 300 µm along the b-axis (Fig. 2c,d,e), tend to be euhedral (Fig. 2d,e), sometimes contain titanite inclusions (Fig. 2d), and show textural equilibrium with white mica and titanite that also formed in the $S_{t2}$ cleavage (Fig. 2d,e). Ep2 crystals are not poikiloblastic and rarely preserve quartz inclusions, thus only a few analyses were possible.

**3.3 Lotos Beach**

The rocks from Lotos Beach exhibit the same structural and petrological progression as those from Delfini (Fig. 1), showing retrogression from eclogite- and blueschist- to greenschist facies. Protoliths of Lotos rocks are primarily fine-grained volcanics. An early $S_s$ foliation was locally retrogressed and re-folded by upright $F_{t2}$ folds with steeply dipping axial planes, E-W-oriented fold hinge lines, and E-W-oriented stretching lineations primarily defined by white mica, chlorite, and actinolite ($D_{t2}$). $D_{t2}$ folding was associated with boudinage of earlier-generation epidote parallel to the fold hinge lines, and simultaneous precipitation of new coarse-grained epidote (ep2), along with quartz, calcite and iron oxides in boudin necks (Fig. 3).

Garnets in Lotos samples occur as ~1 - 3 mm chloritized porphyroblasts (e.g., KCS3), that deflect the external $S_s$ foliation (KCS3). Foliation parallel epidotes (ep1) found within early blueschist-greenschist facies outcrops (SY1402, SY1405, KCS2, KCS3) range in size from ~0.5 – 5 mm (b-axis length), are strongly poikiloblastic, and commonly contain an internal foliation that is oblique to the external matrix $S_s$ foliation (Supplementary Fig. S4). Boudinage of ep1 parallel to stretching lineations is common in thin sections (Supplementary Fig. S4).

**3.4 Megas Gialos**

The rocks from Megas Gialos exhibit the same structural and petrological progression as those from Lotos and Delfini Beaches (Fig. 1). Protoliths of Megas Gialos rocks remain enigmatic, but may be sediments intermixed with volcanics. Rocks from Megas Gialos show retrogression from eclogite- and blueschist- to greenschist facies. An early $S_s$ foliation was locally retrogressed and stretching lineations primarily defined by white mica, chlorite, and actinolite are E-W-oriented.

No garnets were found within the analyzed sample from Megas Gialos. Foliation parallel epidotes (ep1) found within early blueschist-greenschist facies outcrops range in size from ~0.5 – 3 mm (b-axis length), are strongly poikiloblastic, and

commonly contain an internal foliation that is oblique to the external matrix $S_s$ foliation (Supplementary Fig. S5). Boudinage
of ep1 parallel to stretching lineations is common in thin sections (Supplementary Fig. S5).

## 4. Methods

We determined P-T conditions using elastic thermobarometry and oxygen isotope thermometry. Raman spectroscopy
was used to measure Raman shifts of strained quartz inclusions entrapped within epidote or garnet, and a laser fluorination
line and a GasBench II coupled to a gas source mass spectrometer was used to measure oxygen isotope ratios of quartz and
calcite separates, respectively.

### 4.1 Raman Spectroscopy measurements

Our Raman spectroscopy measurements are taken from ~30 μm, ~80 μm, and ~150 μm thin and thick sections, that
consist of sections cut perpendicular to foliation ($S_s$) and parallel to stretching lineations (e.g., KCS1621), and perpendicular
to the $F_{t2}$ fold axial plane (KCS34). Quartz inclusions were measured from multiple epidotes and garnets within individual
sections (Supplementary Table S3). Measured quartz inclusions were small in diameter relative to the host, and were two-to-
three-times the inclusion radial distance from other inclusions, fractures, and the host exterior to avoid overpressures or stress
relaxation (Fig. 4a,b; Campomenosi et al., 2018; Zhong et al., 2020). No geometric corrections were applied (Mazzucchelli et
al., 2018).
Raman spectroscopy measurements of quartz inclusions within garnet and epidote were carried-out at Virginia Tech
(VT) and ETH Zürich (ETHZ) by using JY Horiba LabRam HR800 and DILOR Labram Raman systems, respectively.
Analyses at VT used an 1800 grooves mm$^{-1}$ grating, 100x objective with a 0.9 numerical aperture (NA), 400 μm confocal
aperture, and a 150 μm slit width. Raman spectra were centered at ~360 cm$^{-1}$. We used a 514.57 nm wavelength Ar laser, and
removed the laser interference filter for all analyses to apply a linear drift correction dependent on the position of the 116.04
cm$^{-1}$, 266.29 cm$^{-1}$, and 520.30 cm$^{-1}$ Ar plasma lines (Fig. DR4). Measurements at ETHZ used a 532 nm laser, an 1800 grooves
mm$^{-1}$ grating, a 100x objective with a 0.9 NA, a 200 μm confocal aperture, and a 300 μm slit width. Raman spectra were
centered at ~ 850 cm$^{-1}$.
All Raman spectra was reduced with a Bose-Einstein temperature-dependent population factor (Kuzmany, 2009). All
Raman bands were fit by using PeakFit v4.12 from SYSTAT Software Inc. A Gaussian model was used to fit Ar plasma lines
(only VT analyses), and a Voigt model was used to fit the quartz 128 cm$^{-1}$, 206 cm$^{-1}$, and 464 cm$^{-1}$ bands, epidote bands, and
garnet bands. Raman bands of quartz, epidote, and garnet, and Ar plasma lines were fit simultaneously, and a linear background
subtraction was applied during peak fitting. Baseline-to-baseline deconvolution of quartz and garnet bands was simple and
generally required fitting quartz bands and a few shoulder garnet bands. Deconvolution of quartz and epidote bands required
more complicated deconvolution; we followed a fitting approach similar to that described by Cisneros et al. (2020).

## 4.2 Inclusion and entrapment pressure calculations

The fully encapsulated inclusions preserve strain that causes the Raman active vibrational modes of inclusions to be shifted to higher or lower wavenumbers relative to minerals that are unstrained (fully exposed). We calculated the Raman shift(s) of inclusions ($\omega_{inc}$) relative to Raman shift(s) of an unencapsulated Herkimer quartz standard ($\omega_{ref}$) at ambient conditions ($\Delta\omega = \omega_{inc} - \omega_{ref}$) (Fig. 4). For VT analyses, $\omega_{inc}$ was measured relative to a Herkimer quartz standard that was analyzed 5 times prior to same day analyses. A drift correction was applied to $\omega_{inc}$ by monitoring the position of Ar plasma lines (Supplementary Tables S2; S3). For ETHZ analyses, a Herkimer quartz standard was analyzed 3 times prior to and after quartz inclusion analyses. A time-dependent linear drift correction was applied to $\omega_{inc}$ based on the drift shown by Herkimer quartz analyses that bracketed inclusion analyses ($< 0.2$ cm$^{-1}$).

We calculated residual inclusion pressures ($P_{inc}$) by using hydrostatic calibrations and by accounting for quartz anisotropy. To calculate a $P_{inc}$ from individual quartz Raman bands, we used pressure-dependent Raman shift(s) (P-$\Delta\omega$) of the quartz 128 cm$^{-1}$, 206 cm$^{-1}$, and 464 cm$^{-1}$ bands, that have been experimentally calibrated under hydrostatic stress conditions by using diamond anvil cell experiments (Schmidt and Ziemann, 2000). To account for quartz anisotropy, we calculated $P_{inc}$ from strains. Calculating quartz strains requires that the Raman shift of at least 2 quartz vibrational modes can be measured. When we were able to measure the quartz 128, 206 and 464 cm$^{-1}$ band positions of inclusions, we calculated strains from the $\Delta\omega$ of 3 bands. If only two bands were measured, we calculated strains from the $\Delta\omega$ of 2 bands (Supplementary Table S3). For the remaining analyses with low 128 and 206 cm$^{-1}$ intensities, we report $P_{inc}$ calculated from the 464 cm$^{-1}$ band hydrostatic P-$\Delta\omega$ relationship (Supplementary Table S3). Strains were determined from the $\Delta\omega$ of the 128 cm$^{-1}$, 206 cm$^{-1}$, and 464 cm$^{-1}$ quartz bands by using Strainman (Angel et al., 2019; Murri et al., 2018, 2019), wherein a weighted fit was applied based on the $\Delta\omega$ error associated with each quartz Raman band. Calculated strains were converted to a mean stress [$P_{inc} = (2\sigma_1 + \sigma_3)/3$] using the matrix relationship $\sigma_i = c_{ij}\varepsilon_j$, where $\sigma_i$, $c_{ij}$, and $\varepsilon_j$, are the stress, elastic modulus, and strain matrices, respectively. We used the $\alpha$-quartz trigonal symmetry constraints of Nye (1985) and quartz elastic constants of Wang et al. (2015).

We assumed constant mineral compositions for all modeling (epidote: $X_{ep} = 0.5$ and $X_{cz} = 0.5$; garnet: $X_{Alm} = 0.7$, $X_{Gr} = 0.2$, and $X_{Py} = 0.1$). Garnet compositions have a negligible effect on entrapment pressures ($P_{trap}$) because the thermodynamic and physical properties of garnet end-members are similar (e.g., Supplementary Table S8). Epidote composition has a greater effect on $P_{trap}$, but the compositional dependence is minor $< 1.5$ GPa (Cisneros et al., 2020). To account for epidote and garnet solid solutions, we implemented linear mixing of shear moduli and molar volumes (V). Ideal mixing of molar volumes has been shown to be an appropriate approximation for epidote-clinozoisite solid solutions (Cisneros et al., 2020; Franz and Liebscher, 2004). Garnet molar volumes were modeled using the thermodynamic properties of Holland and Powell (2011) (almandine and pyrope) and Milani et al. (2017) (grossular), and a Tait Equation of State (EoS) with a thermal pressure term. We used the shear moduli of Wang and Ji (2001) (almandine and pyrope) and Isaak et al. (1992) (grossular). Epidote molar volumes were modeled using the thermodynamic properties and shear moduli given by Cisneros et al. (2020), and a Tait EoS and thermal pressure term. Epidote and clinozoisite regressions are based on the P-V-T data of Gatta

et al. (2011) ($X_{ep} = 0.74$), and T-V and P-V data of Pawley et al. (1996) ($X_{ep} = 0$) and Qin et al. (2016) ($X_{ep} = 0.39$), respectively.
Clinozoisite and epidote have similar thermal expansivities but differing bulk moduli (Supplementary Table S4). To account
for the composition of epidotes used in P-V-T experiments, we normalized the composition of our unknown epidotes across
the compositional range of P-V experimental epidotes, i.e., the molar volume of our unknown epidote ($X_{ep} = 0.5$) is estimated
as 31 % ($X_{ep} = 0.74$) and 69 % ($X_{ep} = 0.39$) of each experimental epidote. Quartz molar volumes were modeled using the
thermodynamic properties and approach of Angel et al. (2017a). Entrapment pressures were calculated from residual quartz
$P_{inc}$ by using the Angel et al. (2017b) 1D elastic model equation, and a MATLAB program available in Cisneros and Befus
(2020) that implements mixing of shear moduli and molar volumes. A comparison of entrapment pressures calculated from
the Cisneros and Befus (2020) MATLAB program and EoSFit-Pinc (Angel et al., 2017b) is given in Supplementary Table S4;
entrapment pressure calculations of mineral end-members accounts for the reproducibility of molar volume and elastic
modeling calculations.

### 4.3 Stable isotope measurements

Samples were measured by using a ThermoElectron MAT 253 isotope ratio mass spectrometer (IRMS) at the
University of Texas at Austin. Quartz $\delta^{18}O$ values were measured by laser fluorination (Sharp, 1990), and ~2.0 mg of quartz
were used in each analysis. Quartz from samples SY1613, SY1617, and SY1623 was duplicated to determine isotopic
homogeneity and reproducibility. An internal quartz Lausanne-1 standard ($\delta^{18}O = +18.1‰$) was analyzed with all samples to
evaluate precision and accuracy. All $\delta^{18}O$ values are reported relative to standard mean ocean water (SMOW), where the $\delta^{18}O$
value of NBS-28 is +9.65‰. Measurement precision based on the long-term reproducibility of standards is ± 0.1 ‰ (1 σ).
Precision of Lausanne-1 on the day of analysis was ± 0.3 ‰ (1 σ), whereas samples reproduced with a precision of ± 0.1 ‰
(1 σ) or better (Supplementary Table S5). Calcite $\delta^{18}O$ values were measured on a Thermo Gasbench II coupled to a
ThermoElectron 253 mass spectrometer. Each analysis used 0.25 - 0.5 mg of calcite that was loaded into Exetainer vials,
flushed with ultra-high purity helium, and reacted with 103 % phosphoric acid at 50 °C for ~2 hours. Headspace $CO_2$ was then
transferred to the mass spectrometer. Samples were calibrated to an in-house standard, NBS-18, and NBS-19. Measurement
precision is ± 0.04 ‰ (1 σ) based on the long-term reproducibility of standards.

### 4.4 Stable isotope temperature calculations

Temperatures derived from stable isotope measurements were calculated by using the Sharp and Kirschner (1994)
quartz-calcite oxygen isotope fractionation calibration (A = 0.87 ± 0.06; equation A1; Supplementary Table S5). Isotopic
equilibrium was assumed for all samples. Several observations support that this assumption is appropriate: 1) duplicate $\delta^{18}O$
analysis of quartz and calcite grains give the same isotopic value, suggesting grain isotopic homogeneity, 2) the stage of
deformation that these mineral pairs are related to is not affected by further deformation in either outcrop or thin section, and
3) all quartz-calcite pairs suggest a similar temperature of isotopic equilibrium.
Temperature errors from quartz-calcite oxygen isotope measurements were calculated through the square-root of the
summed quadratures of all sources of uncertainty (equations A2, A3). These uncertainties included $\delta^{18}O$ value errors of quartz
and calcite of $\pm$ 0.1 ‰ (1 σ) and $\pm$ 0.04 ‰ (1 σ), respectively, and errors associated with the Sharp and Kirschner (1994)
quartz-calcite oxygen isotope fractionation calibration (A parameter).

**4.5 Electron probe measurements**

Electron probe analyses were carried-out at ETHZ using a JEOL JXA-8230 Electron Probe Microanalyzer (EPMA).
The EPMA is equipped with five wavelength-dispersive spectrometers. Epidote and pyroxene were analyzed for Si, Al, Na,
Mg, Ca, Cr, K, Ti, Fe, and Mn on TAP (Si, Al), TAPH (Al, Ca), PETJ (Ca, Cr), PETL (K, Ti), and LIFH (Fe, Mn) crystals.
Beam parameters included a 20 nA beam current, 10 µm beam size, and a 15 keV accelerating voltage. All elements were
measured for 30 s on peak and a mean atomic number background correction was applied. Primary calibration standards used
included: albite (Si, Na), anorthite (Al, Ca), synthetic forsterite (Mg), chromite (Cr), microcline (K), synthetic rutile (Ti),
synthetic fayalite (Fe), and synthetic pyrolusite (Mn). Mole fraction expressions from Franz and Liebscher (2004) were used
to calculate epidote ($X_{ep}$), clinozoisite ($X_{cz}$), and tawmawite ($X_{taw}$) compositions. Further information on mineral chemistry
calculations is available in Supplementary Table S6. Garnets were analyzed for Al, Ca, Mn, Fe, Mg on TAP (Al), PETJ (Ca),
LIFL (Mn), LIFH (Fe), and TAPH (Mg) crystals. Si was calculated stoichiometrically. X-ray maps were collected with a 50
nA beam current, 15 keV accelerating voltage, 100 ms dwell time, and 5 µm (KCS34 Garnet 1) and 4 µm (KCS34 Garnet 3)
step sizes. X-ray maps were reduced using CalcImage (Probe for EPMA).

**5. Thermobarometry Results**

Determined pressures were categorized into three groups according to outcrop and microstructural context (Fig. 5;
Fig. 7; Supplementary Table S3): garnet growth near peak metamorphic conditions (Group 1), growth of foliation-parallel
epidote during blueschist-greenschist facies metamorphism (ep1, Group 2), and late-stage epidote growth in the new
crenulation ($S_{t2}$) associated with $F_{t2}$ folds during greenschist facies metamorphism (ep2, Group 3). New ep2 growth is also
supported by the mineral chemistry of different epidote generations within the $S_{t2}$ crenulation. Epidotes show a progressive
chemical evolution that is recorded by an early generation epidote inclusion in titanite that occurs parallel to $S_{t2}$ ($X_{ep} \cong 0.1$),
the ep2 core ($X_{ep} \cong 0.5$), and the ep2 rim ($X_{ep} \cong 0.8$) (Fig. 2g; Supplementary Table S6).
The entrapment temperature ($T_{trap}$) of quartz inclusions in garnet (garnet growth temperature) is estimated as 500 -
550 °C; this is based on good agreement between previous studies on the maximum temperature reached by CBU rocks from
Syros (e.g., Laurent et al., 2018; Ridley, 1984; Schumacher et al., 2008; Skelton et al., 2018; Trotet et al., 2001a). $T_{trap}$ for the
ep2 population (Group 3) is deduced from oxygen isotope thermometry of quartz-calcite boudin-neck precipitates. The mean
temperature from quartz-calcite pairs from boundin necks is 411 $\pm$ 23 °C (n = 4, Supplementary Table S5). $T_{trap}$ for the ep1
population (Group 2) is estimated as being intermediate between garnet and ep2 growth (~400 - 500 °C). As shown by qtz-in-
ep isomekes (constant $P_{inc}$ lines along which fractional volume changes of an inclusion and host are equal), the assumed $T_{trap}$
has a minimal effect on $P_{trap}$ (Fig. 7a; Cisneros et al., 2020).

**5.1 Kalamisia**

Group 1 quartz-inclusions-in-garnet record a mean $P_{inc}$ of $600 \pm 78$ MPa (Fig. 5; Supplementary Table S3). This
corresponds to an entrapment pressure ($P_{trap}$) of $1.43 - 1.49 \pm 0.14$ GPa (n = 5), at an estimated $T_{trap}$ between 500 - 550 °C (Fig.
7a, Supplementary Table S3). Group 2 quartz-inclusions-in-ep1 record a mean $P_{inc}$ of $544 \pm 57$ MPa, corresponding to a $P_{trap}$
of $1.43 \pm 0.12$ GPa (n = 6) at an estimated $T_{trap}$ of 450 °C. No Group 3 epidotes are found within our analyzed section from
Kalamisia.

**5.2 Delfini**

Group 1 records a mean $P_{inc}$ of $731 \pm 54$ MPa (Fig 5; Supplementary Table S3). This corresponds to a $P_{trap}$ of $1.66$ -
$1.72 \pm 0.10$ GPa (n = 22), at an estimated $T_{trap}$ between 500 - 550 °C (Fig. 7a, Supplementary Table S3). Group 2 records a
mean $P_{inc}$ of $518 \pm 52$ MPa, corresponding to a $P_{trap}$ of $1.38 \pm 0.11$ (n = 5) at an estimated $T_{trap}$ of 450 °C. Group 3 records a
mean $P_{inc}$ of $343 \pm 23$ MPa, corresponding to a $P_{trap}$ of $0.98 \pm 0.05$ GPa (n = 3) at 411 °C (Supplementary Table S3).

**5.3 Lotos**

Group 1 records a mean $P_{inc}$ of $751 \pm 76$ MPa (Fig 5; Supplementary Table S3). This corresponds to a $P_{trap}$ of $1.70$ -
$1.76 \pm 0.14$ GPa (n = 2), at an estimated $T_{trap}$ between 500 - 550 °C (Fig. 7a; Supplementary Table S3). Group 2 records a mean
$P_{inc}$ of $531 \pm 78$ MPa, corresponding to a $P_{trap}$ of $1.41 \pm 0.17$ (n = 15) at an estimated $T_{trap}$ of 450 °C. No Group 3 epidotes were
analyzed from Lotos.

**5.4 Megas Gialos**

Group 2 records an average $P_{inc}$ of $494 \pm 29$ MPa (Fig. 5), corresponding to a $P_{trap}$ of $1.33 \pm 0.03$ (n = 6) at an estimated
$T_{trap}$ of 450 °C (Fig. 7a; Supplementary Table S3). No Group 1 garnets or Group 3 epidotes were analyzed from Megas Gialos.

**6. Discussion**

**6.1 Elastic thermobarometry pressure groups**

Group 1 garnets either lack an internal foliation or contain a weak foliation that is defined by inclusions oblique to
the $S_s$ fabric, which indicates a previous stage of deformation (Fig. 2c; Supplementary Figs. S1, S2, S3). Garnets record similar
pressures, regardless of the location of quartz inclusions (Fig. 6, Supplementary Table S3). Pyroxene inclusions within different
garnet zones (core: $X_{jd} \approx 0.84$, rim: $X_{jd} \approx 0.81$) also show no difference in composition, which is consistent with qtz-in-grt
barometry results (Delfini: KCS1621, Supplementary Table S6). Group 2 epidotes (ep1) overgrow garnets, are aligned parallel
to the Ss foliation but sometimes preserve an internal foliation that is oblique to $S_s$, and lack late greenschist facies inclusions
(Fig. 2f,g; Supplementary Figs. S1, S3, S4, S5). Group 3 epidotes (ep2, KCS34, Fig. 2c, d, e) are short in length, are aligned
parallel to a late $S_{t2}$ crenulation, contain minimal quartz inclusions, and only record Group 3 pressures, independent of the
position of quartz inclusions within epidotes.
Based on these observations, the Group 1 $P_{trap}$ estimates from the qtz-in-grt barometer record high-P conditions on
Syros associated with prograde-to-peak garnet growth, and the Group 2 and 3 $P_{trap}$ estimates from the qtz-in-ep barometer
record epidote growth during early blueschist-greenschist facies retrogression (ep1, $D_{t1}$) and subsequent $D_{t2}$ deformation (ep2),
respectively. We interpret the low-P epidote group (Group 3) to be associated with $D_{t2}$ folding, and best recorded in areas that
experienced late greenschist facies mineral growth due to enhanced deformation and/or fluid influx during this stage of
deformation (e.g., core of $F_{t2}$ fold).

## 6.2 Comparison of peak pressure constraints for the CBU on Syros and Sifnos

Based on qtz-in-grt measurements (Group 1), our $P_{trap}$ calculations suggest maximum P conditions of ~1.6 - 1.8 GPa
were reached by the CBU on Syros. Garnets from metasedimentary and metavolcanic rocks record the statistically highest $P_{trap}$
(~1.5 - 1.8 GPa), whereas garnets from metamafic rocks (Kalamisia) record the lowest $P_{trap}$ (~1.3 - 1.6 GPa) (Fig. 7a). Several
observations support that the qtz-in-grt barometry results record max P conditions of the CBU on Syros: 1) quartz inclusion
measurements across core-to-rims of garnets that show prograde growth (decreasing Mn), show no systematic change in $P_{trap}$
(Fig. 6), 2) max pressures from this study are equivalent to qtz-in-grt barometry results from prograde-to-peak eclogites and
blueschists (non-retrogressed) from the CBU on Syros (Behr et al., 2018), 3) retrograde ep1 pressures, do not exceed those
recorded by qtz-in-grt barometry, and 4) several studies from the CBU have used garnets to constrain max pressures, suggesting
that garnets are suitable for constraining maximum pressures (e.g., Laurent et al., 2018; Dragovic et al., 2012, 2015; Groppo
et al., 2009). We herein discuss our qtz-in-grt barometry results as max pressures constraints, but acknowledge that we may
have missed high-P rims that have been found in other studies from the CBU on Syros (e.g., Laurent et al., 2018). We present
a compilation of previous P-T constraints on CBU rocks from Syros and Sifnos, Greece, and discuss how our $P_{trap}$ constraints
compare with previous studies.
Elastic thermobarometry, mineral stability constraints, and multi-phase equilibrium modeling results from Sifnos
CBU rocks suggest maximum P conditions of ~1.8 ± 0.1 GPa (Ashley et al., 2014), ~1.4 ± 0.2 GPa (Matthews and Schliestedt,
1984), and ~2.0 - 2.2 GPa (Dragovic et al., 2012, 2015; Groppo et al., 2009; Trotet et al., 2001a), respectively. Elastic
thermobarometry (Ashley et al., 2014) and garnet modelling results (Dragovic et al., 2012, 2015; Groppo et al., 2009) from
Sifnos, suggest near isobaric conditions during garnet growth. The results of Ashley et al. (2014) are commonly cited as
evidence that the CBU reached high pressure conditions (≥ 2.0 GPa, from elastic thermobarometry); however, their $P_{trap}$
calculations were carried out by using fits to quartz molar volume (P-T-V) data that have recently been re-evaluated (Angel et

al., 2017a) . Improved fits to quartz molar volume experiments "soften" quartz, and remodeling $P_{inc}$ values from Ashley et al. (2014) reduces maximum mean $P_{trap}$ conditions to ~1.6 ± 0.1 GPa (Fig. 7b, Supplementary Table S7).

Elastic thermobarometry, mineral stability constraints, glaucophane-bearing marble mineral equilibria, and multi-phase equilibria modeling results from Syros CBU rocks suggest peak pressure conditions of ~1.5 ± 0.1 GPa (Behr et al., 2018), ~1.4 - 1.9 GPa (Ridley, 1984), ~1.5 GPa (Schumacher et al., 2008), and ~1.9 - 2.4 GPa (Laurent et al., 2018; Skelton et al., 2018; Trotet et al., 2001a), respectively. Elastic thermobarometry results from prograde-to-peak eclogites and blueschists from Syros, Greece were reduced using the approach outlined in Ashley et al. (2016), wherein a correction to $P_{trap}$ is applied based on the assumed $T_{trap}$. Recent studies suggest that not using a temperature-dependent $P_{trap}$ correction produces suitable results that accurately reproduce experimental conditions of quartz entrapment by garnet (Bonazzi et al., 2019; Thomas and Spear, 2018). Recalculation of the Behr et al. (2018) $P_{inc}$ data (no temperature-dependent $P_{trap}$ correction) results in a mean $P_{trap}$ of ~1.7 ± 0.1 GPa (Fig. 7b, Supplementary Table S8). The re-evaluation of data from Ashley et al. (2014) and Behr et al. (2018) suggests that our results are in good agreement with previous elastic thermobarometry constraints, and that to date, no qtz-in-grt elastic thermobarometry results suggest pressures ≥ 2.0 GPa.

Different methodologies applied to CBU rocks from Syros have resulted in a wide range of maximum P estimates. Schumacher et al. (2008) used mineral-equilibria modeling of glaucophane-bearing marbles to place constraints on maximum P-T conditions. Maximum P-T conditions are constrained by the presence of glaucophane + CaCO3 + dolomite + quartz, which suggests that the marbles exceeded the albite/Na-pyroxene + dolomite + quartz → glaucophane + CaCO3 reaction, but did not cross the dolomite + quartz → tremolite + CaCO3 or the glaucophane + aragonite-out reactions. The mineral reaction constraints suggest maximum P-T conditions of ~ 1.5 - 1.6 GPa and 500 °C for the CBU marbles. Ridley (1984) used the stability of paragonite and lack of kyanite to deduce max P constraints of ~1.4 -1.9 GPa. Trotet et al. (2001b, 2001a), Laurent et al. (2018), and Skelton et al. (2018) employed thermodynamic phase-equilibria modeling and supplementary methods to constrain P-T conditions for CBU rocks from Syros. Skelton et al. (2018) used the Powell and Holland (1994) Thermocalc database, Trotet et al. (2001b, 2001a) used the Berman (1991) thermodynamic database and the TWEEQC approach, and Laurent et al. (2018) used empirical thermobarometry, GrtMod (Lanari et al., 2017), and isochemical phase diagrams. Trotet et al. (2001b, 2001a), Laurent et al. (2018), and Skelton et al. (2018) found high-P conditions for the CBU (≥ 1.9 GPa), and results from Laurent et al. (2018) suggest some rocks reached conditions as high as 2.2 ± 0.2 GPa. Results from Laurent et al. (2018) suggest most garnet growth occurred at ~1.7 GPa and 450 ± 50 °C; however, some garnet modeling results suggest that garnet rims grew at ~2.4 GPa and 500 - 550 °C, albeit errors are increasingly large for these results (± 0.4 - 0.9 GPa). These errors reflect the spacing between garnet isopleths (optimal P-T conditions), that result from uncertainties in chemical analyses.

Some GrtMod results suggest prograde core and rim garnet growth at ~1.8 GPa and 475 °C, and ~2.4 GPa and 475 °C, respectively (sample SY1418 from; Laurent et al., 2018); however, the optimal P-T conditions for garnet rims have large errors and plot within uncertainty of garnet core conditions. Garnet results from another sample (SY1401) suggest core and rim garnet growth at ~1.8 GPa and 475 °C, and ~2.4 GPa and 550 °C, respectively. Sample SY1401 is collected from the same locality as ours (Kalamisia), but our qtz-in-grt results from this study suggest that garnets from this outcrop record the

statistically lowest P$_{trap}$. It is possible, however, that we did not sample the same rocks as Laurent et al. (2018), or that we have
not found or analyzed garnets that record high pressures.

Previous studies have also suggested that pressures ≥ 2.0 GPa are unreasonable for Syros because paragonite is

abundant in CBU rocks, but kyanite has not been reported. This suggests that CBU rocks did not cross the reaction paragonite
→ jadeite50 + kyanite + H$_2$O (~1.9 - 2.0 GPa); however, we recognize that the occurrence of kyanite may require high
Al$_2$O$_3$:SiO$_2$ ratios  for metabasites (e.g., Liati and Seidel, 1996), and that the pressure of this reaction is compositionally
dependent. Pseudosections of eclogite CBU rocks show that kyanite would not be found in these bulk compositions below
~2.3 GPa (Skelton et al., 2018). It is possible that the high-P conditions found in previous studies may be real, but may only
be recorded locally within some eclogite blocks.

In general, phase stability relationships (e.g., Matthews and Schliestedt, 1984; Ridley, 1984; Schumacher et al., 2008)

and qtz-in-grt barometry results are in good agreement, but do not agree with high-pressure results (≥ 1.9 GPa) deduced from
thermodynamic modeling using approaches such as GrtMod and TWEEQC.  The difference between our results and those of
previous studies is important to reconcile, because the maximum P conditions reached by the CBU has considerable
implications for the internal architecture of the CBU, its geodynamic evolution, and the mechanisms that can accommodate
exhumation mechanisms of high-P subduction zone rocks from Syros. A comparison of qtz-in-grt barometry with
thermodynamic modeling results from samples that record high pressures would be appropriate for further testing differences
between the two techniques.

## 6.3 Comparison of exhumation P-T conditions

Previous studies have presented varying P-T paths and associated exhumation histories for Syros CBU rocks (Fig.

7a; Laurent et al., 2018; Schumacher et al., 2008; Skelton et al., 2018; Trotet et al., 2001a). We present a compilation of
previous P-T constraints and interpretations and discuss how our results compare with previous studies.

Schumacher et al. (2008) do not provide quantitative constraints for the retrograde P-T path (schematic), and samples

do not have structural context; however, the authors suggest that a "cold" P-T path during exhumation is required for Syros
CBU rocks based on the occurrence of lawsonite + epidote assemblages across Syros, and the P-T path required to avoid
crossing the lawsonite → kyanite + zoisite reaction (Fig. 7b). The authors suggest that exhumation of CBU packages occurred
shortly after juxtaposition near peak metamorphic conditions.

Both Trotet et al. (2001a, 2001b) and Laurent et al. (2018) constrain high-P conditions for the CBU (> 2.0 GPa),

however, their proposed exhumation histories differ. Trotet et al. (2001b) suggested that CBU eclogites, blueschists and
greenschists underwent different T-t histories during exhumation and were juxtaposed late along ductile shear zones. Laurent
et al. (2018) suggested that the entire CBU reached peak metamorphic conditions of ~2.2 GPa, and that units that preserved
blueschist facies assemblages underwent cooling during decompression, whereas rocks of southern Syros from lower structural
levels experienced isobaric heating (~550 °C) at mid-crustal depths (~1.0 GPa) followed by subsequent cooling. Laurent et al.

(2018) interpreted reheating to indicate that CBU rocks on Syros reached high-P conditions, and then transitioned from a forearc to back-arc setting at ~1.0 GPa, thus experiencing a period of increasing temperatures.

Skelton et al. (2018) also estimated peak and exhumation P-T conditions of rocks from Fabrikas (southern Syros), and interpreted exhumation of the CBU within an extrusion wedge (Ring et al., 2020). The authors constrained maximum P-T conditions of ~1.9 GPa and 540 °C, and retrograde conditions of ~1.4 – 1.6 GPa and 510 - 520 °C (blueschist facies) and ~0.3 GPa and 450 °C (greenschist facies) based on Thermocalc end-member activity modeling (Powell and Holland, 1994). Retrograde blueschist conditions (inferred from garnet growth) are similar between their estimates and ours, but greenschist facies conditions vastly differ. However, Skelton et al. (2018) focused on greenschist facies outcrops wherein metamorphism occurred locally over short length scales (e.g. ~10 - 100 m), adjacent to late-stage brittle normal faults. We interpret our $D_{t2}$ stage of greenschist facies metamorphism to pre-date late-stage normal faulting that has been attributed to Neogene block rotations (Cooperdock and Stockli, 2016) or possible coeval granitoid magmatism during Miocene back-arc extension (Keiter et al., 2011).

Gyomlai et al. (2021) estimate max and retrograde P-T conditions, but from metasomatic rocks from the Kampos belt in northern Syros. The authors estimated maximum T conditions of 561 ± 78 °C, and two retrograde pressure-temperature conditions: 1.02 ± 0.15 GPa and 505 ± 155 °C, and 1.03 ± 0.11 GPa and 653 ± 27 °C. The retrograde pressures are reasonable (~1.0 ± 0.1 - 0.2 GPa), but the max temperatures raise questions that the authors discuss. Specifically, temperatures above ~600 °C (at ~1.0 GPa) would lead to serpentine breakdown (Guillot et al., 2015; Wunder and Schreyer, 1997); however, serpentine is abundant across Syros. The authors used the 505 ± 155 °C temperature constraint, and a temperature below 600 °C, to suggest their studied rocks reached temperatures between 500 – 600 °C at ~ 1.0 GPa. Several other studies on retrograde metasomatic rocks from Kampos constrain P-T conditions: ~1.17 – 1.23 GPa and 500 – 550 °C (Breeding et al., 2004), ~ 0.60 – 0.75 GPa and 400 – 430 °C (Marschall et al., 2006), and ~ 1.20 GPa and 430 °C (Miller et al., 2009). Breeding et al. (2004) did not constrain a temperature, but used an estimated temperature from Trotet et al. (2001a), and constrained a pressure of ~1.17 – 1.23 GPa at the estimated T of ~500 – 550 °C using Thermocalc V. 3.2. Marschall et al. (2006) used the garnet-clinopyroxene thermometer and Thermocalc V. 3.01 to calculate temperatures, and estimated a pressure based on jadeite + $SiO_2 \rightarrow$ albite reaction. Miller et al. (2009) used Perple_X and the thermodynamic database of Holland and Powell (1998) to calculate P-T conditions from reaction zones. In general, most studies indicate cooling during decompression for metasomatic rocks from Kampos, with the exception of interpretations by Gyomlai et al. (2021); however, the large uncertainty of their temperature estimate (505 ± 155 °C) makes it difficult to differentiate between cooling during decompression, isothermal decompression, or re-heating.

Our results show that rocks from Kalamisia, Delfini, Lotos, and Megas Gialos, reached peak P-T conditions and underwent cooling during retrograde blueschist and greenschist facies metamorphism (Fig. 7a). Peak P-T conditions of the CBU are ~1.6 - 1.8 GPa and 500 - 550 °C (Group 1 qtz-in-grt $P_{trap}$ estimates), indicating a subduction zone geothermal gradient of ~9 - 10 °C km$^{-1}$ at ~55 - 60 km (assuming 30 MPa km$^{-1}$). Group 2 and 3 qtz-in-ep $P_{trap}$ estimates indicate geothermal gradients of ~10 °C km$^{-1}$ and ~12 °C km$^{-1}$ at ~47 and 33 km depths, respectively (Fig. 7a), demonstrating a similar P-T trajectory during

exhumation. We do not have a temperature constraint for the ep1 population; however, we consider cooling during
decompression from garnet growth (~500 – 550 °C) to ep2 growth (~400 °C), to be the most likely P-T path for CBU rocks
from Syros. Isothermal decompression from ~1.8 GPa and ~500 – 550 °C to ~ 1.0 GPa, would lead to terminal lawsonite
breakdown above ~ 450 °C and produce kyanite + zoisite (Hamelin et al., 2018; Schumacher et al., 2008); however, kyanite
has not been found on Syros, therefore requiring temperatures below ~450 °C at ~ 1.0 GPa. It is possible that sluggish kinetics
did not lead to lawsonite breakdown, but given the prevalent evidence of retrograde deformation on Syros and the extensive
presence of retrograde overprinting/mineral growth, we consider kinetic-limitations to be unlikely. Furthermore, the chemical
evolution of amphiboles (magnesio-riebeckite → winchite → actinolite) suggests that CBU rocks from Syros followed a cold
P-T path during decompression (c.f., Kotowski et al., 2020). Our P-T constraints are also inconsistent with reheating to ~550
°C and 1.0 GPa, wherein amphibolite facies mineralogy may be stable. Our samples and the sample from which Laurent et al.
(2018) determined reheating (SY1407), preserve no mineralogical evidence for having reached epidote-amphibolite facies
(Fig. 7b; e.g., pargasite/hornblende, biotite/muscovite). Instead, the matrix mineralogy of sample SY1407 (glaucophane,
phengite, rutile) suggests that these rocks formed under a cold geothermal gradient, rather than in a back-arc setting with an
elevated geothermal gradient. Laurent et al. (2018) suggest that sample SY1407 records albite-epidote-blueschist conditions,
a field metamorphic facies that can expand to higher T conditions; however, a pseudosection created for a similar bulk
composition suggests that the determined P-T constraints (~1.0 GPa and 550 °C) are within epidote-amphibolite facies (Trotet
et al., 2001a). Furthermore, results from sample SY1407 of Laurent et al. (2018) sometimes disagree when using local vs. bulk
compositions for modeling. Models that use bulk compositions and consider Mn suggest that the core and mantle of the garnet
record P-T conditions of ~1.8 GPa and 475 °C, whereas models that use local compositions or do not consider Mn suggest that
the garnets do not record conditions above ~1.0 GPa (model residuals are lower using local bulk composition models).
Our results suggest that rocks from different Syros outcrops record similar peak and exhumation P-T conditions, but
experienced different extents of deformation and thus recrystallization during exhumation. The similar peak pressures (> 0.8
GPa) between different Syros outcrops suggests that these rocks belong to the Upper Cycladic Blueschist Nappe (Grasemann
et al., 2018), even though in some cases significant retrogression overprinted indicators that would suggest these rocks reached
P conditions above ~0.8 GPa. The observation of similar P-T conditions reached at different locations is inconsistent with
results that suggest individual P-T paths for rocks that preserve different metamorphic facies (Trotet et al., 2001b, a), and
different sections of the CBU (Laurent et al., 2018); however, we do not have T constraints for rocks from southern Syros. Our
results are in better agreement with a P-T evolution resembling that of Schumacher et al. (2008), and a geothermal gradient of
~10 – 12 °C km$^{-1}$ that has also been proposed for CBU rocks from Sifnos, Greece (Schmädicke and Will, 2003).

**6.4 Limitations of elastic thermobarometry**

Elastic thermobarometry has rapidly gained interest due to its limited dependence on mineral and fluid chemistry.
Recent hydrostatic experiments that grow garnet around quartz have also shown the quartz-in-garnet barometer is accurate
from ~ 0.8 – 3.0 GPa (± 0.1 - 0.2 GPa, Thomas and Spear, 2018; Bonazzi et al., 2019). The results suggest that the applied 1-
dimensional elastic model that assumes a spherical inclusion and isotropic inclusion-host pairs (Guiraud and Powell, 2006;
Angel et al., 2017b), and the currently applied EoS' (Angel et al., 2017a; Holland and Powell, 2011; Milani et al., 2017),
sufficiently replicate the elastic behaviour of an isotropic mineral (quartz) in a near isotropic host (garnet). Nonetheless,
multiple secondary processes may affect quartz-in-garnet entrapment conditions: 1) mineral anisotropy (e.g., Murri et al.,
2018), 2) inclusion shape effects (e.g., Cesare et al., 2021; Mazzucchelli et al., 2018), 3) relaxation adjacent to fractures or the
host exterior, or overpressures adjacent to other inclusions (e.g., Zhong et al., 2020), 4) non-ideal tensile strain (e.g., Cisneros
and Befus, 2020), or 5) non-elastic strain (i.e., viscous strain, e.g., Zhang, 1998). We propose that none of these processes have
affected our quartz-in-garnet barometry results for the following reasons: 1) $P_{inc}$ values calculated from different quartz bands
and by accounting for anisotropy (strains) center around the hydrostatic stress lines (1:1 line, Fig. 5), and $P_{inc}$ calculated from
strains changes the final $P_{trap}$ by < 0.2 GPa (relative to $P_{inc}$ calculated from the 464 cm$^{-1}$ band). 2) Near spherical quartz
inclusions were analysed to minimize shape effects, and measurements were taken from the center of quartz inclusions to avoid
stress effects at inclusion-host boundaries. 3) Quartz inclusions were a minimum two-to-three-times the radial distance away
from fractures, cleavage, and the host exterior, or other inclusions to minimize under- or overpressures, respectively. 4) All
quartz inclusions from this study exist under compression, thus tensile strain limits are not relevant. 5) The maximum estimated
temperature of CBU rocks from Syros is ~ 500 - 550 °C, and garnet flow laws predict that viscous creep of garnet occurs above
~ 650 °C at geologic strain rates (Wang and Ji, 2001; Ji and Martignole, 1994); therefore, viscous strain of garnet is unlikely
to have occurred. Considering the current state-of-knowledge in elastic thermobarometry, we propose that our pressure results
have been minimally influenced by secondary effects.
In contrast, the quartz-in-epidote barometer is less studied. Recent studies have explored the suitability of using an
isotropic elastic model to model the elastic evolution of two anisotropic minerals (Cisneros et al., 2020). Results showed that
an isotropic elastic model suitably simulates the pressure evolution of two anisotropic minerals during heating, and that the
calculated entrapment pressures agree with independent thermobarometry constraints. However, it is unknown if isotropic
elastic models correctly simulate the elastic evolution of anisocoric mineral pairs during compression, and additional processes
may influence the entrapment pressures calculated from quartz-inclusions-in-epidote: 1) the orientation of quartz inclusions
relative to the orientation of epidote, and 2) the material properties of epidote (i.e., at what conditions does viscous creep
become important for epidote). 1) Cisneros et al. (2020) showed that the orientation of quartz inclusions relative to epidote
may have had a minimal effect on the elastic evolution of quartz-epidote pairs, but the orientation of quartz and epidote were
not determined. We hypothesize that in this study, the mutual orientation of quartz-epidote inclusion-host pairs had a minimal
effect on the calculated entrapment pressures. If the mutual orientation of quartz-epidote pairs had a large effect, we expect
that the $P_{inc}$ calculated from different quartz-inclusions-in-epidote would exhibit significant scatter; however, $P_{inc}$ values from
different quartz-inclusions-in-epidote are similar, and $P_{inc}$ values from different quartz bands and strains, center around the
hydrostatic stress line (Fig. 5). The $P_{inc}$ scatter from different quartz-inclusions-in-epidote (same ep population, e.g., ep2) and
the $P_{inc}$ variation from different quartz bands and strains, generally does not exceed that of quartz-inclusions-in-garnet. The
minimal $P_{inc}$ variation between quartz-inclusions-in-epidote from the same epidote population may result from the orientation

of quartz and epidote parallel to the primary foliation. The orientation of quartz-epidote pairs may lead to a bulk stress tensor that produces minimal orientation-dependent effects, or the lower bulk modulus of epidote (relative to garnet) may result in a small stress anisotropy. 2) No epidote flow law exists (to the best of our knowledge); therefore, the temperature at which viscous strain will be important for epidote is unknown. Nonetheless, in contrast to garnet (isotropic), evidence for viscous creep in epidote can be observed in thin section. In epidotes from this study, we have observed no thin-section scale evidence of dislocation creep; however, μm-scale viscous creep in epidote adjacent to quartz inclusions cannot be excluded.

**6.5 Implications for exhumation mechanisms**

Our results indicate that the CBU followed a "cooling during decompression" P-T trajectory that required a heat sink at depth to cool rocks during exhumation. Cooling could be achieved under a steady-state subduction zone thermal gradient with slab-top temperatures similar to those of warm subduction zones, such as in Cascadia (e.g., Syracuse et al., 2010; Walowski et al., 2015). This would suggest that exhumation was achieved parallel to the subducting plate, in a subduction channel geometry prior to core-complex formation. Results from this study cannot differentiate between extrusion wedge models ("extrusion" of a wedge of CBU rocks within a subduction channel) that require a kinematically necessary thrust fault at the base (the subducting slab) and a kinematically necessary normal fault at the top (upper plate), and other general subduction channel models (e.g., Ring et al., 2020). Subduction channel and extrusion models have slight differences, i.e., the extrusion wedge model calls for a specific geometry that should produce opposing shear sense indicators at distinct locations that define the base (subduction plate) and top (upper plate) of the wedge (within a subduction channel). A subduction channel model has a looser definition (without a specific geometric structure) that merely reflects the plate interface structure (discrete or broad interface), and does not require this deformation. Because we do not present sufficient kinematic information in this study to differentiate these models, we prefer to use a general "subduction channel" model nomenclature, to indicate that we interpret CBU rocks to have been exhumed parallel to the subducting plate, within a broad, viscous shear zone that defines the subduction interface.

During this phase of exhumation, CBU rocks remained within a cold forearc until they reached the mid-crust (~1.0 GPa), and exhibit a progressive change in kinematics, from N-S stretching lineations during subduction (e.g., Behr et al., 2018; Laurent et al., 2016; Philippon et al., 2011), to lineations that swing towards the NE (this study, Roche et al., 2016: Sifnos) and E-W during exhumation (c.f., Kotowski and Behr, 2019; Laurent et al., 2016). We propose that N-S ($D_s$) lineations (subduction-related) and exhumation-related upright folds that generate NE ($D_{t1}$) and E-W ($D_{t2}$) extension parallel to fold hinge lines, document the transition from subduction to exhumation as rocks turn the corner to be exhumed within the subduction channel. Stretching lineations in the footwall of the North and West Cycladic Detachment Systems have top-to-the- NE and SW orientations, respectively (e.g., Brichau et al., 2007; Grasemann et al., 2012; Jolivet et al., 2010; Mehl et al., 2005). The inferred P-T conditions and kinematics of our studied samples are consistent with Syros recording early deformation and metamorphism within a forearc setting, whereas adjacent Cycladic islands that border the North and West Cycladic Detachment Systems record late-stage kinematics and greenschist facies metamorphism that capture the CBU transition to a

warmer back-arc setting (e.g., Laurent et al., 2016; Ring et al., 2020; Roche et al., 2016; Schmädicke and Will, 2003). Our
data suggests that during generation of exhumation-related upright folds ($D_{t1-t2}$), rocks from the CBU on southern Syros (below
the Kampos nappe) followed similar P-T conditions exhumation (Fig. 7). It is unclear if the upper Kampos nappe exhibited
the same deformation because it preserves less structural coherency; however, rocks from Kampos and southern Syros seem
to have experienced similar P-T conditions during exhumation. Rocks from different sections of the CBU may have reached
peak P conditions at different times, and thus experienced the same exhumation-related deformation at different times
(Kotowski et al., 2020); however, our data suggest that rocks from different sections of the southern CBU on Syros were
exhumed within a forearc setting up to ~33 km depth. We propose that CBU on Syros may not record back-arc deformation
until the Vari detachment accommodated exhumation of the CBU at ~ 10 – 8 Ma (~5 - 7 km depth, Soukis and Stockli, 2013).
Back-arc related deformation occurs directly adjacent to the Vari detachment, as evidenced by semi-brittle to brittle cataclastic
deformation (greenschist facies) that affects the Upper Unit and the underlying CBU (Soukis and Stockli, 2013).

## 7. Conclusions

This work highlights the potential of using elastic thermobarometry in combination with structural (macro and micro)

and petrographic constraints, to better constrain P-T conditions of challenging rock assemblages. Our results allow us to place
robust P-T constraints on distinct textural fabrics that are related to well-constrained outcrop scale structures. In particular, the
work highlights how the qtz-in-ep barometer is well suited for constraining formation conditions of epidote, a common mineral
that is found within a large range of geologic settings and P-T conditions. Combining the qtz-in-ep barometer with other elastic
thermobarometers (e.g., qtz-in-grt) allows determination of protracted P-T histories from minerals that record different
geologic stages within single rocks samples.

Our new results show that CBU rocks from Syros, Greece, experienced similar P-T conditions during subduction and

exhumation, inconsistent with results that suggest different P-T histories for CBU rocks for Syros or increasing temperatures
during exhumation. Our targeted stages of deformation and metamorphism suggest that CBU rocks from Syros record cooling
during decompression, consistent with exhumation within a subduction channel and early deformation and metamorphism
within a forearc (at least to ~33 km depth), prior to Miocene core-complex formation and transition to a warmer back-arc
setting.

## Appendix A: Stable isotope temperature error calculations

Temperature errors from oxygen isotope measurements were calculated through the square-root of the summed

quadratures of all sources of uncertainty. These uncertainties included error of $\delta^{18}O$ values of quartz (qtz) and calcite (cc) of $\pm$
0.1 ‰ (1 σ) and $\pm$ 0.04 ‰ (1 σ), respectively, and errors associated with the Sharp and Kirschner (1994) quartz-calcite oxygen

isotope fractionation calibration (A parameter). Errors from the sum of propagated analytical errors, were propagated through the empirical calibration of quartz-calcite oxygen isotope fraction that was used for temperature calculations:

$$\Delta_{qtz-cc} = \frac{A \times 10^6}{T^2} \#A1$$

where A = 0.87 ± 0.06 (1 σ). The square-root of the summed quadratures is expressed as:

$$\sigma_T = \sqrt{\sigma_A^2 \left(\frac{\partial T}{\partial A}\right)^2 + \sigma_{\Delta_{qtz-cc}}^2 \left(\frac{\partial T}{\partial \Delta_{qtz-cc}}\right)^2} \#A2$$

$$\sigma_T = \sqrt{\sigma_A^2 \left(\frac{0.5 * 10^3}{\sqrt{A} * \sqrt{\Delta_{qtz-cc}}}\right)^2 + \sigma_{\Delta_{qtz-cc}}^2 \left(-0.5 * \frac{\sqrt{A} * 10^3}{\Delta_{qtz-cc}^{1.5}}\right)^2} \#A3$$

## Author Contribution

All authors contributed to this manuscript. M. Cisneros developed the epidote barometer, collected the data, and wrote the manuscript. J. Barnes, W. Behr, A. Kotowski, D. Stockli, and K. Soukis helped with conceiving the project, field work, and writing.

## Acknowledgements

We thank J. Schumacher and V. Laurent for constructive reviews that helped improve this manuscript, and F. Rossetti for editorial handling and additional comments that helped improve this manuscript. We thank N. Raia for field work assistance, J. Allaz for assistance on the microprobe at ETH Zürich, and C. Farley and R. Bobnar for access to the Raman Spectrometer at Virginia Tech. This work was supported by a GSA Student Research Grant and a Ford Foundation Fellowship awarded to M.C, an NSF Graduate Research Fellowship awarded to A.K., and NSF Grant (EAR-1725110) awarded to J.B., W.B., and D.S.

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

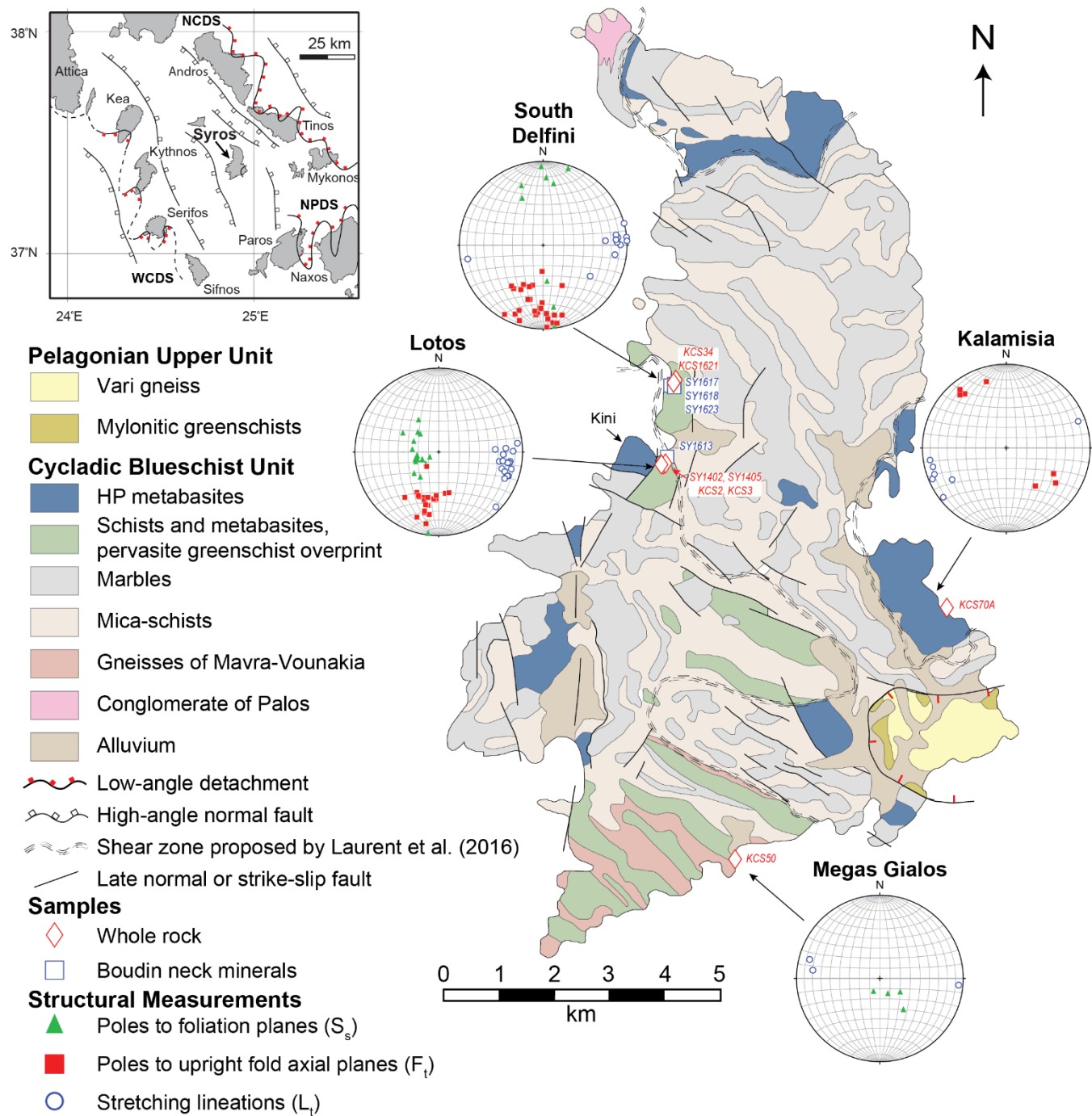


**Figure 1. Simplified geologic map of Syros, Greece [modified from Keiter et al. (2011)]. Inset map shows Syros relative to the North and West Cycladic, and Naxos-Paros Detachment Systems (NCSD, WCDS, NPDS, modified from Grasemann et al., 2012). Shear zones within the CBU and the Vari detachment are after Laurent et al., 2016 and Soukis and Stockli (2013), respectively. Stereonets from each studied outcrop are shown, and arrows indicate the outcrop location.**

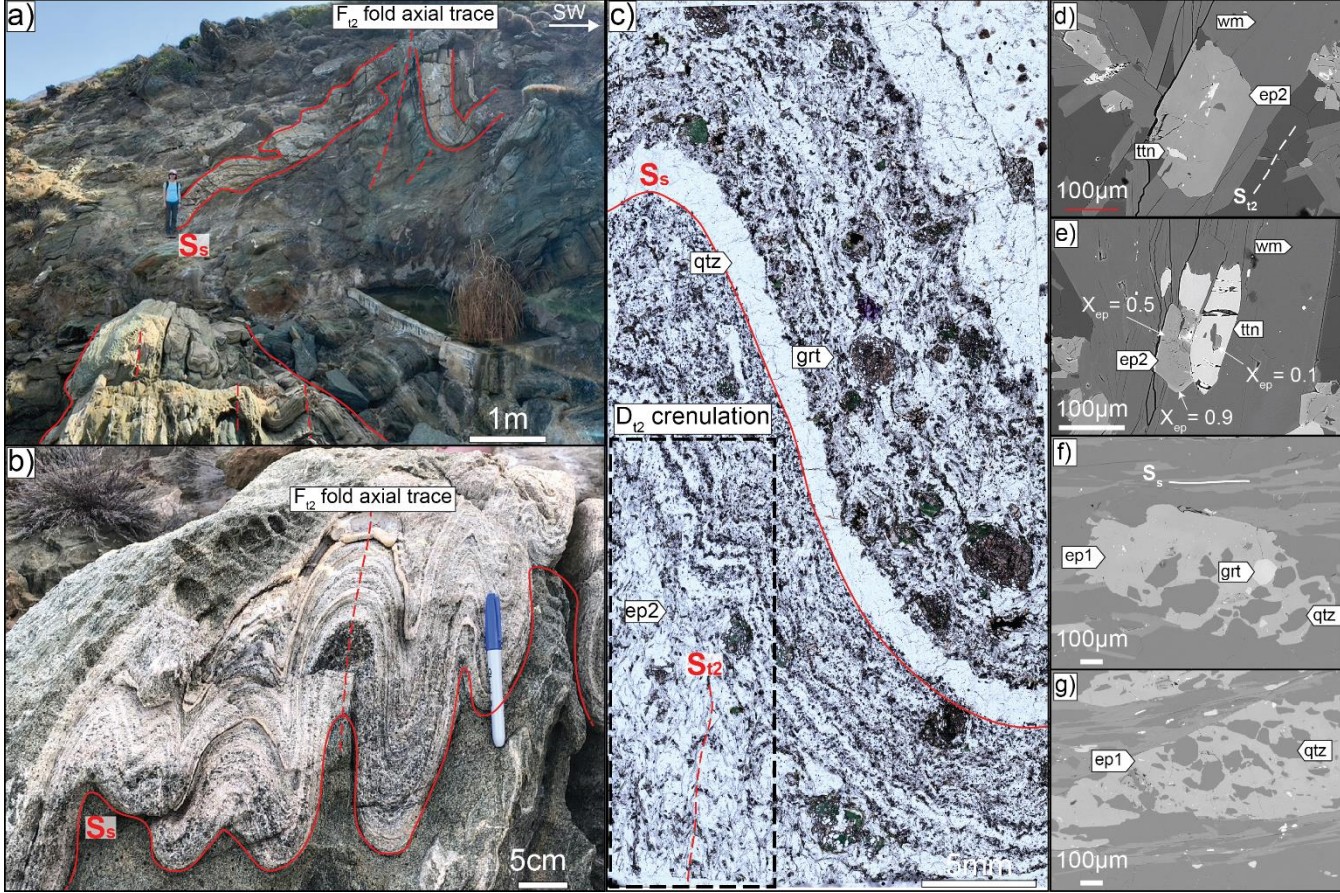

858

Figure 2. Outcrop, micrograph, and electron images showing stages of retrograde deformation present in southern Delfini. a) Upright folds ($F_{t2}$) that refold the primary $S_s$ foliation. b): Core of $F_{t2}$ folds (below Fig. 2a, KCS34). c): Plane light image of sample KCS34; sample cut perpendicular to the $F_{t2}$ fold axial plane. Epidotes (ep2) from the upright fold exhibit recrystallization as indicated by alignment with a late $S_{t2}$ crenulation, and a reduction in inclusions and grain size. d) Ep2 with late titanite (ttn) inclusions. Ep2 is parallel to white mica (wm) that defines $S_{t2}$ (KCS34). e) Ep2 in textural equilibrium with ttn (KCS34). f) Ep1 parallel to $S_s$, with garnet (grt) and quartz (qtz) inclusions that do not define an internal foliation (KCS1621). g) Poikiloblastic ep1 parallel to $S_s$, with a weak internal foliation defined by qtz (KCS1621).










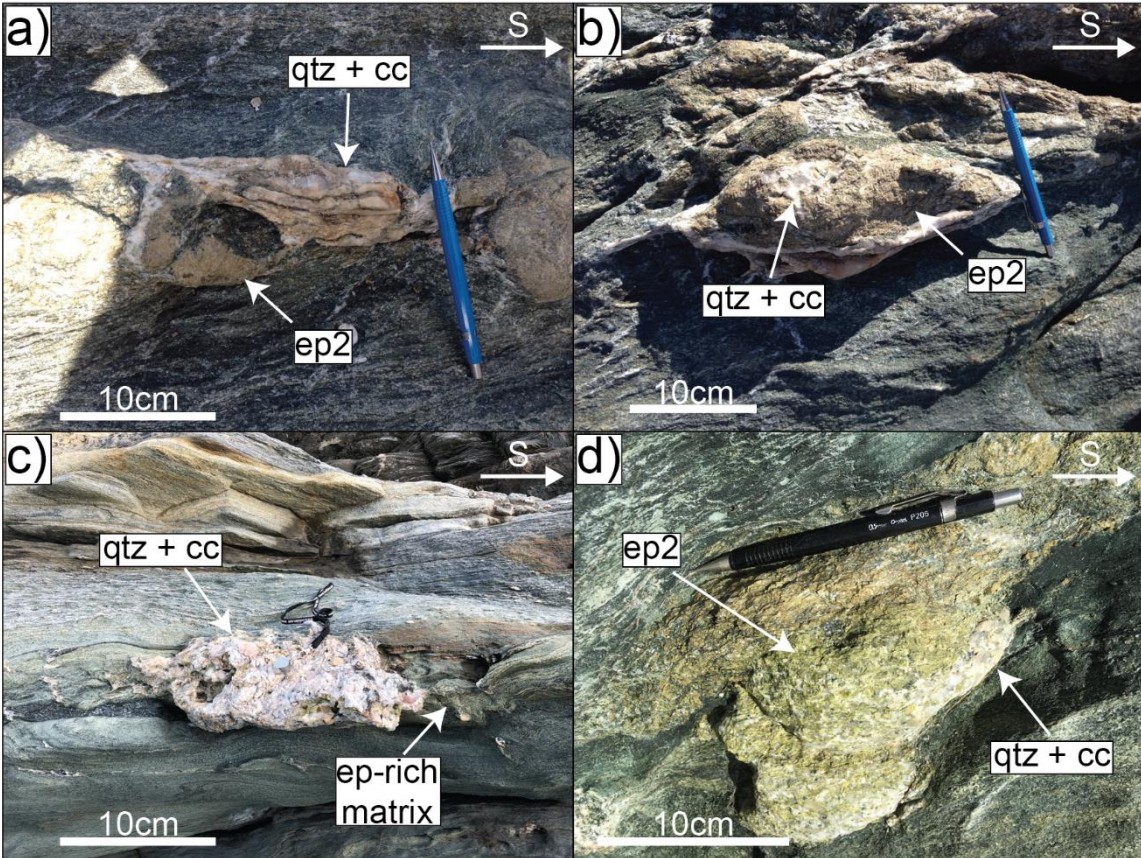

**Figure 3. Outcrop photos of epidote boudins sampled for oxygen isotope thermometry. a) SY1613 (Lotos), b) SY1617 (Delfini), c)**
**SY1618 (Delfini), d) SY1623 (Delfini).**

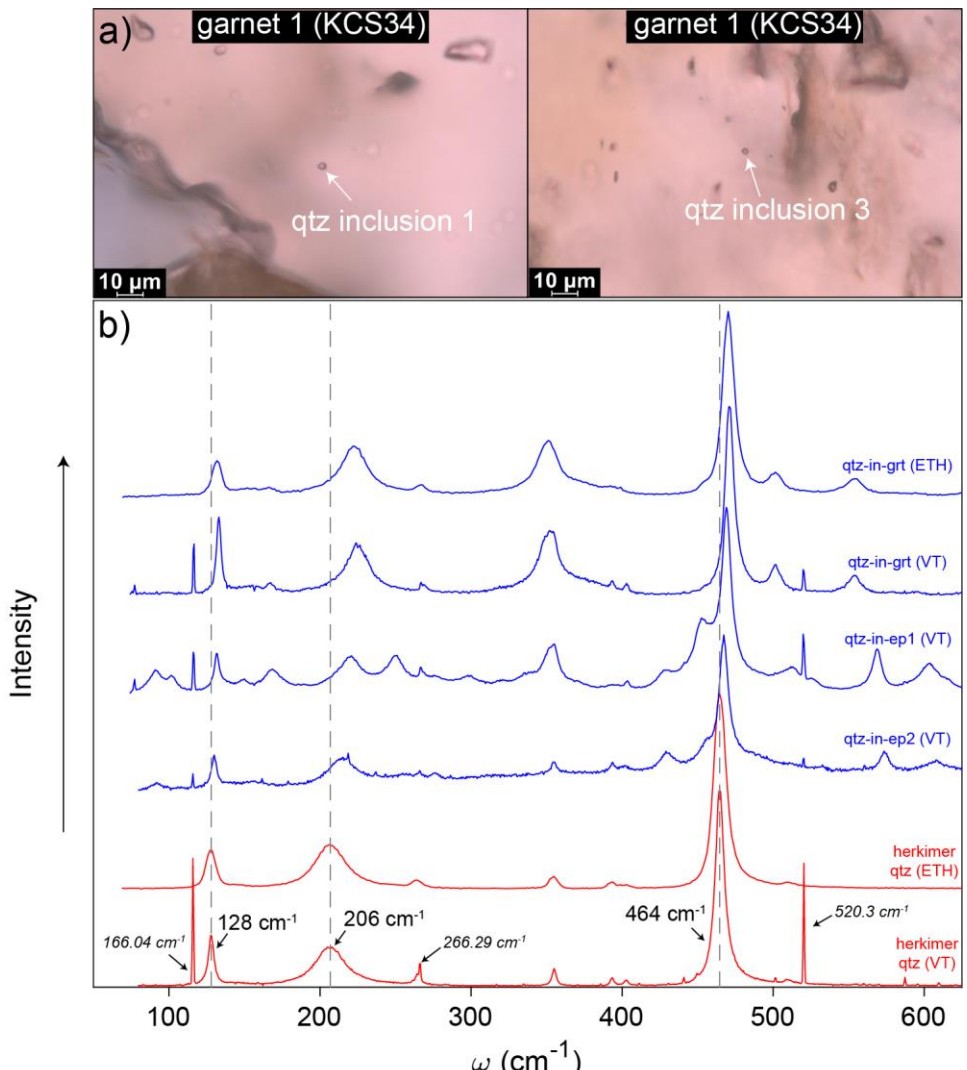

**Figure 4. Photomicrographs of measured quartz inclusions in garnet from Delfini (a) and Raman spectrums of unstrained Herkimer**
**quartz and strained quartz inclusions (b). b) Shown for comparison are Herkimer quartz (red) and quartz inclusion (blue)**
**measurements from Virginia Tech and ETH Zürich. Quartz bands and Ar plasma lines (only VT analyses) are numerically labelled.**

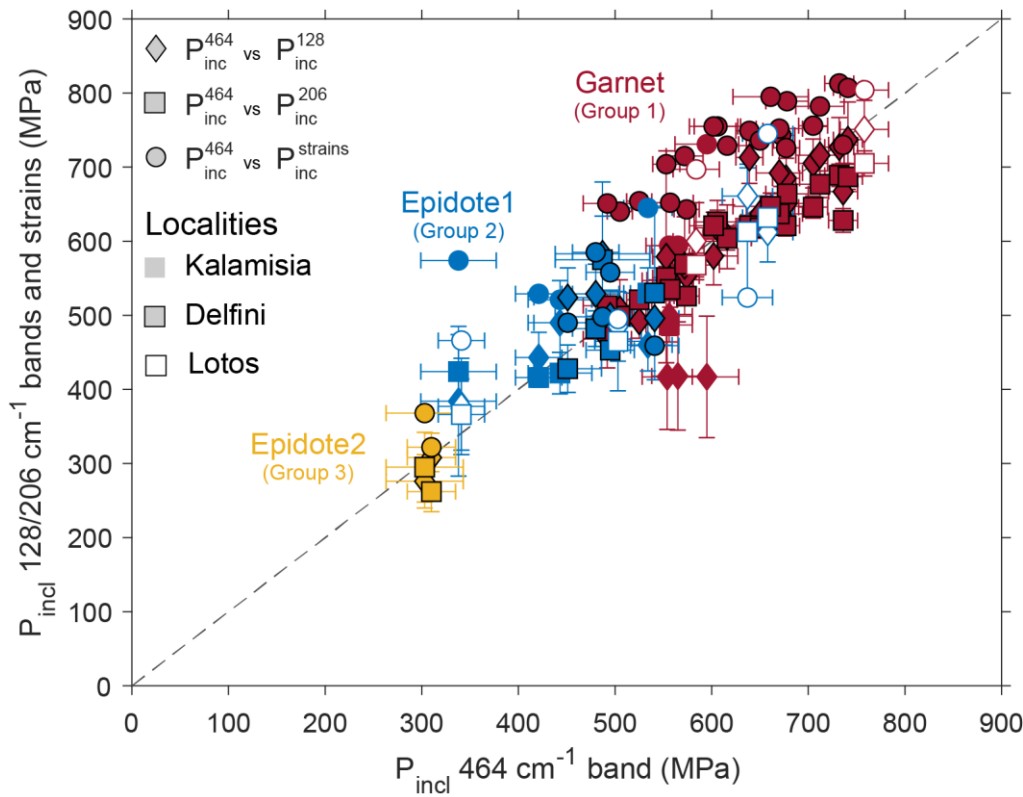


**Figure 5. Comparison of P$_{inc}$ determined from different quartz bands using hydrostatic calibrations, and by using phonon-mode Grüneisen tensors (strains). Red, blue, and yellow symbols indicate qtz-in-grt (Group 1), qtz-in-ep1 (Group 2), and qtz-in-ep2 (Group 3) results, respectively. Diamonds, squares, and circles indicate $P_{inc}^{464}$ vs $P_{inc}^{128}$, $P_{inc}^{464}$ vs $P_{inc}^{206}$, and $P_{inc}^{464}$ vs $P_{inc}^{strains}$ results, respectively. No border, filled, and open symbols indicate analyses from Kalamisia, Delfini, and Lotos samples, respectively.**














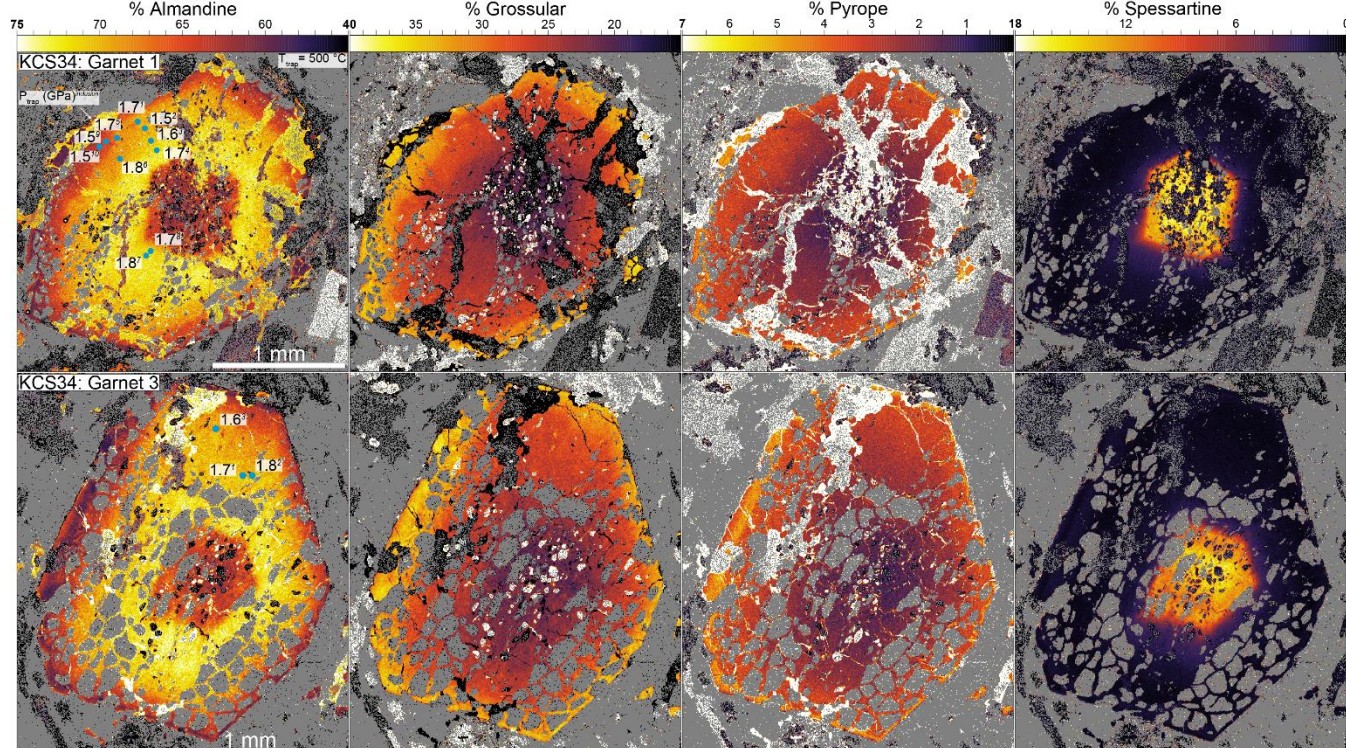


**Figure 6. Compositional x-ray maps of two garnets from sample KCS34 (Delfini). Blue dots indicate the location of measured inclusions; systematic $P_{trap}$ differences are not observed across garnets ($P_{trap}$ units are GPa, calculated at $T_{trap} = 500\ ^\circ C$.). Subscripts indicate the inclusion number (see Supplementary Table S3).**



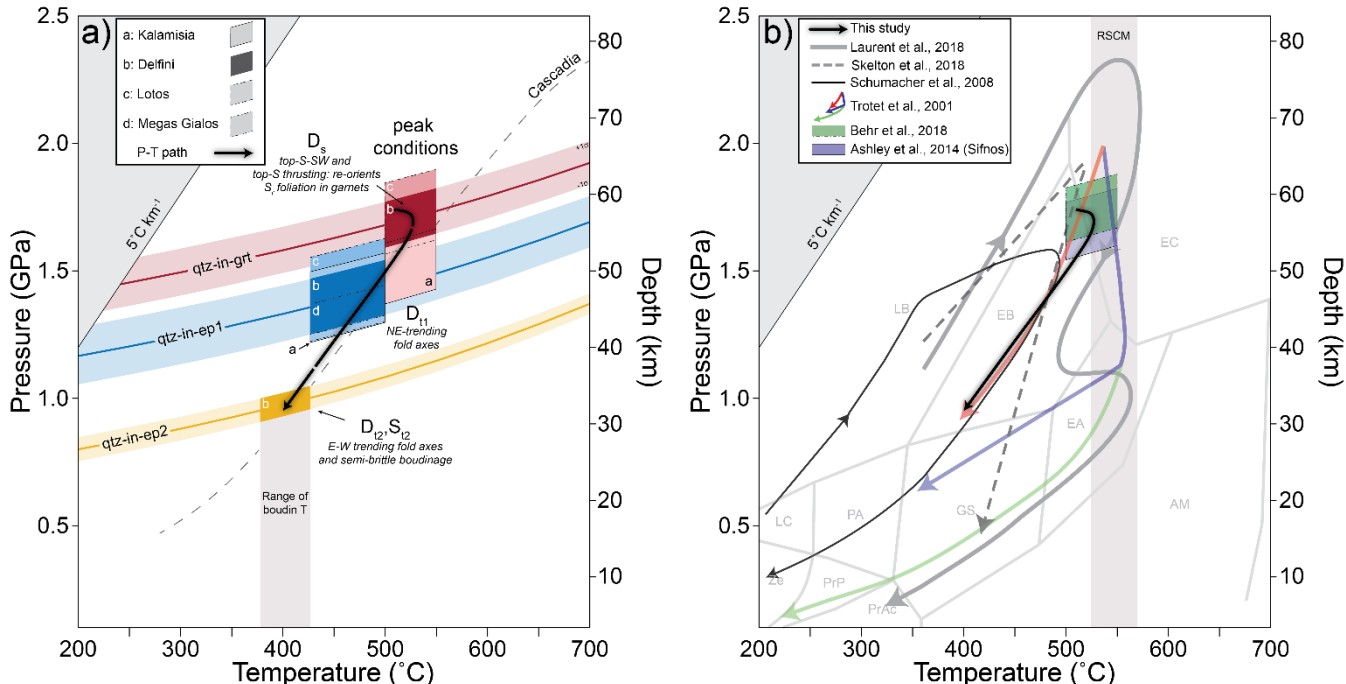

**Figure 7. (a) P-T conditions deduced from elastic thermobarometry and oxygen isotope thermometry superimposed on modeled Cascadia slap-top geotherm (Syracuse et al., 2010) and b) reference P-T conditions. (a) $P_{trap}$ from Groups 1, 2, and 3, that reflect peak (qtz-in-garnet), retrograde blueschist-greenschist facies (qtz-in-ep1), and late greenschist facies (qtz-in-ep2) conditions. Solid red, blue, and yellow lines and rectangles are the $P_{trap}$ isomekes (calculated from the mean residual inclusion pressure of each group) and our best-estimate entrapment conditions, respectively. Transparent lines are $P_{trap}$ errors (1σ around the mean) for analyses from Delfini samples. Grey box bounds the range of temperatures calculated from oxygen isotope thermometry of quartz-calcite boudin neck precipitates. b) Recalculated $P_{trap}$ values from Behr et al. (2018) (Syros) and Ashley et al. (2014) (Sifnos) and are shown in purple (solid border) and green (dashed border) rectangles, respectively. Metamorphic facies are taken from (Peacock, 1993). Metamorphic facies fields (Peacock, 1993): zeolite (ZE), prehnite-pumpellyite (PrP), prehnite-actinolite (PrAc), pumpellyite-actinolite (PA), lawsonite-chlorite (LC), greenschist (GS), lawsonite-blueschist (LB), epidote-blueschist (EB), epidote-amphibolite (EA), amphibolite (AM), eclogite (EC). RSCM = Raman Spectroscopy of Carbonaceous Material (data from Laurent et al., 2018).**