# Peer review of "Insights from elastic thermobarometry into exhumation of high- 2 pressure metamorphic rocks from Syros, Greece"

_Solid Earth, 2020_

## Short Comment (SC1) · 20 Sep 2020

Ioannis Baziotis

ibaziotis@aua.gr

1. Lines 230-235: Within the "electron probe measurements" be more specific. In particular, clarify which elements measured on which spectrometer. Additionally, give the element measured (in parenthesis) every after standard.

2. Line 234: What you mean by "primary standards".

3. line 344: Are you sure that the approach TWEEQC estimated equilibrium conditions?

4. Please define within the text the Xep and Xcz. 5. The authors measured epidote and

omphacite using EPMA, but i cannot find any text about omphacite chemistry. Right?

---

## Author Comment (AC1) · 22 Sep 2020

We thank I. Baziotis for comments that help clarify analytical methods in this manuscript. The comments are addressed below, and these changes will be incorporated into the revised version of the manuscript. Replies are italicized, and changes made that will be made to the text are highlighted in red (also italicized).

1. Lines 230-235: Within the "electron probe measurements" be more specific. In particular, clarify which elements measured on which spectrometer. Additionally, give the element measured (in parenthesis) every after standard.

[Figure]

*We have clarified the crystals used for each element and the "primary calibration" standards that were assigned to each element.*

*"The EPMA is equipped with five wavelength-dispersive spectrometers. Epidote and omphacite were analyzed for Si Al, Na, Mg, Ca, Cr K, Ti, Fe, and Mn on TAP (Si, Al), TAPH (Na, Mg), PETJ (Ca, Cr), PETL (K, Ti), and LIFH (Fe, Mn) crystals."*

*"Primary calibration standards used included: albite (Si, Na), anorthite (Al, Ca), synthetic forsterite (Mg), chromite (Cr), microcline (K), synthetic rutile (Ti), synthetic fayalite (Fe), and synthetic pyrolusite (Mn)."*

2. Line 234: What you mean by "primary standards".

*We refer to primary standards as the "primary calibration standards" that are used to calibrate x-ray intensities of specific elements during microprobe measurements. We would refer to secondary standards as those standards whose composition is known, but that are not assigned as a primary standard for a specific element (i.e., the standard x-ray intensity for a specific element is not used to calibrate the x-ray intensities of unknowns); the secondary standard can be used to evaluate the accuracy of measurements.*

*We will change the text to read "primary calibration standards" for clarity:*

*"Primary calibration standards used included: albite (Si, Na), anorthite (Al, Ca), synthetic forsterite (Mg), chromite (Cr), microcline (K), synthetic rutile (Ti), synthetic*

*fayalite (Fe), and synthetic pyrolusite (Mn)."*

3. line 344: Are you sure that the approach TWEEQC estimated equilibrium conditions?

*Given that the TWEEQ results are not our results, we prefer to not speculate on whether the results represent equilibrium conditions. The authors (Trotet et al., 2001) provide several lines of evidence (mineral habit, textural observations, and the intersection of independent reactions) that support equilibrium conditions; however, we do note that retrograde rocks from the CBU on Syros commonly preserve multiple stages of protracted mineral growth within a single thin section (e.g., see descriptions for sample KCS34 from Delfini), perhaps making equilibrium assumptions more difficult to prove valid.*

4. Please define within the text the Xep and Xcz.

*This is a good point for clarity, we use the mol fraction expressions from Franz and Liebscher (2004) for epidote ($X_{ep}$), clinozosite ($X_{cz}$), and tawmawite ($X_{taw}$), wherein $X_{ep}$, $X_{cz}$, and $X_{taw}$ are:*

$$X_{ep} = \frac{Fe^{3+}}{Fe^{3+} + Al + Cr^{3+} - 2}$$

$$X_{cz} = \frac{Al - 2}{Fe^{3+} + Al + Cr^{3+} - 2}$$

$$X_{taw} = \frac{Cr^{3+}}{Fe^{3+} + Al + Cr^{3+} - 2}$$

*More mineral chemistry calculation details are provided in Supplementary Table S6. We note that $X_{taw}$ does not exceed 0.01 for any of our measured epidotes.*

*This additional information will be added to the end of section 4.5 (electron probe measurements):*

*"Mol fraction expressions from Franz and Liebscher (2004) were used to calculate epidote ($X_{ep}$), clinozosite ($X_{cz}$), and tawmawite ($X_{taw}$) compositions. Further information on mineral chemistry calculations are available in Supplementary Table S6."*

5. The authors measured epidote and omphacite using EPMA, but i cannot find any text about omphacite chemistry. Right?

*The chemistry of omphacite inclusions in garnet is mentioned on lines 275 – 277. Further information on the chemistry of omphacite inclusions in garnet can be found in Supplemental Table S6. The chemistry of omphacite inclusions is not a significant portion of the story we present in the manuscript, and is therefore not extensively discussed in the main text. We will add a brief clarification on the omphacite inclusion (in garnet) mineral chemistry that supports similar entrapment pressures determined from quartz inclusions within different garnet zones:*

*"Omphacite inclusions within different garnet zones (core: $X_{jd} \approx 0.84$, rim: $X_{jd} \approx 0.81$) also show no difference in composition, which is consistent with qtz-in-grt barometry results (Delfini: KCS1621, Supplementary Table S6)."*

References:

Franz, G. and Liebscher, A.: Physical and Chemical Properties of the Epidote Minerals–An Introduction–, Reviews in Mineralogy and Geochemistry, 56(1), 1–81, doi:10.2138/gsrmg.56.1.1, 2004.

Trotet, F., Vidal, O. and Jolivet, L.: Exhumation of Syros and Sifnos metamorphic rocks (Cyclades, Greece). New constraints on the P-T paths, European Journal of Mineralogy, 13(5), 901–902, doi:10.1127/0935-1221/2001/0013-0901, 2001.

---

## Referee Comment (RC1) · John Schumacher (Referee) · 6 Nov 2020

This is an excellent paper. Well written and organized. The data is of very high quality. I actually have no real critique. The retrograde P-T derived from the detailed work presented here is the major result of this paper, and it is very similar to the one my colleagues and I derived based on the phase relations in glaucophane-bearing marbles and associated rocks twelve years ago. If I were to run these calculations again using updated solution models the results would be nearly identical to the results presented here. Consequently, I am very enthusiastic about these results.

My only very minor criticism is the reference to "metasedimentary" rocks. There should

be a description of the kind of sediments they are interpreted to be. Too commonly, this term gets interpreted as shales (pelitic schists). There are certainly sedimentary rocks on Syros: (1) carbonates (marbles) and (2) very minor quartzites. In a few places, there are rocks that could have shale-like protoliths, but, most of the other rocks are either magmatic, volcanic or volcano-clastic that span a range of basic to felsic compositions.

―――――――――――――――――――――

---

## Referee Comment (RC2) · Valentin Laurent (Referee) · 18 Nov 2020

The manuscript "Insights from elastic thermobarometry into exhumation of high-pressure metamorphic rocks from Syros, Greece " by Miguel Cisneros, Jaime D. Barnes, Whitney M. Behr, Alissa J. Kotowski, Daniel F. Stockli, and Konstantinos Soukis submitted for publication in Solid Earth combines elastic barometry (quartz-in-garnet and quartz-in-epidote barometry) with oxygen isotope thermometry to quantify the pressure-temperature (P-T) evolution of retrograde metamorphic rocks of the Cycladic Blueschist Unit (CBU). The work is well structured and presents interesting P-T data, especially the ones obtained from epidote and Qtz/Calcite boudins as they

allow to add robust constraints on the retrograde evolution of the CBU. However, there are important deficiencies in their bibliographic review that need to be addressed by considering and citing relevant literature (see my comments in the annotated PDF). For example, lines 61-62 the authors write 'the exhumation history of the CBU between ∼52 and ∼25 Ma remains enigmatic and poorly constrained'. This is not true and they can't write that without exploring all (and citing some of) the works that have studied blueschist-facies rocks in the Cyclades and mainland Greece (I estimate it to be around 30-40 studies or even more). It is very important to note that these rocks are some of the most studied HP-LT rocks worldwide and the authors can't say that to justify the 'novelty' of their study. Additionnally, the data are sometimes over-interpreted. For example, there are no evidence in this study that garnets crystallized at peak P-T conditions while the authors mainly use their garnet P estimations to say that maximum P-T conditions reached by the CBU is ∼1.7 GPa / 500-550 ËŽC. In the contrary, many studies (e.g. Groppo et al., 2009; Dragovic et al., 2012; Ashley et al., 2014; Laurent et al., 2018) suggest that most garnets in the CBU crystallized before peak P-T conditions, during the late prograde evolution of the CBU, at pressures very close to the ones that have been measured in this study (∼1.7 GPa - I remind that the T of garnet crystallisation has not been constrained in this study). Finally, there are some issues with the figures. For example, Fig. 1 is supposed to be a geological map of Syros but the tectonic structures (e.g. faults and shear zones) are not represented. As it is, this map is more a lithological map of Syros and I have noted many inconsistencies for the lithology in some area (see the annotated PDF). Moreover, in Figures 2b, 5a , 5b, 5c and 5d the scale is clearly not correct. In my opinion the work has to be profoundly reconsidered before publication. I do have several suggestions written in the annotated PDF (114 comments) that will hopefully serve to further strengthen the paper.

Please also note the supplement to this comment:
https://se.copernicus.org/preprints/se-2020-154/se-2020-154-RC2-supplement.pdf

---

## Author Comment (AC2) · 17 Dec 2020

*We thank the reviewer for the positive review of this manuscript. We address the reviewers comment below, wherein the reply to the comment is italicized and changes made to the main text are in red.*

My only very minor criticism is the reference to "metasedimentary" rocks. There should be a description of the kind of sediments they are interpreted to be. Too commonly, this term gets interpreted as shales (pelitic schists). There are certainly sedimentary rocks on Syros: (1) carbonates (marbles) and (2) very minor quartzites. In a few places, there are rocks that could have shale-like protoliths, but, most of the other rocks are either magmatic, volcanic or volcano-clastic that span a range of basic to felsic compositions.

*We agree with the reviewer that metamorphosed shales (pelitic schists) are uncommon on Syros. In this paper, the metasediments we refer to from Delfini are quartz-rich lenses that are intermixed with metavolcanic rocks (e.g., Figure 2). It's difficult to give an exact protolith for these rocks given the strong greenschist facies overprint, but they likely represent graywackes or sandstones variably intermixed with mafic tuffitic intercalations (e.g., Keiter et al., 2011). A short description has been added to describe the sediments (lines 110 – 111 and lines 111-112):*

*"Metasedimentary rocks (quartz-rich lenses intermixed with metavolcanics) at Delfini Beach show retrogression from eclogite- and blueschist- to greenschist facies (Fig. 1). "*

*"Protoliths of Delfini rocks remain enigmatic, but may be graywackes or sandstones variably intermixed with mafic tuffitic intercalations."*

References

Keiter, M., Ballhaus, C. and Tomaschek, F.: A new geological map of the Island of Syros (Aegean Sea, Greece): implications for lithostratigraphy and structural history of the Cycladic Blueschist Unit, Geological Society of America., 2011.

---

## Author Comment (AC3) · 17 Dec 2020

*We thank the reviewer for the thorough evaluation that helped improve this manuscript. Replies to comments are italicized and changes made to the main text are in red.*

***General Comments***

*The reviewer has 3 general comments related to this manuscript:*

1) "However, there are important deficiencies in their bibliographic review that need to be addressed by considering and citing relevant literature."
2) "Fig. 1 is supposed to be a geological map of Syros but the tectonic structures (e.g. faults and shear zones) are not represented. As it is, this map is more a lithological map of Syros and I have noted many inconsistencies for the lithology in some area (see the annotated PDF)."
3) "For example, there are no evidence in this study that garnets crystallized at peak P-T conditions while the authors mainly use their garnet P estimations to say that maximum P-T conditions reached by the CBU is ~1.7 GPa / 500-550 °C. In the contrary, many studies (e.g. Groppo et al., 2009; Dragovic et al., 2012; Ashley et al., 2014; Laurent et al., 2018) suggest that most garnets in the CBU crystallized before peak P-T conditions, during the late prograde evolution of the CBU, at pressures very close to the ones that have been measured in this study (~1.7 GPa - I remind that the T of garnet crystallisation has not been constrained in this study)."

1) *Citations and additional text have been added where appropriate (see comments and text).*
2) *The Figure 1 map has been updated to include additional faults and shear zones. For a detailed structural evolution of Syros that discusses localized shear zones we refer to the preprint of Kotowski et al., (in review).*

*We spend additional time addressing comment 3, i.e., conditions recorded by quartz-in-garnet barometry (peak or not peak). Below we summarize evidence that suggests quartz-in-garnet barometry on rocks from the CBU on Syros records near-to or peak P conditions.*

1) ***$P_{trap}$ transects across garnets:*** *We found no evidence for systematic changes in $P_{inc}$ (and thus $P_{trap}$) across garnet chemical zones. We add an additional figure that shows the location of measured quartz inclusions relative to chemical zonations seen in garnet (Fig. 5). This figure highlights that some of our measurements include quartz inclusions near garnet rims. According to some thermodynamic modeling, garnet rims in the CBU on Syros grew at higher pressures than garnet cores; however, our pressure data do not show evidence for this. Based on inclusion measurements across garnet zones, we believe we have captured the full extent of P conditions recorded by the garnets (this study).*
2) ***Retrograde pressure constraints:*** *Pressures from quartz-in-epidote barometry (retrograde epidote) do not exceed those recorded by quartz-in-garnet barometry, supporting the interpretation that maximum pressures reached by the CBU did not exceed those recorded by the quartz-in-garnet barometer (wherein retrograde epidote growth at pressures higher than garnet growth would be expected if garnets reached higher pressures than our data suggests).*

3) ***Constraints from Sifnos:*** *Contrary to the reviewer's comments (that most garnets grew below peak P), Ashley et al., (2014), Dragovic et al., (2012), (2015), and Groppo et al. (2009), found that a majority of garnet growth occurred near to or at peak P conditions, and suggest that garnets grow during nearly isobaric heating. Please refer to Fig. 5 of Ashley et al. (2014), the abstract and Fig. 6D of Dragovic et al. (2015), and the discussion in section 8 and Figure 11 of Groppo et al. (2009). Dragovic et al. (2012), (2015) used garnet isopleth intersections to constrain that a majority of garnet growth in a blueschist and quartzofeldspathic gneiss from the CBU on Sifnos occurs rapidly (< 1 My) at near-to or peak P conditions (e.g., Fig 6b, Dragovic et al., 2015); however, they calculate higher P conditions compared to our study [P ≈ 2.0 – 2.2 GPa (core-to-rim) at T ≈ 460 – 560 °C]. Groppo et al. (2009) use thermodynamic modeling and also constrain garnet growth conditions of ~2.1 GPa between T ≈ 450 – 565 °C (P is constant). Ashley et al. (2014) carried-out quartz-in-garnet barometry on samples collected near those from Dragovic et al. (2012) and Groppo et al. (2009). Ashley et al. (2014) found no systematic* $P_{trap}$ *difference across the garnets [see Ashley et al., 2014 (Fig. 5), or Supplementary Table S7], and constrain lower pressures (~1.6 GPa, pressures updated in this study with revised fits to quartz molar volume data). All studies suggest that garnets record peak P conditions, and that garnet growth occurred under nearly isobaric conditions at max P (unrelated to max T). Absolute pressures vary between studies; our results are most consistent with those of Ashley et al. (2014) (also based on elastic thermobarometry)*

4) ***Previous garnet pressure constraints:*** *All previous studies that have constrained maximum pressures for the CBU on Syros and Sifnos (e.g. Ashley et al., 2014; Dragovic et al., 2012, 2015; Groppo et al., 2009; Laurent et al., 2018; Trotet et al., 2001a) have used garnets to constrain peak pressures.*

   *Below are direct quotes from references discussed above that constrain garnet growth conditions:*

   **Dragovic et al. 2012 (section 9): "**These isopleths converge on an average P–T of garnet core growth of ~2.0 GPa and ~460 °C, which falls just within the assemblage chlorite–chloritoid–glaucophane–garnet–phengite–lawsonite–quartz–rutile. This is consistent with garnet inclusions of all these minerals except chlorite which is predicted to be in very low abundance. Fig. 7D shows a diagram for the garnet rim chemistry on the matrix pseudosection. These isopleths converge on an average P–T of garnet rim growth of ~2.2 GPa and ~560 C, which falls in the assemblage omphacite–glaucophane–garnet–phengite–paragonite–lawsonite–quartz–rutile."
   **Dragovic et al. 2012 (section 9): "**The P–T constraints on the span of garnet growth are also consistent with the P–T envelope determined for Sifnos blueschists by Groppo et al. (2009) shown for reference in Fig. 8 which summarizes the P–T constraints. ***Note that garnet growth spanned less than 1 Ma during near isobaric heating from 460 to 560 °C. This equates to a minimum heating rate of 100 °C/Ma."***
   **Dragovic et al. 2015 (abstract):** "Our data reveal three distinct phases of garnet growth: initial growth at 53.4±2.6Ma (~0.8 GPa and ~300 °C), followed by a period of very limited growth until a second phase, at 47.22±0.36Ma, ***and then a major pulse of growth, responsible for the majority of the final garnet volume, at 44.96±0.53Ma (2.06–2.19 GPa and 490–550 °C)***. This suggests a >2 order of magnitude acceleration in volumetric growth

rate from crystal core to rim, with the final growth pulse occurring rapidly (<0.8 My), during a period of nearly isobaric heating at >75 °C/My."

*Below are the P-T results from Ashley et al., 2014, Dragovic et al., 2012,2015, and Groppo et al., 2009.*

***Dragovic et al. (2012)*** ***Ashley et al. (2014)***

[Figure]

[Figure]

*Given our observations and the preceding results, we believe garnets that record the highest pressure fabrics, are best suited for extracting peak P conditions, and that we have suitably analyzed sufficient quartz inclusions in different garnet zones to say that our samples do not record pressure conditions that exceeded ~1.8 GPa (peak P). Nonetheless, as discussed in the original manuscript, the possibility exists that we have not sampled the exact localities within the CBU on Syros that record max P conditions, or that some rocks from the CBU reached greater depths. However, we consider it more likely that different techniques are recording different pressures.*

*Lines 360 – 361: "It is possible, however, that we did not sample the same rocks as Laurent et al. (2018), or that we have not found or analyzed garnets that record high pressures."*

*We do not agree that max temperatures and max pressures are necessarily correlated, and thus constraining max T is needed to constrain peak conditions of the CBU. Peak temperatures may reflect the maximum depth reached by the CBU during subduction, or can reflect the maximum temperature reached after detachment of rocks from the down-going plate, and accretion to the warm upper plate. Wherein, this may occur at a constant depth (pressure), but an increasing temperature due to thermal equilibration with the warm upper plate or shear heating during (or*

*post) accretion with the upper plate (e.g., Gerya et al., 2002; Ruh et al., 2015). Previous studies (e.g., Ashley et al., 2014; Dragovic et al., 2012, 2015; Groppo et al., 2009) have noted that pressures change minimally during garnet growth, but temperature increases by ~90 °C.*

**Replies to Comments**

**Page 1**

Line 1: elastic thermobarometry or barometry? Qtz-in-garnet and Qtz-in-epidote is only a barometer.

*Quartz inclusions in garnet and epidote do serve as barometers; however, elastic thermobarometry refers to the general applicability of the technique. Some inclusion-host pairs can be thermometers (e.g., Cisneros and Befus, 2020; Zhong et al., 2019).*

Line 10: An abstract usually starts with a sentence providing a basic introduction to the field followed by 2-3 sentences of more detailed background. Then a sentence clearly stating the general problem being addressed in the study. All of that is missing here and I think this would enhance the quality of the abstract.

*We have added 2 introductory sentences to the abstract (lines 10-13):*

*"Retrograde metamorphic rocks provide key insights into the pressure-temperature (P-T) evolution of exhumed material, and resultant P-T constraints have direct implications for the mechanical and thermal conditions of subduction interfaces. However, constraining P-T conditions of retrograde metamorphic rocks has historically been challenging and has resulted in debate about the conditions experienced by these rocks. In this work, we…"*

Line 29: A bit weird to only cite articles written by the same author (i.e. Spear) for such a general statement.

*Additional references that highlight the difficulties with constraining P-T conditions of retrograde metamorphic reactions have been added (lines 30-33):*

*"Historically, one of the most challenging aspects of thermobarometry has been deciphering the P-T evolution of rocks during their exhumation from peak depths back to the surface (e.g., Essene, 1989; Kohn and Spear, 2000; Pattison et al., 2003; Schliestedt and Matthews, 1987; Spear and Pattison, 2017; Spear and Selverstone, 1983)."*

**Page 2**

Line 46: What is enigmatic exactly? Be more specific here.
We know that the CBU was buried at eclogite facies P-T conditions (20 ± 2 kbar / 550 ˚C; Trotet et al., 2001; Groppo et al., 2009; Ashley et al., 2014; Dragovic et al., 2012, 2015; Laurent et al., 2018; Brooks et al., 2019; Skelton et al., 2019 or 15kbar - 500˚C Schumacher et al., 2008; Behr et al., 2018) before to be exhumed. Prograde deformation is characterised by top-to-the S sense of

shear (Philippon et al., 2012) while exhumation is characterised by top-to-the E/ENE sense of shear. Early exhumation was mainly accomodated below the Vari Detachment which represents the Eocene roof of the subduction channel (Laurent et al., 2016; Roche et al., 2016). While the detailed shape of the retrograde P-T path differs from one study to another, there is a good consensus to say that early exhumation was accompanied by cooling directly after peak metamorphism. We also know with high-precision the timing of peak metamorphism that has been constrained using 5 different dating techniques (U/Pb dating on zircon, Sm/Nd and Lu/Hf on garnet + Rb/Sr and 40Ar/39Ar dating of white mica) at 52-49 Ma (Tomaschek et al., 2003; Putlitz et al., 2005; Lagos et al., 2007; Bröcker et al., 2013; Dragovic et al., 2015; Cliff et al., 2016; Lister and Forster, 2016; Laurent et al., 2017; Uunk et al., 2018). Finally we also have a good idea of the timing of exhumation (exhumation within the blueschist-facies 49-37 Ma; transition from BS to GS facies at 35-30 Ma and final ductile exhumation in the GS facies at 20-18 Ma.

*We agree with several points highlighted by the reviewer here, that suggest the early exhumation history remains enigmatic. Specifically, (for the CBU on Syros), the max and retrograde P-T conditions vary, the relationship between distinct deformation events and the timing of distinct recrystallization events has not been well constrained, a data-driven constraint on the bulk strain in all of the CBU during exhumation (coaxial or non-coaxial) is unknown (c.f. Ring et al., 2020), the physical mechanisms that drove exhumation, and the degree of structural repetition of the CBU (and when this occurs) remains enigmatic. We don't discuss all of these aspects in this manuscript so they are not highlighted. We have added an additional sentence to describe this (lines 69 – 75):*

*"Previous work has constrained some aspects of the early exhumation history of the CBU on Syros, including: the timing of peak and retrograde metamorphism (e.g., Bröcker et al., 2013; Cliff et al., 2017; Lagos et al., 2007; Laurent et al., 2017; Skelton et al., 2018; Tomaschek et al., 2003), prograde and exhumation-related kinematics (e.g., Behr et al., 2018; Keiter et al., 2011; Kotowski and Behr, 2019; Laurent et al., 2016; Philippon et al., 2011; Rosenbaum et al., 2002), and the retrograde P-T path (e.g., Laurent et al., 2018; Ring et al., 2020; Schumacher et al., 2008; Skelton et al., 2018; Trotet et al., 2001a, 2001b); however, debate remains about the relationship between deformation events and retrograde metamorphism, the maximum pressure reached by different CBU rock types, the retrograde P-T path, and the mechanisms and kinematics of CBU exhumation."*

*Here, we primarily address the differences in previous P-T constraints (peak and retrograde), and opposing models for early exhumation that make the exhumation history of the CBU on Syros enigmatic (discussed on lines 80-92):*

Line 48: 2014a or 2014b?

*Only one Ashley 2014 reference (Ashley et al. 2014) cited in the manuscript (2014 is not split into a and b references).*

Line 49: Yes I really agree with the use of the word 'near' here.

*Details on what pressures we believe the quartz-in-garnet barometer records are now further discussed in the beginning of this reply.*

Line 51: which criteria have you used to conclude that it is 'more robust'.

*This sub-sentence ("is more robust than what is commonly possible with conventional thermobarometry"), refers to being able to extract quantitative P-T information from outcrop and microstructurally constrained single mineral growth. Something that usually requires 2 or more minerals in equilibrium with conventional thermobarometry.*

Line 51: This retrograde P-T-D path is not clear for me. I would suggest to clearly draw it in your figure 6.

*The retrograde P-T path is now drawn in Figure 7 (previously Figure 6).*

Line 53: There are important deficiencies in your bibliography review that need to be addressed by considering and citing relevant litterature (see all my comments below).

*Further references have now been added where appropriate; refer to replies herein.*

Lines 56-57: Be more precise here. What do you mean exactly by exhumation? Early exhumation? Late exhumation? What happens between 52 and 25 Ma? A lot of studies have worked on that (again see my previous comments above).

*This specific sentence is referring to back-arc extension related exhumation, as stated in the text (lines 62-63):*

*"followed by exhumation during Oligo-Miocene (~25 Ma) back-arc extension (e.g., Jolivet and Brun, 2010; Ring et al., 2010)."*

*Please refer to the sentences that follow for more information on exhumation between ~52-25 Ma.*

Line 56: much more references must be cited here (see my previous comment above).

*We have added additional references that also constrain the time at which the CBU reached peak metamorphic conditions (lines 59 – 62):*

*"CBU rocks on Syros record Eocene subduction (~52 - 49 Ma) to peak blueschist-eclogite facies conditions (Bröcker et al., 2013; Cliff et al., 2017; Lagos et al., 2007; Laurent et al., 2017; Lister and Forster, 2016; Putlitz et al., 2005; Tomaschek et al., 2003; Uunk et al., 2018),"*

Lines 58-59: Do you mean 'in the CBU'?

*We do not mean in the CBU. We refer to the general Cycladic islands, wherein deformation occurs both in the footwall and hanging wall adjacent to large-scale extensional detachments.*

**Page 3**

Lines 61-63: I strongly disagree here. You can't say that without citing all the works that have studied blueschist-facies rocks in the Cyclades (again, see my previous comment above). And it's not only about Syros and Sifnos (where at least 10-12 studies have studied 'what happens between 52 and 25 Ma'). There are other studies that focussed on Tinos, Andros and many other Cycladic Island and which has constrained the early exhumation history of the CBU. I am not saying that there is no debate, but you can't say that 'the exhumation history of the CBU between ~52 and ~25 Ma remains enigmatic and poorly constrained'. It is very important to note that these rocks are some of the most studied HP-LT rocks worldwide.

*We have reworded to additional sentences on the key questions that remain about the early exhumation history of the CBU on Syros (see comment above, lines 69 – 75):*

*"Previous work has constrained some aspects of the early exhumation history of the CBU on Syros, including: the timing of peak and retrograde metamorphism (e.g., Bröcker et al., 2013; Cliff et al., 2017; Lagos et al., 2007; Laurent et al., 2017; Skelton et al., 2018; Tomaschek et al., 2003), prograde and exhumation-related kinematics (e.g., Behr et al., 2018; Keiter et al., 2011; Kotowski and Behr, 2019; Laurent et al., 2016; Philippon et al., 2011; Rosenbaum et al., 2002), and the retrograde P-T path (e.g., Laurent et al., 2018; Ring et al., 2020; Schumacher et al., 2008; Skelton et al., 2018; Trotet et al., 2001a, 2001b); however, debate remains about the relationship between deformation events and retrograde metamorphism, the maximum pressure reached by different CBU rock types, the retrograde P-T path, and the mechanisms and kinematics of CBU exhumation."*

Line 64: , we focus

*", we" has been added (line 76).*

*"In this work, we focus on rocks within the CBU,"*

Line 65: Can you precise what you mean by conventional thermobarometry exactly?

*We refer to conventional thermobarometry as techniques that primarily rely on chemical equilibrium to constrain P-T conditions. This has been added on lines 79-82:*

*"Some conventional thermobarometry (i.e., thermobarometry techniques that rely on chemical equilibrium) suggests that the CBU on Syros reached peak P-T conditions of ~1.5 GPa and ~500 °C (Ridley, 1984). Trotet et al. (2001a) and Laurent et al. (2018) suggest higher peak P-T conditions of ~2.0 - 2.4 GPa and ~500 - 550 °C;"*

Line 66: It is quite important to note that Ridley, Trotet and us have all used different thermobarometric techniques.

*This is true and important. The different techniques that were used are discussed in section 6.2 (lines 314 – 376).*

*Lines 339 – 341: "Different methodologies applied to CBU rocks from Syros have resulted in a wide range of maximum P estimates. Schumacher et al. (2008) used mineral-equilibria modeling of glaucophane-bearing marbles to place constraints on maximum P-T conditions."*

*Lines 345-349: "Trotet et al. (2001b, 2001a), Laurent et al. (2018), and Skelton et al. (2018) employed thermodynamic phase-equilibria modeling and supplementary methods to constrain P-T conditions for CBU rocks from Syros. Skelton et al. (2018) used the Powell and Holland (1994) Thermocalc database, Trotet et al. (2001b, 2001a) used the Berman (1991) thermodynamic database and the TWEEQC approach, and Laurent et al. (2018) used empirical thermobarometry, GrtMod (Lanari et al., 2017), and isochemical phase diagrams."*

Line 67: So this is not conventional thermobarometry?

*This is "conventional" thermobarometry. We hope this has been clarified with the sentence above.*

Lines 66-67: Why original?

*Ridley (1984) was the first published study to provide quantitative P-T constraints for the CBU on Syros, thus we refer to his work as the "original" work in this sentence. Granted, earlier unpublished work [Ridley (1984) results based on Dixon (1969) thesis and Dixon (1976) abstract] had also constrained minimum P conditions for the CBU on Syros. Ridley (1984) used the upper paragonite stability to estimate CBU pressures do not exceed ~ 2.0 GPa at ~ 500 °C.*

Line 68: I know where Kini Beach is but 99% of the reader will probably don't know. Please refer to a figure here to locate Kini Beach.

*Kini beach is now labelled on Figure 1.*

Line 70: I think here you refer to our JMG paper in 2018. Please note that our P-T path also show cooling during decompression and so it is not a 'different model' but an additional finding that happens at the transition from blueschist to greenschist facies.

*This is a good point worth clarifying. The Laurent et al. (2018) P-T path does initially show cooling during decompression, but nonetheless differs from other P-T paths in that it shows heating at ~ 1 GPa, and only for a specific sub-section of the CBU (Kampos subunit keeps cooling during decompression). This is important, as this could relate to the depth at which the CBU preserved on Syros transitioned from a cooler forearc to a back-arc, and suggests that different sections of the CBU experienced different P-T histories during exhumation. In this work, stable isotope thermometry from several greenschist facies outcrops across Syros does not support the high temperatures of ~ 550 °C. Instead, stable isotope data suggests temperatures of ~ 400 °C at ~ 1 GPa. This makes this a different model, rather than a model that supports re-heating at the base of the CBU on southern Syros (Laurent et al., 2018). The implications of this are further discussed in a pre-print available on EOS [Kotowski et al., (in review)]. Briefly, we relate greenschist facies metamorphism that occurs at ~35 Ma to a discrete deformation event, wherein strain was primarily localized within sediments and metavolcanics (non-metabasites). This strain localization seems to have propagated towards southern Syros (bottom of the structural pile), in response to*

*thrusting, underplating, and exhumation-related deformation of CBU slices. We propose that this occurred in the subduction channel (not back-arc related), and discuss the implications of the deformation, timing, and P-T constraints on how the CBU subduction complex was configured between ~52 – 35 Ma. We have clarified this sentence to reflect the reviewers comment (lines 83-86).*

*"Published exhumation P-T paths for the CBU on Syros are also highly variable, ranging from cooling during decompression, near-isothermal decompression, to cooling during decompression followed by reheating at moderate pressures (Laurent et al., 2018; Schumacher et al., 2008; Skelton et al., 2018; Trotet et al., 2001a)."*

Lines 71-72: Again you can't say that. Even if you only want to focus on Syros there are some studies that have combined structural geology, petrology and thermobarometry. And if you look at the scale of the CBU there are many.

*There are indeed many studies on the CBU that combine petrology and thermobarometry. But very few combine outcrop and thin section context to interpret those P-T constraints, and can relate the P-T constraints to specifc deformation events. We believe this is key for accurate interpretations of the CBU's tectonic history. We hope the new sentences above helps clarify this for readers.*

Lines 73-74: A very large majority of the deformation observed on Syros is asymetric with shear sense indicators observed everywhere.

*We are citing the work of previous studies here; however, we don't agree with this statement. Laurent et al. (2016) show images that suggest top-to-the-east shear sense indicators are prevalent in the CBU on Syros (exhumation-related). There are indeed shear sense indicators across Syros, but very little evidence for non-coaxial deformation under retrograde blueschist-to-greenschist facies conditions prior to core-complex related exhumation. Our observations support that much of the deformation in the CBU on Syros appears to be dominantly coaxial, characterized by shear sense indicators oriented in opposing orientations. This has been discussed previously by Bond et al. (2007) and Rosenbaum et al. (2002). Bond et al. (2007) state in their abstract that they believe that "the role of major detachment faults [in accommodating extension] may have been over-emphasized in previous studies". Our own observations suggest that statistically significant evidence for a clear asymmetric shear sense (non-coaxial deformation) is only locally preserved in the CBU at the base of the Vari detachment [e.g., at Fabrikas, see Ring et al. (2020) for reference].*

Lines 74-75: Ok for Lister and Forster but why are you saying that Trotet et al. (2001a, 2001b) involve a 'complex thrusting and extension model'? And please, even for Lister and Forster (2016) explain in a bit more detail what you mean by complex 'complex thrusting and extension model'

*Both models call for accretion at the base of a wedge and exhumation accommodated by ductile extensional shear zones; however, the Lister and Forster (2016) model has added complications (multiple mode switches). We've expanded this section to better describe the authors models (lines 87-91):*

*"Because of these conflicting P-T paths, several models have been proposed to explain the exhumation history of the CBU, including coaxial vertical thinning (Rosenbaum et al., 2002), extrusion wedge tectonics (Keiter et al., 2011; Ring et al., 2020), multiple cycles of thrusting and extension (Lister and Forster, 2016), continuous accretion and syn-orogenic extension (Trotet et al., 2001a, 2001b), and subduction channel exhumation (Laurent et al., 2016)."*

Line 74: The extrusion wedge models of Keiter and Ring are completely different. Keiter argued that rocks of the CBU were not deformed after peak of metamorphism (at least no major phase of deformation). We all know that this is completely wrong as retrograde deformation is observed everywhere on Syros. The model of Ring is completely different and I don't understand why you oppose that to the Laurent et al. (2016) model as Ring model involves also exhumation in the subduction channel. In the same way, we also explained in the 2016 paper (see also Roche et al., 2016) that at large-scale the exhumation of the CBU is mainly accomodated along a top-to-the East detachment at the top (Vari Detachment) and a top-to-the South thrust at the base (observed on Ios, Huet et al., 2009; Laurent et al., 2017).

*All three of these studies suggest subduction channel exhumation. Keiter et al. (2011) and Ring et al. (2020) argue for extrusion models ("extrusion" of a wedge CBU rocks within a subduction channel), i.e., kinematically necessary thrust fault at the base (the subducting slab) and a kinematically necessary normal fault at the top (upper plate). We prefer not to discuss the differences in strain accommodation during exhumation in both of those studies here. Laurent et al. (2016) propose a similar geometry (as stated above) but do not specifically state an extrusion wedge in the manuscript; therefore, we did not group this reference with the others. Theoretically, subduction channel and extrusion models do mean slightly different things, the extrusion wedge model calls for a specific geometry that should produce opposing shear sense indicators at distinct locations that define the base (subduction plate) and top (upper plate) of the wedge (within a subduction channel). A subduction channel model has a looser definition (without a specific geometric structure) that merely reflects the plate interface structure (discrete or broad interface), and does not require this deformation. Therefore, here we split the references accordingly. If the reviewer is suggesting that Laurent et al. (2016) do propose an extrusion wedge, then we can group their manuscript under "extrusion wedge" with Keiter et al. (2011) and Ring et al. (2020).*

Line 78: How do you know that each locality has the same P-T progression?

*Please refer to the results and discussion for this information.*

Line 85: under / after. It's a detail but it is better to say syn- or post- (e.g. syn-BS facies folding).

*The text has been changed (lines 101 – 103):*

*", and NE-SW-oriented stretching lineations primarily defined by white mica, glaucophane, and epidote; this indicates syn-blueschist facies folding ($D_{t1}$)."*

Line 86: Important to say a bit more about this sample.

*Thin section images now individually uploaded in the supplementary material), but we are unsure about what further information the reviewer would like to see here.*

Line 86: On the picture I can see garnets that seem far smaller than 1 mm).

*We are unsure about this comment. There are no garnets that appear far smaller than 1 mm (section image below and higher resolution in supplementary material; black scale bar is 5 mm).*

[Figure]

Line 86: Can you at least write the sample name on this figure please?

*The sample name is now given in the individual file names.*

Lines 87-88: I agree with you but it's weird that on your picture and on the garnet that you have labelled, we have the impression to see glaucophane included within the garnet. I understand that this can be due to 3D fracturing of the garnet but it is perhaps not the best example to show that Gln is not included within Grt.

*The reviewer is correct; the labelled garnet does have glaucophane appearing to be in garnet. We agree that these are likely not glaucophane inclusions, wherein the fractures seen in garnet directly beneath glaucophane, are not propagating through the glaucophane. Please refer to the remaining garnets if this specific garnet illustration is not clear.*

Lines 89: I agree

*Ok.*

Line 82: Can you please show that? So finally, it means that this sample is an eclogite retromorphosed in the BS-facies. This is important to mention.

*This is the opening sentence of this section/sample description (lines 99-100):*

*"Mafic rocks from Kalamisia preserve retrograde blueschist facies metamorphism (Fig. 1). Protoliths of Kalamisia rocks are fine-grained basalts. They exhibit an early foliation ($S_s$) characterized by relict blueschist and eclogite facies minerals."*

Line 89: Interesting. The reader will prefer to see that on a figure.

*A higher resolution thin section scan (supplementary figure) has now been added to allow readers to see if they agree with our thin section descriptions.*

Line 90: Ok I agree. I can see from your table S1 that you have also observed albite in this sample. So it is strongly probable that this retrograde parageneses crystalized in the AEBS facies (Evans, 1990).

*The reviewer is correct; this is an albite-bearing blueschist. We have not used our sample mineralogy/descriptions to give exact sub-facies fields for where these rocks fall, and instead refer to the quantitative P-T results presented in this work. As shown in Trotet et al., 2011, the AEBS facies field covers a large P-T range (~0.8 – 1.5 GPa and 400 – 600 °C, stability field III in Fig. 7a), and the stability field is strongly dependent on the chosen phase activities.*

**Page 4**

Line 96: Can you please orientate your pictures in Fig. 2?

*Orientations have been added to the field descriptions.*

Line 97-98: Boudinage is usually associated to extension not folding. Please explain.

*Not clear what the reviewer means here. Folding and boudinage very commonly go hand-in-hand, but occur in different orientations with respect to the principal stress field and corresponding finite strain. E.g., shortening (folding) commonly occurs in one orientation, and is kinematically linked with elongation-extension (e.g., boudinage) along an opposing axis at ~90°. We relate epidote boudinage and extension to folding during $D_{t2}$ (E-W oriented on Syros), wherein a $\sigma_1$ compressive stress causes folding, and extension occurs parallel to $F_{t2}$-fold hinge lines. The same folding event ($D_{t2}$) also causes re-orientation of lineations towards the E-W (image from Jean-Pierre Burg structural geology notes).*

[Figure]

Line 101: is St and St2 the same thing? If so be consistent throughout the text.

*Good catch, this was a typo. Fixed (lines 119 – 120):*

*"a new generation of fine-grained epidote (also interpreted as ep2) grows within a newly developed crenulation cleavage ($S_{t2}$, Fig. 2c,d,e)."*

Lines 102-103: Is it possible to upload high-resolution images as I can't see anything when zooming on this picture.

*Definitely. Figure 2 has now been uploaded as an individual figure at higher resolution.*

Line 106: I can't see that on this figure as the resolution is not good enough.

*We agree, the decrease in image resolution when uploading is a problem. Instead, we've uploaded each thin section image as an individual supplementary material figure to maintain a higher resolution.*

**Page 5**

Line 130: Are you analysing the same lithology each time? You must better described the samples you are looking at each time.

*We're unsure about what needs to be clarified here. We have added a general statement about the number of sections looked at for each outcrop and general protolith descriptions for each section (lines 96-97, below). The sample names are given under each locality description. They exhibit the same microstructures, aside from sample KCS34 from Delfini (records folding and described in more detail), and so are described in bulk to avoid redundancy. Further information for each individual sample can be found in Supplementary Table S1. There are two key things that we need to describe to adequately communicate the context of garnets (Group 1), epidote1 (Group 2), and epidote2 (Group 3): the outcrop structure, and the microstructures. If the context of these is not described well enough to interpret the P-T results, then we can change the sample description.*

*"1 – 4 samples from each locality were examined petrographically."*

Line 134: The Methods section should appear in the text after the Geological Setting and before the Field and microstructural observations section.

*We consider the original organization, with sample descriptions and microstructures provided before describing the methods used to analyze microstructural features to be a more logical way to present our results.*

Line 143: Quantify

*This is quantified in the same sentence (lines 162 – 164):*

*"Measured quartz inclusions were small in diameter relative to the host, and were two-to-three-times the inclusion radial distance from other inclusions, fractures, and the host exterior to avoid overpressures or stress relaxation (Fig. 4a,b; Campomenosi et al., 2018; Zhong et al., 2020)."*

*In other words, what the above studies have shown (e.g., Campomenosi et al., 2018; Zhong et al., 2020) is that what matters is not so much the size of the garnet (or epidote), but that the garnet is big relative to the inclusion; i.e., that the quartz inclusion is two-to-three times the radial distance (of quartz, from the center of the inclusion) away from the exterior of the garnet (or fractures, other inclusions, etc).*

**Page 6**

Lin 184: From my experience, epidotes and garnets are most of the time zoned in the CBU.

*This reviewer is correct, they are commonly zoned; however, the compositional dependence of the quartz-in-garnet barometer is negligible because the thermal expansivities and bulk moduli of different garnet end-members are nearly identical (see example of compositional dependence below). Thermodynamic properties between epidote and clinozoisite vary more (almost identical thermal expansivity, but differing bulk moduli); however, this compositional dependence is also not significant for the quartz-in-epidote barometer below 1.5 GPa (see Fig. 1E, Cisneros et al., 2020). Furthermore, the assumed composition of epidote used for modeling closely approximates the composition of epidotes in KCS34 ($X_{ep}$ = 0.51; see Supplementary Table S6). This is described on lines 202-204:*

*"Garnet compositions have a negligible effect on entrapment pressures ($P_{trap}$) because the thermodynamic and physical properties of garnet end-members are similar (e.g., Supplementary Table S8). Epidote composition has a greater effect on $P_{trap}$, but the compositional dependence is minor < 1.5 GPa (Cisneros et al., 2020)."*

*Garnet compositional dependence* *Epidote compositional dependence*

[Figure]

[Figure]

**Page 7**

Line 208: You really need to locate all your sample on your Fig. 1 because here I am completely lost; where were these samples collected?

*Sample names have been added to Figure 1. For exact sample locations (GPS coordinates) and locations please refer to Supplementary Table S1.*

**Page 8**

Lines 222-223: If I'm correct calcite and quartz were always observed in boudins, showing that they are in chemical equilibrium as they crystallized simultaneously.

*This is indeed evidence that supports chemical equilibrium between quartz and calcite; however, the coexistence of quartz-calcite pairs in boudins is not in itself evidence for chemical equilibrium. E.g., the quartz can precipitate from a fluid first, followed by calcite precipitation after, with a lag-time (and conditions) between the precipitation of both minerals being unknown (or multiple mineral growth generations in the boudin neck). For these samples, we rely on multiple observations that suggest the quartz-calcite pairs record chemical equilibrium conditions (e.g., similar temperatures from different boudin-neck quartz-calcite pairs, highly reproducible stable isotope analyses, lack of re-fracturing of boudin-neck minerals, etc.). The best we can do after that is assume chemical equilibrium is valid, as with most other chemically-dependent techniques.*

Lines 231 and 232: ,

*The commas have been added (lines 251 – 252).*

*"Epidote and pyroxene were analyzed for Si, Al, Na, Mg, Ca, Cr, K, Ti, Fe, and Mn on…"*

Line 239: How do you know that garnet crystallize near peak metamorphic conditions!? You have no arguments for that! It's only the highest pressure conditions that you have measured. In our

JMG paper in 2018, we found that most garnets crystallised during the prograde path, before the CBU reached peak metamorphic conditions.

*Please refer to our opening discussion for this comment. As an additional note, Figure 17 in Laurent et al. (2016), which summarizes their P-T results, demonstrates that the P-T conditions of the majority of garnet growth overlap with the few estimates of higher peak P-T within reported error. The uncertainties in pressure estimates for the 3 samples that yield higher P are > 0.5 GPa.*

Line 239: you can say: 'during retrogression in the AEBS-facies'.

*We don't think there is a need for assigning the mineralogy here to a metamorphic sub-facies. We rely on the quantitative P-T constraints to constrain pressures. And as noted in the comment below, we still do not see biotite in this sample. The paragenetic assemblage of Trotet et al. (2001) also predicts biotite for AEBS-facies (some rock compositions), but to the best of our knowledge, no study has found biotite in the CBU on Syros.*

Lines 245-246: 500-550 ˚C is definitely the maximum temperature reached by rocks of the CBU. But how can you tell that garnets crystallized at Tmax??? Again, in our JMG paper we show that garnet DON'T crystallized at Tmax. So if we look to your pressure data obtained from quartz inclusion in garnets, they are really close to what we found for a large majority of the garnets we analysed (17 ± 2 kbar and 450 ± 50°C), which is to be honest reassuring. Note that on Sifnos, prograde garnet growth event has been retrieved at ~20 kbar and 450–500°C (Dragovic et al., 2012; Groppo et al., 2009).

*Please refer to our opening discussion for this comment. The temperatures from Dragovic et al., 2012, 2015 and Groppo et al., 2009 (~450–500°C) that the reviewer quotes in this comment are not correct. For example, Dragovic et al (2012) state in in the abstract: "Garnet core and rim chemistry indicates that growth began at 2.0 GPa and 460 °C and ended at 2.2 GPa and 560 °C."*

*Groppo et al. (2009) state in the abstract: "The resulting P–T conditions could be fitted by a simple trajectory consisting of a smooth clockwise P–T loop, with two distinct high-pressure events at T=450–500 °C, P > 2.0 GPa (assemblage A) and T=525–565 °C, P > 2.1 GPa (assemblage B),"*

**Page 9**

Line 256: unit is missing here (even if it's quite obvious) :-)

*Good catch. The unit has been added (line 282):*

*"MPa, corresponding to a $P_{trap}$ of 1.43 ± 0.12 GPa (n = 6)…"*

Line 256: For me the epidotes data are the most interesting of this study as they allow to add P-T constraints on the exhumation path, while for garnet you only have constrained a pressure of crystallisation.

*All of the pressure constraints presented here (quartz-in-epidote and quartz-in-garnet barometers) constrain pressures of mineral crystallization (epidote and garnet growth). The challenge is attributing mineral growth to a definitive stage of deformation (and metamorphism). This is why we discuss the kinematic context of porphyroblasts (both garnet and epidote), and identify our garnet hosts as pre-to-syn kinematic with respect to prograde-to-peak fabric development. Unless this is referring to the additional temperature constraints we have from stable isotope thermometry and our lack of T constraints for garnet growth? If so, please see the opening discussion.*

Line 259: You should specified on Fig. 5 the Group 1, 2 and 3 as they can be confused with the Epidote 1 and Epidote 2.

*This is a good point for clarity. The appropriate Groups are stated in the figure caption (lines 733-735), and they have been added to Figure 5.*

*"Figure 5. Comparison of $P_{inc}$ determined from different quartz bands using hydrostatic calibrations, and by using phonon-mode Grüneisen tensors (strains). Red, blue, and yellow symbols indicate qtz-in-grt (Group 1), qtz-in-ep1 (Group 2), and qtz-in-ep2 (Group 3) results, respectively."*

Line 260: Again this is very consistent with the growth of garnets during the late prograde P-T path but we showed that this garnet growth event happened at ~470 ˚C). You are not constraining the maximum pressure reached by the CBU here or at least you have no evidence for that.

*Please refer to our opening discussion for this comment.*

Line 265: Same remark as before

*Please refer to our opening discussion for this comment.*

Line 273-274: I agree. And this is consistent with a growth of garnet during the late prograde P-T path.

*Please refer to our opening discussion for this comment.*

Lines 274-275: This is not clear for me as the data reported in Table S3 for a single sample are not similar (from 1,67 to 1,32 GPa in KCS70A; from 1,82 to 1,50 in KCS34; and from 1,80 to 1,60 GPa in KCS3 with only 2 analyses). These differences seem to be significant compared to the low uncertainty reported for the mean. Are you sure that considering Group mean is a good strategy as garnets find in a single area (this is possibly also true at the scale of a sample) can have crystallised at different pressures?

*The reviewer is correct, pressure differences between quartz inclusions exist (from the same and different garnets). But we are concerned with seeing systematic pressure differences across garnets or between different garnets, which we do not observe (e.g., see Figure 6). Therefore, we attribute the absolute pressure variations to real scatter (e.g., $P_{inc}$ inclusion stress differences from different size of inclusions, minor inclusion shape effects, quartz/garnet orientation effects, etc).*

*Previous studies have shown that from simple hydrostatic experiments, absolute inclusion pressure differences from garnets grown at one pressure exceed 0.2 GPa (e.g., Bonazzi et al., 2019; Thomas and Spear, 2018). Since we don't see any systematic changes in $P_{trap}$ across garnets, we report the sample mean and the standard deviation around the mean as the uncertainty. The uncertainty of the mean is calculated as the standard deviation (σ) for each sample analyzed. If we reported the standard error, the uncertainty would be lower.*

**Page 10**

Lines 277-278: repetition

*Yes, part of what is stated here is also mentioned in the results section, but we think it is needed here to further emphasize the microstructural context for readers.*

Line 277: I think you mean Table S6. The composition you show in Table S6 is quite different to the 'usual' compo of omphacite I have measured in many sample. The low proportions of Mg and Ca suggest more jadeitic composition I think. This seems to be supported by your ~80% Xjd.

*Good catch. We were referring to Table S6. This has been changed (line 302):*

*"Delfini: KCS1621, Supplementary Table S6)."*

*The reviewer is also correct about the composition of these pyroxenes. They are not "omphacite", but instead jadeite. This has been fixed (lines 301 - 302):*

*"Pyroxene inclusions within different garnet zones (core: $X_{jd} \approx 0.84$, rim: $X_{jd} \approx 0.81$) also show no difference in composition,"*

Lines 279-280: Some repetitions here and anyway these info are observations and should be included in the Results section.

*This sentence also further emphasizes the microstructural context of the minerals for readers.*

Lines 282: Yes I agree and so they grow at a T different than Tmax. And so they don't crystallise at peak metamorphic conditions.

*Please refer to our opening discussion for this comment.*

Line 282: But you just said before that garnet crystallized during the prograde path and so not at peak P-T conditions!

*Please refer to our opening discussion for this comment. This has been changed to read "prograde-to-peak" (lines 308 – 309):*

*"Based on these observations, the Group 1 $P_{trap}$ estimates from the qtz-in-grt barometer record high-P conditions on Syros associated with prograde-to-peak garnet growth,"*

Lines 295-296: How can the results of Ashley et al. 2014 of ~1.8 GPa (as you mention just above) be used as an argument to say that the CBU reached pressure > 2 GPa?

*We are not suggesting the CBU has reached pressures > 2.0 GPa, rather, previous work has used the pressures derived in Ashley et al. (2014) to support that the CBU reached peak P conditions of ~2.0 GPa. Please see comment 1 on page 4 (this document), Roche et al. (2016), and Laurent et al. (2018) for reference on work that uses the Ashley et al. (2014) results to support P's of ~2.1 GPa or greater. Granted, an individual quartz $P_{trap}$ reaches ~2.1 GPa (see Ashley et al., 2014 or Supplementary Table S6 column H). However, when recalculated with updated fits to quartz molar volume data, the max individual quartz $P_{trap}$ is reduced to 1.9 GPa (see Supplementary Table S7).*

Line 299: Perfectly consistent with a late prograde crystallisation at T ~470˚C.

*Please refer to our opening discussion for this comment.*

Line 303: Be careful here in our JMG paper we estimated peak pressure of 22 ± 2 kbar which is not exactly the same as 24 kbar.

*The maximum pressure reached by the CBU was suggested to be between 2.2-2.4 GPa in Laurent et al. (2018) (e.g., Figure 17 from Laurent et al. 2018, or Figure 7 this study).*

Line 308: Same remark! You just need to constrain the T (what we have done) to show that garnet don't crystallise at peak P-T.

*Please refer to our opening discussion for this comment.*

**Page 11**

Lines 316-317: The Schumacher et al. data are perfectly correct but they are just showing that this assemblage forms at ~1.5 GPa / 500 ˚C. But absolutely not that the peak P-T conditions of the CBU is ~1.5 GPa / 500 ˚C (of course for me and not for them).

*Schumacher et al. (2008) provide several lines of evidence for their samples that suggest they constrained peak P-T conditions; this evidence is also extensively discussed on lines 340-344:*

*"Schumacher et al. (2008) used mineral-equilibria modeling of glaucophane-bearing marbles to place constraints on maximum P-T conditions. Maximum P-T conditions are constrained by the presence of glaucophane + $CaCO_3$ + dolomite + quartz, which suggests that the marbles exceeded the albite/Na-pyroxene + dolomite + quartz → glaucophane + $CaCO_3$ reaction, but did not cross the dolomite + quartz → tremolite + $CaCO_3$ or the glaucophane + aragonite-out reactions. The mineral reaction constraints suggest maximum P-T conditions of ~ 1.5 - 1.6 GPa and 500 °C for the CBU marbles."*

Lines 323: 22 ± 2 kbar!

*This has been changed to read 2.2 ± 0.2 GPa (lines 349 – 351):*

*"Trotet et al. (2001b, 2001a), Laurent et al. (2018), and Skelton et al. (2018) found high-P conditions for the CBU (≥ 1.9 GPa), and results from Laurent et al. (2018) suggest some rocks reached conditions as high as 2.2 ± 0.2 GPa."*

Line 325: 1.7 ± 0.2 / 450 ± 50 ˚C! Can you please correctly report the data.

*This has been changed to read 1.7 ± 0.2 GPa and 450 ± 50 °C (lines 351 – 353):*

*"Results from Laurent et al. (2018) suggest most garnet growth occurred at ~1.7 GPa and 450 ± 50 °C; however, some garnet modeling results suggest that garnet rims grew at ~2.4 GPa and 500 - 550 °C, albeit errors are increasingly large for these results (± 0.4 - 0.9 GPa)."*

Line 326: Our peak P-T conditions are better estimated using the Grt/Omp/Ph thermobarometer. We only suggest a possible other garnet growth phase at 22-24 kbar but clearly less well constrained following our data as we only have 1 garnet compo plotting up there and 'errors' are higher. Be careful that the error bars you can see in our JMG paper are not errors but represent the uncertainty in the optimal P–T conditions taken as the spacing between the garnet isopleths (see Lanari et al., 2017). The residual values represent the quality of the isopleths intersection, i.e. the quality between the modelled and observed garnet compositions. Please report that info correctly.

*The reviewer is correct, the errors in Laurent et al., 2018 (Figure 15) show the spacing between garnet isopleths (optimal P-T conditions), that result from uncertainties in chemical analyses. The error is now reported this way in this manuscript (lines 351 - 354):*

*"Results from Laurent et al. (2018) suggest most garnet growth occurred at ~1.7 GPa and 450 ± 50 °C; however, some garnet modeling results suggest that garnet rims grew at ~2.4 GPa and 500 - 550 °C, albeit errors are increasingly large for these results (± 0.4 - 0.9 GPa). These errors reflect the spacing between garnet isopleths (optimal P-T conditions), that result from uncertainties in chemical analyses."*

Lines 327-328: Please look carefully here. The error bars clearly indicate that the garnet where the optimal P-T conditions plot at ~2.4 GPa / 475 ˚C are also stable at ~ 1.8 GPa. I agree with you that these 2 garnets here are more likely to have crystallized at ~1.8 GPa. You really have to understand the error bars as a P-T domain where the specific garnet composition is stable.

*The reviewer is correct, the optimal P-T conditions plot at ~2.4 GPa and 475 ˚C are within uncertainty of the 1.8 GPa garnet pressure constraints. This is now stated (lines 355 - 357):*

*"Some GrtMod results suggest prograde core and rim garnet growth at ~1.8 GPa and 475 °C, and ~2.4 GPa and 475 °C, respectively (sample SY1418 from; Laurent et al., 2018); however, the optimal P-T conditions for garnet rims have large errors and plot within uncertainty of garnet core conditions."*

Lines 328-329: Not at all (see my comments above). Please remove.

*This comment has been removed, but as suggested above, the optimal conditions plot at ~2.4 GPa and 475 ˚C (Laurent et al., 2018, figure 17). The uncertainty in the stable P-T domain is still an error.*

Line 331: Again it's not really 2.4 GPa but more something at 22 ± 2 kbar (look to the error bars that you can't ignore here).

*This is not what the data shows for this sample (SY1401) in Laurent et al., 2018 (2.2 ± 0.2 GPa, Figure 15). We understand that the best estimate given the results for all samples is garnet rim growth at 2.2 ± 0.2 GPa, but the optimal P conditions for this sample are ~2.4 GPa. The errors for this sample seem to be better estimated by 2.4 ± 0.4 GPa. This can be changed if the reviewer does not think 2.4 GPa is the representative optimal P condition for GarnetMod rims results for this sample.*

Lines 337-339: Of course!

*Ok.*

Lines 340-341: Higher P conditions are better shown by the composition of phengite and omphacite.

*We are unsure about this comment. The data from Laurent et al., 2018 show that some garnet rims (using the garnet-pyroxene-phengite themobarometer) grow at high P conditions (determined to be peak). For further studies that use thermodynamic modeling and show garnet growth at high-pressure (peak) conditions see Dragovic et al., (2012), (2015), and Groppo et al., (2009).*

**Page 12**

Lines 347-349: Clearly garnet is not the good target to estimate peak P-T as a large quantity of garnets are clearly not crystallizing at peak P-T conditions but during the late prograde P-T path. Would you be able with this technique to measure quartz inclusion in omphacite? No idea if you can find some but this would be a better approach to estimate peak P-T conditions of the CBU.

*We disagree with the statement that garnet is not a good target for estimating peak P-T conditions. We describe the textural relationship of garnets in all of our samples (retrogressed), which suggests that garnets are the best candidate for extracting the max P conditions from these rocks. Further details are discussed in the opening paragraphs of this reply.*

*Quartz-inclusions-in-omphacite may not be suitable for elastic thermobarometry. The bulk modulus of omphacite is low enough (such that the bulk modulus difference between quartz and omphacite is not large), and thus it is neither a great barometer or thermometer.*

Line 355: Yes I agree

*Ok.*

Line 365: We never said that this was rapid... We actually think that it was unexpectedly slow for so well preserved HP-LT rocks... Some of our not yet published data (so you can't know of course) plus other studies (e.g. Cliff et al., 2016) suggest that the CBU was still in blueschist P-T conditions at 37 Ma while peak of metamorphism is dated at 52-50 Ma.

*This is a good point. This has been changed to reflect the difference between the Laurent et al. 2018 study and this study, in regards to the T conditions of the CBU at ~ 1.0 GPa (lines 391 – 393). Our results do not support transition to a back-arc at ~1.0 GPa, but rather the CBU remained with a cool fore-arc.*

*"Laurent et al. (2018) interpreted reheating to indicate that CBU rocks on Syros reached high-P conditions, and then transitioned from a forearc to back-arc setting at ~1.0 GPa, thus experiencing a period of increasing temperatures."*

**Page 13**

Lines 378-379: These are not peak P-T conditions . Just a minimum pressure condition.

*Please refer to our opening discussion for this comment.*

Lines 383-385: I have many points to add here... We have done an isochemical phase diagram (pseudosection) for this sample. And the mineral assemblage stable at these conditions is: Grt (matching perfectly the Grt composition measured and the estimated Vol% in the rock), Ab (observed in this sample and also observed included within the garnet), Bte (not observed in this sample but expected to be only few Vol% of the assemblage), Paragonite (observed in this sample), Omphacite (not observed but few Vol% again), Chl (observed), Ep (observed), Sph (not observed but few Vol%) and qtz (observed). So your argument here is not valid, please remove it. I agree Gln is not stable at these specific conditions but Gln is stable at just slightly higher P (in the AEBS facies) and so have just been preserved.

*33% (3 of 9 minerals) of the sample mineralogy has not been observed, and glaucophane is missing in the pseudosection prediction. Glaucophane is predicted to have formed in textural equilibrium with the above assemblage (based on the comment below). Furthermore, in this study (sample KCS70A from Kalamisia), we observe glaucophane and epidote growth nearly simultaneously (also see thin section scan of sample KCS70A in supplementary material).*

Lines 385-386: I agree with this statement. That's why the rocks that have been the most impacted by this heating phase on Syros are the one that are the most overprinted in the GS-facies.

*The data presented in this study does not support with this statement. The data presented in this study suggests that greenschist facies rocks associated with $D_{t2}$ deformation formed at ~400 °C (stable isotope thermometry of quartz-calcite pairs from boudin necks) and ~1.0 GPa (qtz-in-ep2 barometry). Therefore, the rocks most overprinted at greenschist facies (this study) are not related to re-heating, but rather distinct deformation events and recrystallization. Furthermore, we relate*

*the retrograde blueschist facies metamorphism seen here to distinct folding ($D_{t1}$). We think $D_{t1}$ and $D_{t2}$ are the same folding event, but strain progressively localizes into weaker lithologies (e.g., metavolcanics and metasediments, as opposed to rheologically stronger metabasite) during CBU exhumation, and thus these rocks that are retrogressed to greenschist facies assemblages preserve further rotation of lineations towards the E-W (unrelated to heating). An exception is blueschist facies rocks below the Vari detachment (at Fabrikas), where we document E-W lineations (as opposed to NE-SW lineations) and asymmetric shear (also c.f., Ring et al. 2020), that we attribute to proximity to the dominant strain boundary (Vari detachment) and thus full re-orientation of lineations towards the E-W (at blueschist facies conditions). We refer to Kotowski et al., (in review) for further details.*

Lines 386-389: Yes and this is true! In this sample sample we can observe textural equilibrium between albite, glaucophane and epidote.

*We also observe near textural equilibrium between glaucophane and epidote. Here we refer to P-T conditions for exhumation constrained in Laurent et al., 2018 (P ≈ 1.0 GPa and T ≈ 550 °C) from garnet modeling, wherein the mineralogy of the sample used to constrain these P-T conditions (the sample lacks biotite but has glaucophane in equilibrium with epidote and albite, as stated above) more closely approximates blueschist facies conditions, rather than epidote-amphibolite facies (where the P-T conditions calculated from GrtMod fall).*

Lines 389-392: Yes and if you look carefully the residue (i.e. the quality of the prediction) is largely better when using the local bulk composition. This is true for most of our analyses and is due to the fact that rocks from the CBU are strongly foliated.

*This is a good point for clarity. The residuals are generally lower for modeling that uses local compositions. This has been added (lines 417-420):*

"*Models that use bulk compositions suggest that the core and mantle of the garnet record P-T conditions of ~1.8 GPa and 475 °C, whereas models that use local compositions suggest that the garnets do not record conditions above ~1.0 GPa (model residuals are lower using local bulk composition models).*"

Lines 393-394: Wrong. Both data are shown in our JMG paper and so the reader can see how the chosen bulk rock compo impact the P-T estimates! We have done that while many studies don't! I don't understand why you are trying to strenuously criticise our work practically all along your paper while your data are consistent with our findings (in a certain manner explained previously). The constraints you add on the exhumation path don't go against our proposed P-T path, they are just complementing them (plot them on your figure 6a and you will see). We can definitely be wrong but please note also that this phase of reheating has also been shown on other Cycladic Islands (e.g. Tinos, Andros). And of course more works are needed to precisely quantify the 'amplitude' of this phase. The only thing that we know is that it has not exceeded 550 ˚C (from our RSCM data). Note also that the amplitude of this heating phase is probably lower on Syros than on Tinos and Andros.

*We had added this sentence to the previous version of the manuscript to highlight how the chosen compositions may affect the P-T results, solely as a discussion point to compare/contrast results and highlight potential explanations for the observed discrepancies. We still think this sentence is valid, but we have now removed the final sub-sentence. In regards to the comparison of our P-T path and that of Laurent et al. (2018), our data/results diverge in this case. We now plot the P-T path from this study and compare it with that of Laurent et al. (2018) (amongst other studies) on Figure 7b. Boudin neck precipitates (quartz-calcite) related to $F_{t2}$ folds suggest colder temperatures (~400 °C; mean = 411 ± 23 °C), whereas Laurent et al. (2018) suggest a temperature of ~550 °C (both at ~1 GPa). That is, our data do not support re-heating. Furthermore, we don't see evidence for different sections of the CBU having experienced different exhumation P-T paths. We propose that different sections of the CBU seem to follow similar P-T paths during exhumation, based on the results of this study.*

Line 395: I agree with that. Peak P-T conditions are similar in all the CBU (instead it is in the Upper Cycladic Blueschist Unit --> see Grasemann et al., 2018).

*The results in this study suggest that different sections of the CBU reach similar pressures, but we have not analyzed enough samples throughout Syros to say this with certainty. So we think this is still an open question (do all CBU rocks reach the same peak pressures)? The Laurent et al. (2018) results do clearly show that different sections of the CBU on Syros reach similar T's, but as discussed in the opening discussion, the peak T may not necessarily correlate with the peak P. The Upper Cycladic Blueschist Nappe is further discussed in the lines that follow this comment (lines 423-425):*

*"The similar peak pressures (> 0.8 GPa) between different Syros outcrops suggests that these rocks belong to the Upper Cycladic Blueschist Nappe (Grasemann et al., 2018)."*

Line 396: This is most of the time true. However, you can also observed very little deformed samples/outcrops that are strongly overprinted in the GS facies. So deformation don't explain everything.

*For the rocks that we use in this study, recrystallization is related to distinct deformation (folding) events. It is true, there are places on Syros where greenschist facies overprinting appears static. However, we have only observed static greenschist facies overprinting on Syros adjacent to late-stage normal faults (see Kleine et al., 2014; Skelton et al., 2018 for examples). These faults post-date the early exhumation history that we were interested in decoding with this work.*

Lines 399-401: Please explain why? What do you mean by ' similar P-T conditions reached at different locations'? If you mean similar P-T estimates measured in different outcrops I really don't understand why this would be inconsistent with 'results that suggest individual [...] and different sections of the CBU'.

*The results from this study, i.e., the P-T results determined from different outcrops, suggest similar P-T conditions (peak and retrograde) at each location. Previous work on the CBU on Syros has suggested that the base of the CBU experienced heating (Laurent et al., 2018), or that eclogites, blueschists, and greenschists, followed different P-T paths upon exhumation (Trotet et al., 2001b,*

*2001a). Our results are not consistent with these previous studies. Rather, some rocks have progressively have accommodated strain and are retrogressed at different facies (unrelated difference in P-T conditions experienced).*

Line 399: :-) This should be included from the beginning in your geological setting.

*This has been moved to the geologic setting and is reworded here (lines 77-79 and 423 - 425).*

*"The CBU has been separated into the "Upper Cycladic Blueschist Nappe" and the "Lower Cycladic Blueschist Nappe" on Milos Island; the Upper Nappe records peak pressure conditions above ~0.8 GPa (Grasemann et al., 2018)."*

"*The similar peak pressures (> 0.8 GPa) between different Syros outcrops suggests that these rocks belong to the Upper Cycladic Blueschist Nappe."*

Lines 401-402: Again I don't understand why. What would be interesting here is to measure the quartz inclusions in the garnet of SY-14-07. We can (and should) do that.

*Our results suggest no re-heating at ~ 1.0 GPa, but rather cooling to ~400 °C (at 1.0 GPa). Our results also don't suggest different CBU rocks on Syros followed different P-T paths. This is discussed on lines 425-427 and 453-455:*

*"The observation of similar P-T conditions reached at different locations is inconsistent with results that suggest individual P-T paths for rocks that preserve different metamorphic facies (Trotet et al., 2001b, 2001a), and different sections of the CBU (Laurent et al., 2018); however, we do not have T constraints for rocks from southern Syros."*

*"Our new results show that CBU rocks from Syros, Greece, experienced similar P-T conditions during subduction and exhumation, inconsistent with results that suggest different P-T histories for CBU rocks for Syros or increasing temperatures during exhumation."*

**Page 14**

Lines 410-412: References are incomplete here. There are a lot of studies who studied/reported stretching lineations directions before you (and us!). As a parenthesis, we already suggested (before Behr et al., 2018) an early N–S oriented prograde event in the subduction channel (Laurent et al., 2016, section 6.2). And also you have to consider info from many of the Cycladic islands where the CBU is observed when thinking about the exhumation of this HP-LT unit. For example, you will see that only Syros show E-W stretching lineations. They are more oriented NE-SW on many of other islands. You have to mention (and explain) that here.

*Further studies that also document stretching lineations are now referenced (lines 435-438):*

*"During this phase of exhumation, CBU rocks remained within a cold forearc until they reached the mid-crust (~1.0 GPa), and exhibit a progressive change in kinematics, from N-S stretching lineations during subduction (e.g., Behr et al., 2018; Laurent et al., 2016; Philippon et al., 2011),*

*to lineations that swing towards the E-W during exhumation (c.f., Kotowski and Behr, 2019; Laurent et al., 2016)."*

*The difference between detachment-related lineations on neighboring islands and those recorded on Syros are now mentioned and referenced (lines 438 – 440):*

*"Stretching lineations in the footwall of the North and West Cycladic Detachment Systems have top-to-the- NE and SW orientations, respectively (e.g., Brichau et al., 2007; Grasemann et al., 2012; Jolivet et al., 2010; Mehl et al., 2005)."*

"Lines 412-415: Please add references and say that your model agree with previous models proposing that. But yes I agree and this is what we think also (see Roche et al., 2016 for reconstructions).

*We have added references; however, we note that our model does not agree with some studies that suggest the CBU on Syros records back-arc conditions (lines 440 – 444):*

*"The inferred P-T conditions and kinematics of our studied samples are consistent with Syros recording early deformation and metamorphism within a forearc setting, whereas adjacent Cycladic islands that border the North and West Cycladic Detachment Systems record late-stage kinematics and greenschist facies metamorphism that capture the CBU transition to a warmer back-arc setting (e.g., Laurent et al., 2016; Ring et al., 2020; Roche et al., 2016; Schmädicke and Will, 2003)."*

**Figure Comments**

**Figure 1**

This map is missing all the shear zones and faults. Why exactly?

*Some faults and shear zones have been added. For a detailed Syros map we refer to Keiter et al. (2011).*

Vari detachment

*Vari detachment now drawn.*

Major normal fault here

*Added.*

Observed in which minerals? ("referring to stretching lineations")

*The minerals that define stretching lineations in each locality are extensively discussed in the text. Please refer to:*

*Lines 101-103: "The early $S_s$ fabric is re-folded by upright folds ($F_{t1}$) with steeply dipping axial planes, NE-SW-oriented fold hinge lines, and NE-SW-oriented stretching lineations primarily defined by white mica, glaucophane, and epidote; this indicates folding under blueschist facies conditions ($D_{t1}$)."*

*Lines 114-117 and herein: "This early fabric was locally retrogressed and re-folded by upright folds (considered $F_{t2}$) with steeply dipping axial planes, E-W-oriented fold hinge lines, and E-W-oriented stretching lineations primarily defined by white mica, chlorite, and actinolite (considered $D_{t2}$, Fig. 2a,b); this indicates folding under greenschist facies conditions"*

Normal? (referring to "high-angle fault")

*"normal" added to map. Now reads: "High-angle normal fault".*

This area is composed of HP metabasites with a lot of eclogites. If you don't trust me, you will probably better trust Kotowski and behr (2019) Fig.10 of Skelton et al., (2019).

*We have changed the Fabrikas map portion to show the high-pressure metabasites. The reviewer is correct, there are many HP blueschists and eclogites at Fabrikas. However, as opposed to outcrops such as Kalamisia and Kini, this outcrop also contains quartz-rich schists and marbles that make-up a significant volumetric proportion of this outcrop [c.f., Kotowski and Behr 2019 (fig. 10), Skelton et al., 2019 (fig. 7), Kleine et al., 2014 (fig. 3)]. Furthermore, significant portions of the outcrop are cross-cut by late-stage normal faults that cause localized greenschist facies metamorphism (c.f., Skelton et al., 2019; Kleine et al., 2014). This portion of the map was colored as blueschist-greenschists facies schists because the outcrop is not solely HP metabasites (though the reviewer is correct, there are many), as opposed to other outcrops (e.g., Kini, Kalamisa) that have a greater volumetric proportion of HP metabasites.*

This area is mainly composed of the gneiss of Komito and metabasites locally preserving BS paragenes (and eclogite)

*Gneisses have been added.*

Geological maps should show the tectonic structures!

*We have added more structures on the geologic map.*

Why are you completely ignoring the map shown in Laurent et al. (2016)?

*Structures from Laurent et al. (2016) have now been added.*

**Figure 2**

Add orientation on your field pictures

*Orientations added.*

5 mm? Pen is 2 cm long?

*Thanks for catching this error. The scale bars are correct, but should read cm in Fig 2b.*

**Figure 3**

10mm or cm?

*Cm, as above.*

Add orientation and use a correct scale (except if you have a really small pen) :).

*The reviewer is correct, this was a typo and should be cm. Scale is fixed and orientations have been added.*

**Figure 4**

Can you verify if scale is correct?

*Scale is correct.*

**Figure 5**

Line 670: Group 1?

*The reviewer is correct, the 1 was missing in the figure caption. Fixed (line 745):*

*"Red, blue, and yellow symbols indicate qtz-in-grt (Group 1)…"*

**Figure 6**

Can you explain why T˚ of garnet crystallisation is not reported as in Ashley et al. (2014)? I understand you have recalculated the P but why are T˚ here constrained between 500-550 ˚C while they are mainly between 450-500˚C in their paper? In Behr et al. (2018), the authors estimate T˚ of garnet crystallisation to be comprised between 450 and 550˚C (cf their Fig. 3). Why is it only 500-550 ˚C here?

*Please note that temperature was not quantitatively constrained in Ashley et al. (2014) and Behr et al. (2018). Ashley et al. (2014) used the T constraints from Dragovic et al. (2012) (garnet core) and assumed a constant T increase from core-to-rim, based on the volumetric distance from the center of the garnet. The study does not constrain that most garnet growth occurs between 450 – 500 °C, but rather, the study is limited by the density of inclusions that can be measured in each garnet zone (for quartz-in-garnet barometry). The max T is estimated to be ~550 °C, based on their calculations. We use the max T estimates (~500 – 550 °C), since we are trying to define the peak P-T conditions and retrograde P-T path. We emphasize that the T we assume has a minimal*

*effect on our max P estimate. We use the mean inclusion pressure from Ashley et al. (2014) data, since no systematic changes across garnets are observed. Behr et al. (2018) do not constrain a temperature, they use an estimated T based on previous studies. Since most studies agree that max T is ~ 500 °C (to ~ 550 °C), we use the max T constraints to show our max P (minimal T dependence).*

---

## Referee Report (RR1)

[referee-annotated manuscript omitted]

---

## Author Response (AR2)

*We thank the editor for giving us comments that have helped us better address the reviewers and editors' comments.*

"What should I expect now.... After having gone through your response letter and the revised manuscript, I noted that your detailed responses were not fully incorporated in the revised version, resulting in a slightly modified revised version (this is one of the issues raised by Dr. Laurent). On this regard, I think that commenting further the points raised by Dr. Laurent could reinforce the scientific rationale of your study and robustness of the new approach. Further discussion is needed regarding (i) the peak pressure and (ii) the thermal evolution (i.e. cooling vs. heating exhumation) of the CBU as defined in your study: discussing the limitations and the achievements of the new approach compared to the state-of-the-art. Furthermore, I think your manuscript would benefit of a more extensive discussion in terms of regional implications. I also found section 6.4 not exhaustive and it should be thus expanded more, since, as stated in lines 371-374, your data are relevant to refine the exhumation history and hence the tectonic/geodynamic scenario of the CBU evolution. Your structural and kinematic data should converge in a refined P-T-deformation history for the CBU that is instead just barely discussed."

*We have added 2028 words (previous version = 7465 words, new version = 9493 words) to address the 4 major points brought up by the editor:*

*1) discuss the peak pressures more*
*2) discuss the thermal evolution more (cooling vs heating)*
*3) discuss limitations of elastic thermobarometry*
*4) discuss the tectonic model more*

*1) and 2) are further discussed, and the Gyolami 2021 paper (mentioned by the reviewer) is now discussed in detail (as well as some other Kampos data). 3) We have added a new section (6.4) to discuss the limitations of elastic thermobarometry (over 1 page of discussion) and 4) expanded section 6.5 (exhumation implications) to better describe our tectonic model. Further text/figure modifications were made to address the reviewer's comments.*

*We thank the reviewer for the additional comments that have helped improve this manuscript. We address the reviewer's comments below (italicized). Changes made to the text are in red font.*

"1) Maximum P-T conditions of the CBU

      I did insist in my 1st review that pressure results obtained by the authors on garnet crystallization could be interpreted as not representing maximum P conditions as these results match almost perfectly the P conditions reported by Laurent et al. (2018) for their 1st event of garnet crystallization on Syros (16-18 kbar). In their response to my comment, the authors argued that they have several lines of evidence to say that these are the maximum P-T conditions reached by the CBU (note that only a pressure was determined, the maximum temperature is just inferred from previous studies). In my opinion this is a major issue of the present work… In their paper, the authors are presenting and discussing the different thermobarometric estimations previously obtained in the CBU of both Syros and Sifnos islands, opposing the studies yielding ~1.5 – 1.6 GPa to the studies yielding > 1.9 GPa. This is great but it is not enough. The CBU (and more precisely the Upper Cycladic Nappe as described by Grasemann et al. 2018) is observed in other Cycladic islands such as Tinos, Andros, Ios, Sikinos, Milos and even Naxos. And on practically every of these islands, the most recent published articles that have estimated the peak P-T conditions of the CBU yields consistent results > 1.9 GPa (e.g. Lamont et al. 2020 on Tinos; Huet et al. 2015 on Andros; Augier et al. 2014 on Sikinos; Grasemann et al. 2018 on Milos; Peillod et al. 2021 on Naxos; peak P-T conditions are poorly constrained on Ios, see Huet, 2010). Additionally, we can add another new published study on Syros, Gyomlai et al. (2021) (not discussed in the reviewed study as published after re-submission) who found maximum P-T conditions on Syros to be ~2 GPa (I will come back on this study in point 2 as results show a reheating phase at 1.0 – 1.2 GPa that contradicts another conclusion of this work).

*We address this comment in multiple parts:*

**"Missing" high-P rims:** *We agree with the reviewer that we may have "missed" the high-P rims that the reviewer found in Laurent et al. (2018), and directly stated this in the manuscript:*

*Lines XX – XX: "Sample SY1401 is collected from the same locality as ours (Kalamisia), but our qtz-in-grt results from this study suggest that garnets from this outcrop record the statistically lowest $P_{trap}$. It is possible, however, that we did not sample the same rocks as Laurent et al. (2018), or that we have not found or analyzed garnets that record high pressures."*

*To make this more explicit, we now also state this in the opening of section 6.3 (lines 328 – 329):*

*"We herein discuss our qtz-in-grt barometry results as max pressures obtained from our sample suite, but acknowledge that we may have missed high-P rims that have been found in other studies from the CBU on Syros (e.g., Laurent et al., 2018)."*

     *We have no argument against "missing" the high-P garnet rims; it's a real possibility. However, despite that it is possible, we have several reasons to think it is not very likely. We have added further text in the manuscript that justifies our interpretation that the garnets in our study are recording peak P conditions (lines 321 – 329):*

*"Several observations support that the qtz-in-grt barometry results record max P conditions of the CBU on Syros: 1) quartz inclusion transects across garnet core-to-rims show no systematic change in $P_{trap}$ (Fig. 6), suggesting that pressures conditions did not change significantly during garnet growth, 2) max pressures from this study are equivalent to qtz-in-grt barometry results from prograde-to-peak eclogites and blueschists (non-retrogressed) from the CBU on Syros (Behr et al., 2018), 3) retrograde ep1 pressures, do not exceed those recorded by qtz-in-grt barometry, and 4) several studies from the CBU have used garnets to constrain max pressures, suggesting that garnets are suitable for constraining maximum pressures (e.g., Laurent et al., 2018; Dragovic et al., 2012, 2015; Groppo et al., 2009)."*

***Retrogressed samples not recording peak conditions:*** *Within line-by-line comments, the reviewer has also commented on our rocks being retrogressed (correctly so and clearly stated in the original text), and therefore do not record maximum P-T conditions. Retrogression should have no effect on the analyzed garnets (and thus pressures), the garnets still record prograde-to-peak metamorphic conditions, even though the matrix is retrogressed. We refer to the data from Behr et al. (2018) or Ashley et al. (2014) for further studies that record the same maximum P conditions from prograde-to-peak blueschists, eclogites, and metasediments (compared in this paper).*

***Pressure comparison with Laurent et al. (2018):*** *We mentioned in the previous replies and in this manuscript that there are significant differences in interpreted results based on application of identical techniques-- garnet modeling-- but conducted by different authors (Laurent et al., 2018; Groppo et al., 2009; Skelton et al., 2018; Dragovic et al., 2012, 2015). Initial garnet growth at ~1.6-1.8 GPa is the result from one of these studies (Laurent et al., 2018), but our work suggests that these seem to be the conditions throughout the duration of garnet growth for the samples we examined (from core-to-rim inclusion transects). Other studies document garnet growth that begins at ~ 1.1 - 1.2 GPa (Skelton et al., 2018). Our results better agree with other garnet modeling results that document near isobaric garnet growth (Groppo et al., 2009; Dragovic et al., 2012, 2015). Albeit, our pressures are statistically lower than the peak pressures from the aforementioned studies (~2.0 – 2.2 GPa). Our results best agree with the thermodynamic modeling results from Skelton et al. (2018) (~1.9 GPa peak pressure from garnet inner rims). This is extensively discussed in section 6.2 (3 pages of discussion).*

***Comparison with CBU rocks from other islands:*** *We had originally focused on comparing reference P-T conditions from the CBU on Syros and Sifnos, because there are reference quartz-in-garnet barometry data from the CBU on Sifnos. Clearly, other studies on other islands have constrained peak P-T conditions that significantly exceed 1.9 GPa (e.g., Tinos: Lamont et al., 2020, max $P \approx 2.3 – 2.6$ GPa). We highlight multiple thermodynamic modeling studies from Syros that record P conditions that exceed ~ 1.9 GPa. However, previous studies from the CBU (from the same island and different islands), significantly disagree. In this case, we think a detailed comparison of all data from other islands is beyond the scope of this manuscript. We provided a detailed comparison (6 pages of discussion) for the CBU on Syros (and some Sifnos data) to show how thermodynamic modeling, phase stability constraints, and elastic thermobarometry results all suggest different max P conditions. We believe that this is the most important comparison for the work from this study. In general, phase stability and elastic thermobarometry constraints generally*

*agree, and agree with the lower P estimates from thermodynamic modeling, but disagree with the higher P estimates for the CBU on Syros (max P ≈ 2.2 ± 0.2 GPa, Laurent et al. 2018). There are many complications with discussing the CBU in all of the islands in detail; e.g., the reviewer mentions above that Huet et al. (2015) found high-P conditions (> 1.9 GPa) on Andros. However, high-P conditions have never been accurately estimated on Andros (Huet et al. 2015). Huet et al. (2015) constrain maximum P conditions of ~ 1.5 GPa, and used the P conditions from neighboring islands (i.e., Tinos), to suggest that high-P conditions were found on Andros.*

*Huet et al., (2015): "**a. The P-T-t path of the Attic–Cycladic Blueschist unit on Andros** Since more information about the P-T-t evolution of the Attic–Cycladic Blueschist unit is available, it is discussed first. The available peak P-T conditions for this unit are 450–500 °C and minimum pressure of 10 kbar (Reinecke, 1986). Following Bröcker & Franz (2006), **we use peak data from Tinos island – 500– 520 °C and 16–18 kbar (Parra, Vidal & Jolivet, 2002)** – which is compatible with the wider peak range of 450–550 °C and 12–20 kbar (Fig. 7a) determined by Bröcker (1990)."*

[Figure]

*Figure 8 Huet et al., (2015): the dashed line indicates that peak P-T conditions are inferred.*

*We think discussing other island CBU constraints in the same detail as done for the CBU on Syros, would be too much auxiliary information. We have nonetheless added a description of results from previous studies on other islands in the manuscript (lines 79 –85):*

*"Previous studies have reported a wide range of maximum P-T conditions for rocks from the Upper Cycladic Blueschist Nappe on different Cycladic islands [Sifnos: ~1.4 – 2.2 GPa and 450 – 550 °C (e.g., Schmädicke and Will, 2003; Groppo et al., 2009; Dragovic et al., 2012, 2015; Schliestedt and Matthews, 1987; Matthews and Schliestedt, 1984; Ashley et al., 2014; Spear et al., 2006); Tinos: ~1.4 – 2.6 GPa and ~450 – 550 °C (e.g., Bröcker et al., 1993; Lamont et al., 2020; Parra et al., 2002); Naxos: ~1.2 – 2.0 GPa and ~450 – 600 °C (e.g., Avigad, 1998; Peillod et al., 2017, 2021); Sikinos: ~1.1 – 1.7 GPa and ~ 500 °C (e.g., Augier et al., 2015; Gupta and Bickle, 2004)]."*

**Data from Gyomlai et al. (2021):** *The reviewer mentions:*

"Additionally, we can add another new published study on Syros, Gyomlai et al. (2021) (not discussed in the reviewed study as published after re-submission) who found maximum P-T conditions on Syros to be ~2 GPa"

*We discuss the peak and retrograde data from Gyomlai et al., 2021 in more detail because the data are from the CBU on Syros (lines 417 – 433). We had not focused on describing studies from Kampos; we now include results from other studies on Kampos as well. Gyomlai et al. (2021) report a pressure value for one sample, in passing, of 1.9 ± 1.99 GPa (the absolute error is technically within error of our P estimates), but the large error (± 1.99 GPa), means that this value is not valid, a point the authors of that paper fully acknowledge:*

*Gyomlai et al. (2021): "Matrix sample L_p135 (L transect): the metasomatic assemblage of Ca-amphibole ($Na_{0.438} Ca_{1.545} Mg_{4.120} Fe2+_{0.561} Fe3+_{0.206} Mn_{0.016} K_{0.019} Al_{0.285} Si_{7.890} O_{22} (OH)_2$), chlorite ($Mg_{4.153} Fe2+_{0.838} Mn_{0.009} Al_{1.912} Si_{3.063} O_{10} (OH)_8$) and talc ($Mg_{2.794} Fe2+_{0.138} Si_{4.030} O_{10} (OH)_2$) **is unconstrained for pressure (1.90 ± 1.99 GPa) but yield temperatures of 561 ± 78 °C**."*

There is also a sentence in the response to my comments that worries me… The authors says 'However, we consider it more likely that different techniques are recording different pressures'. What does that mean? Different thermobarometric techniques yield to different maximum P-T conditions for the same rocks? But in this case, it also means that some of these techniques can't be used to retrieve maximum P-T conditions in these rocks. And in this case, should we better trust 'conventional thermobarometry' as stated by the authors, a technique that have proved to be reliable in every HP-LT belt observed all over the world, or the relatively new technique used by the authors in this study? Perhaps, the question that should be treated in this paper is more: why is the elastic thermobarometry technique yielding to lower P-T conditions than more conventional thermobarometry? Note that Peillod et al. (2021) used the Qtz-in-Grt technique in rocks of the CBU in Naxos and find maximum pressures of ~ 2 GPa.

*We do suggest that different techniques/methods are resulting in different pressures for the CBU. This is simply what the data show. This is by no means a new observation-- disagreements over pressure-temperature estimates from different techniques are all over the literature from every orogen. These disagreements are what fuel further study and encourage new approaches and comparisons between approaches, as we have done here. Specifically for the CBU, the thermodynamic modeling technique in particular also indicates higher pressures on other Cycladic islands, in comparison to other techniques. E.g., max P conditions from Tinos vary significantly. Lamont et al. (2020) used thermodynamic modeling, and constrain a max P up to ~2.3 - 2.6 GPa. Parra et al. (2002) used chlorite thermobarometry, and constrain a max P of ~ 1.8 GPa, and Bröcker et al. (1993) used more qualitative constraints (jadeite content in pyroxene and Si-in-phengite), and estimate minimum pressures of ~ 1.5 GPa.*

*In regards to max pressures from thermodynamic modeling vs elastic thermobarometry for the CBU on Syros, the max pressures from quartz-in-garnet barometry tend to be lower than the absolute values from thermodynamic modeling. Thermodynamic modeling has certainly been used*

*globally, but the more appropriate question is whether the determined P-T conditions are accurate. We do not have a definite answer for which technique provides the accurate pressures, but we have interpreted our results with the support from external constraints (e.g., rock mineralogy, phase stabilities, equilibria reactions), since pressure-temperatures results should be consistent with basic field and petrographic observations. Discrepancies between techniques is a common problem, e.g., see discrepancy in results between graphite thermometry (Laurent et al., 2018) and Zr-in-rutile thermometry (e.g., Spear et al., 2006) for max T of the CBU. We agree, some techniques are clearly not providing the correct results; however, it is difficult to state which technique is providing more accurate results. Recent experiments have shown that quartz-in-garnet barometry can provide accurate pressures (± 0.1 – 0.2 GPa), when strain is purely elastic (Thomas and Spear, 2018), and we believe that the garnet grains from Syros have only experienced elastic relaxation due to the low maximum temperatures reached by these rocks, and the petrography of the rocks. Garnet flow laws predict that viscous strain of garnet won't occur below ~ 650 °C at geologic strain rates (Wang and Ji, 1999; Ji and Martignole, 1994). In this sense, many recent studies have started to use elastic thermobarometry as the more accurate barometer, due to the sensitivity of thermodynamic modeling to many input components, and potential overstepping of metamorphic reactions. The debate about which technique is providing the accurate pressures, is a very active debate (e.g., see Wolfe and Spear, 2018; Spear and Pattison, 2017). The quartz-in-garnet barometry data could absolutely be incorrect; however, we currently don't have a good explanation for why it would be incorrect. We have added a new section (section 6.4, lines 468 - 511) that explains possible sources of error with elastic thermobarometry results.*

Lines 86-87 the authors write 'The range of previous P-T conditions reflects the lack of comprehensive studies that combine structural geology, petrology, and thermobarometry across the CBU'. I already told the authors that they can't say a statement like that. The CBU is one of the best studied HP-LT belts worldwide for 40-50 years now. In the literature, there are a multitude of research groups and studies that have combined structural geology, petrology and thermobarometry in the CBU at a level of integration that is far more advanced than in this study (e.g. Ring's group; Grasemann's group; Jolivet's group; Lister's group; and so many others...). I personally dedicated most of my PhD thesis on determining the tectonometamorphic history of the CBU by coupling and integrating structural geology (Laurent et al., 2016; Roche et al., 2016), thermobarometry (Laurent et al., 2018) and geochronology (Laurent et al. 2017) with more than 8 months spent at studying in the field the CBU on Syros, Sifnos, Tinos and Ios. So no, the range of previous P-T conditions doesn't reflect the lack of comprehensive studies that combine structural geology, petrology and thermobarometry across the CBU. Such P-T range is observed in the literature of most HP-LT belts across the world and is just the result of the numerous different research groups that have studied these rocks since decades. It is important to note that the implications of the results of this study on the tectonometamorphic history and exhumation model of the CBU is not really clear (see my points 2 and 3 below)."

*This is a fair point. We did not intend to say that no studies have combined structural geology, petrology, and P-T constraints. Clearly several have. We have removed this sentence to avoid confusion.*

2) Results of this study show cooling during exhumation and do not support a phase of reheating at 10-12 kbar.

Another main conclusion of this work is that the P-T path obtained for the CBU on Syros show a constant cooling during retrogression, from peak to greenschist-facies P-T conditions. While I quite agree with that, I am not really sure that the data obtained in this study are sufficient to make such conclusion. The authors make it clear in their paper that the entrapment temperature (Ttrap) of quartz inclusions in garnet (garnet growth temperature) is estimated at 500-550°C based on good agreement between previous studies on the maximum temperature reached by CBU rocks from Syros. Ttrap for the ep2 population is deduced from oxygen isotope thermometry of quartz-calcite boudin-neck precipitates (411 ± 23°C). However, Ttrap for the ep1 population is not constrained (in this study or any previously published studies) and estimated as being intermediate between garnet and ep2 growth (~400-500 °C). This last hypothesis means that the P-T path can't show anything else than constant cooling during exhumation. Trotet et al. (or anyone else) could legitimately argue that if you consider a Ttrap of 500-550 °C for Ep1 (as this is not constrained in the study), your P-T path would show a 1st event of isothermal exhumation from peak P-T conditions to blueschist facies conditions and then a 2nd phase of cooling from blueschist to greenschist facies conditions (something really similar to what Trotet et al., 2001 proposed). In summary, what I want to highlight here is that the conclusion that the P-T path proposed in this study shows constant cooling during exhumation is only based on the unconstrained assumption of Ttrap for Ep1 that can't yield to another result.

*We have petrologic and tectonic reasons for not considering isothermal decompression during the early stages of exhumation (between 500-550 °C and ~411 °C), and especially the lack of evidence for reheating. Petrological: Isothermal decompression can be argued for, but that would lead to some interesting petrologic consequences for rocks from Syros, Greece. These rocks would then cross the lawsonite --> kyanite + zoisite reaction (terminal retrograde lawsonite-out reaction), but kyanite has not been found on Syros. A second petrologic reason, is that there is no evidence of mineralogy that suggests isothermal decompression, or transition to a stability field where biotite would be stable. Amphibole zonations (magnesio-riebeckite ▯ winchite ▯ actinolite) also indicate that the rocks underwent cooling during decompression (c.f., Kotowski et al., in review); high-temperature amphibole compositions have not been found on Syros (e.g., pargasite/hornblende). Tectonic: Cooling during decompression would be achieved due to convective heat transfer from the exhuming rocks (CBU on Syros) into the cooler, subducting lithosphere. This would be consistent with subduction channel exhumation, but isothermal decompression would be much more difficult to explain within a subduction channel. Adding isothermal decompression to the even earlier P-T path, would be altering the data to show something that the data simply does not support. Furthermore, explaining significant cooling at mid-crustal depths (after isothermal decompression), is challenging. We have now described evidence for why we don't think isothermal decompression is appropriate (lines 439 – 447):*

"*We do not have a temperature constraint for the ep1 population; however, we consider cooling during decompression from garnet growth (~500 – 550 °C) to ep2 growth (~400 °C), to be the most likely P-T path for CBU rocks from Syros. Isothermal decompression from ~1.8 GPa and*

*~500 – 550 °C to ~ 1.0 GPa, would lead to terminal lawsonite breakdown above ~ 450 °C and produce kyanite + zoisite (Hamelin et al., 2018; Schumacher et al., 2008); however, kyanite has not been found on Syros, therefore requiring temperatures below ~450 °C at ~ 1.0 GPa. It is possible that sluggish kinetics did not lead to lawsonite breakdown, but given the prevalent evidence of retrograde deformation on Syros and the extensive presence of retrograde overprinting/mineral growth, we consider kinetic-limitations to be unlikely. Furthermore, the chemical evolution of amphiboles (magnesio-riebeckite ▯winchite ▯actinolite) suggests that CBU rocks from Syros followed a cold P-T path during decompression (c.f., Kotowski et al., 2020)."*

*We remind the reviewer that previous studies have imposed cooling during decompression during initial CBU exhumation (Laurent et al., 2018; Gyomlai et al., 2021), but there was actually no data from those studies that pinned cooling during decompression. Cooling during decompression was schematically imposed, because it was the tectonic model that the authors preferred. We show the data from previous studies below:*

[Figure]

    *Laurent et al. (2018): Figure 17d*        *Gyomlai et al. (2021): Figure 16*

*Our data does show the initial cooling during decompression, but does not support re-heating at ~ 1.0 GPa.*

My second comment is that this study claims their results don't support the reheating phase at 10-12 kbar and from ~500 to 550°C proposed by Laurent et al. (2018). I did insist in my 1st review that for me, the results of this study don't contradict the existence of this reheating phase but rather add more constrains on the greenschist P-T conditions during exhumation. The authors of this study completely have the right to disagree with what Laurent et al. (2018) proposed (even more considering that this is shows by only 1 garnet). However, I would like to highlight that a newly published study (published after the resubmission of this study – Gyomlai et al. 2021 in Lithos) has determined exactly the same reheating phase after looking to fluid-rock interactions and metasomatism in the northern block-in-matrix structures on Syros. A difference is that they

were able to constrain this reheating phase from rocks located at the top of the CBU on Syros, implying that all subunits of the CBU has undergone this reheating phase. As mentioned in my previous review, this reheating phase has already been described in the CBU of Tinos and Andros and has also recently been shown in the CBU of Naxos (Peillod et al., 2021). So here again, I think that if we consider the previously published studies in different Cycladic islands where the CBU is observed (and again some very recently published studies that the authors can't have considered as published after re-submission), it seems that there are various pieces of evidence suggesting the existence of this reheating phase, contradicting one of the main conclusion of this study.

*We remain unsure about how to address this comment. The absolute temperatures from our quartz-calcite boudin data (stable isotope thermometry) in this work simply does not indicate reheating. Clearly, multiple previous studies from the CBU on Syros have constrained different exhumation P-T paths; however, most suggest cooling during decompression (Miller et al., 2009; Marschall, 2006; Trotet et al., 2001; Hamelin et al., 2018), or isothermal decompression (Trotet et al., 2001; Breeding et al., 2004). Gyomlai et al. (2021) is the second study to propose reheating in the CBU on Syros, but it is unclear if their data support reheating. Gyomlai et al., (2021) estimate max T to be 561 ± 78 °C (no pressure constraint), and two retrograde pressure-temperature conditions for the proposed re-heating event at ~ 1.0 GPa: 1.02 ± 0.15 GPa and 505 ± 155 °C, and 1.03 ± 0.11 GPa and 653 ± 27 °C. The retrograde pressures are reasonable (~1.0 GPa with small errors), but the max temperatures have some issues that the authors themselves discuss in the manuscript. The main issue is the T constraint above ~ 600 °C, which would lead to serpentine breakdown across the island of Syros (Guillot et al., 2015; Wunder and Schreyer, 1997); because this result conflicts with the basic observation of serpentinite stability on Syros, the authors disregard this temperature estimate. Furthermore, terminal lawsonite breakdown (retrograde lawsonite-out reaction) should be expected above ~ 450 °C at ~ 1.0 GPa (Schumacher et al., 2008; Hamelin et al., 2018), something we now further discuss in the manuscript (lines 439* *– 447**). The authors are left with a rather uncertain temperature estimate, therefore, of 505 ± 155 °C, which means the temperature could have been as low as 350 °C, or as high as 660 °C. At a T > ~500 °C and ~ 1.0 GPa, we should also be observing biotite across Syros, but biotite has never been found. The authors used the absolute value of 505 °C, to suggest that heating occurred at ~1.0 GPa between 500 – 600 °C (below serpentine breakdown). The absolute temperatures of the data can be treated as being correct, but the error bars cannot simply be ignored, along with supplementary petrologic constraints. These data could also be interpreted as linear cooling during decompression down to ~350° C at 1.0 GPa, isothermal decompression, or re-heating. The point being, it is impossible to differentiate between those different possibilities from the presented data. Results from Gyomlai et al. (2021) are now further discussed on lines 417 – 433:*

*"Gyomlai et al. (2021) estimate max and retrograde P-T conditions, but from metasomatic rocks from the Kampos belt in northern Syros. The authors estimate maximum T conditions of 561 ± 78 °C, and two retrograde pressure-temperature conditions: 1.02 ± 0.15 GPa and 505 ± 155 °C, and 1.03 ± 0.11 GPa and 653 ± 27 °C. The retrograde pressures are reasonable (~1.0 ± 0.1 - 0.2 GPa), but the max temperatures raise questions that the authors discuss. Specifically, temperatures above ~600 °C (at ~1.0 GPa) would lead to serpentine breakdown (Guillot et al., 2015; Wunder*

*and Schreyer, 1997); however, serpentine is abundant across Syros. The authors used the 505 ± 155 °C temperature constraint, and a temperature below 600 °C, to suggest their studied rocks reached temperatures between 500 – 600 °C at ~ 1.0 GPa. Several other studies on retrograde metasomatic rocks from Kampos constrain P-T conditions: ~1.17 – 1.23 GPa and 500 – 550 °C (Breeding et al., 2004), ~ 0.60 – 0.75 GPa and 400 – 430 °C (Marschall et al., 2006), and ~ 1.20 GPa and 430 °C (Miller et al., 2009). Breeding et al. (2004) did not constrain a temperature, but used an estimated temperature from Trotet et al. (2001a), and constrained a pressure of ~1.17 – 1.23 GPa at the estimated T of ~500 – 550 °C) using Thermocalc V. 3.2. Marschall et al. (2006) used the garnet-clinopyroxene thermometer and Thermocalc V. 3.01 to calculate temperatures, and estimated a pressure based on jadeite + SiO₂ □ albite reaction. Miller et al. (2009) used Perple_X and the thermodynamic database of Holland and Powell (1998) to calculate P-T conditions from reaction zones. In general, most studies indicate cooling during decompression for metasomatic rocks from Kampos, with the exception of interpretations by Gyomlai et al. (2021); however, the large uncertainty in their temperature estimate (505 ± 155 °C) makes it difficult to differentiate between cooling during decompression, isothermal decompression, or re-heating."*

*We don't have an issue with the heating event on places like Naxos, which is next to a migmatite dome, and where biotite is prevalent (and other islands adjacent to large-scale detachment systems, where biotite is also found). We reiterate, that if the above temperatures were reached by rocks from Syros, biotite and/or pargasite/hornblende should be common across the island, but these phases have never been observed (we have also searched). Furthermore, the presence of lawsonite pseudomorphs and absence of kyanite suggests that temperatures did not exceed ~ 450 °C at ~ 1 GPa (Schumacher et al., 2008; Hamelin et al., 2018). These conditions would all verge into lawsonite-absent fields, where lawsonite would break down to kyanite + zoisite. It is possible that sluggish kinetics did not lead to lawsonite breakdown, but given the prevalent evidence of retrograde deformation on Syros and the extensive presence of retrograde overprinting/mineral growth, we consider kinetically-limited breakdown to be unlikely.*

3) Implications of the results on the tectonometamorphic history and exhumation model of the CBU.

Last thing I would like to highlight is about one of the promises of this study, which was to provide a 'more robust' P-T-D path 'than what is commonly possible with conventional thermobarometry' (as this is important to determine the tectonometamorphic history of the CBU and exhumation mechanisms). In my opinion the present study doesn't propose a more robust P-T-D path for reasons exposed above and hereafter. The P-T-D path presented in Figure 7 is mostly incomplete, with only the Dt2 deformation event being shown. What about the other fabrics described in this study? And more importantly, what about the numerous previously published structural studies on the CBU of Syros, Sifnos and the other Cycladic units? There is no discussion of the previously published P-T-D path in the CBU. As actually written, it is practically impossible to understand the view of the authors about the full tectonometamorphic evolution of the CBU.

*We have added further information on the relationship between deformation, mineral growth, and P-T conditions to figure 7. This is also now further discussed in sections 6.4 and 6.5. Initially, we only labelled Dt2 because ep2 is the only mineral for which we know the P-T conditions and can relate it to a distinct stage of mineral growth (Dt2). Ep1 grows transitionally during Dt1-Dt2, and the exact stage of epidote (and when it grows relative to these stages of deformation), is much more difficult to reconcile. The stage of garnet growth (relative to deformation) is also difficult to reconcile, and does not represent a distinct point in P-T space (grows during Ds and Dr). Previous P-T paths from the CBU (Syros) are extensively discussed in sections 6.1 - 6.3.*

The short section (Section 6.4) on the implications for exhumation mechanisms where an exhumation model of the CBU is proposed is not enough detailed in my opinion. Be more specific on what is exactly the input of your study on the tectonometamorphic history and the exhumation model of the CBU? When do you think exactly there is a rotation from N-S to E-W stretching lineations (after peak metam or later during exhumation?) and make it clear how your data support this model (actually I don't see any measurements of N-S stretching lineations in your Fig. 1). Should we understand that, in your opinion, there is no record of deformation acquired during back-arc setting in Syros? You presented different exhumation models (e.g. subduction channel vs. extrusion wedge models) in the geological setting and it would be great to come back on this here. Why do you prefer the subduction channel model compare to other models? Perhaps illustrating your exhumation model would help. I think that this section should be entirely redrawn and further developed to clearly expose the implications of the results on the tectonometamorphic history of the CBU.

*We agree with the reviewer that more information can be added for the tectonic model in this manuscript. This section was originally short, to leave the tectonic model for the manuscript of Kotowski et al. (in review). We have added more information to section 6.5 (previously 6.4) to further discuss the tectonic model. Several comments mentioned above, were previously discussed, e.g.,*

*When does rotation from N-S to E-W lineations occur (lines 516 – 531): "This would suggest that exhumation was achieved parallel to the subducting plate, in a subduction channel geometry prior to core-complex formation. During this phase of exhumation, CBU rocks remained within a cold forearc until they reached the mid-crust (~1.0 GPa), and exhibit a progressive change in kinematics, from N-S stretching lineations during subduction (e.g., Behr et al., 2018; Laurent et al., 2016; Philippon et al., 2011), to lineations that swing towards the E-W during exhumation (c.f., Kotowski and Behr, 2019; Laurent et al., 2016)."*

*Should we understand that Syros does not record deformation in a back-arc setting? (lines 535 – 549): "The inferred P-T conditions and kinematics of our studied samples are consistent with Syros recording early deformation and metamorphism within a forearc setting, whereas adjacent Cycladic islands that border the North and West Cycladic Detachment Systems record late-stage kinematics and greenschist facies metamorphism that capture the CBU transition to a warmer back-arc setting (e.g., Laurent et al., 2016; Ring et al., 2020; Roche et al., 2016; Schmädicke and Will, 2003)."*

*We have added additional information to the two above comments, but in short: yes, we do not think Syros records significant back-arc deformation. Clearly, some back-arc deformation does occur locally in the footwall of the Vari detachment. But the deformation in the footwall of the Vari detachment is significantly more localized in comparison to deformation in the footwall of large-scale detachment systems (e.g., Jolivet et al., 2010; Grasemann et al., 2012; Lamont et al., 2020; Soukis and Stockli, 2013). The reviewer is also correct, no prograde lineations were measured for this study; we are referring to prograde lineations documented in previous studies (Behr et al., 2018; Philippon et al., 2011; Laurent et al., 2016).*

**Line-by-line comments:**

Line 48: not only retrograde.

*Extracting retrograde P-T conditions was the focus of this manuscript (see abstract), but some prograde-to-peak conditions were also constrained. This has been added (lines 47-49):*

*"The purpose of this study is to illustrate the potential of using elastic thermobarometry in combination with structural and microstructural observations, to better understand the P-T-deformation (D) conditions of prograde-to-peak and retrograde mineral growth in subduction-related HP/LT metamorphic rocks."*

Lines 55-56: sorry but I quite disagree. For me your P-T path is not as robust as the ones proposed in previously published works (see my main review report).

*We don't necessarily understand why not, but we can agree to disagree. Each mineral we analyzed had microstructural and structural context, and we could distinguish distinct stages of mineral growth. This sub-sentence ("is more robust than what is commonly possible with conventional thermobarometry"), referred to being able to extract quantitative P-T information from outcrop and microstructurally constrained single mineral growth. Something that usually requires 2 or more minerals in equilibrium with conventional thermobarometry (e.g., our quartz-calcite stable isotope thermometry).*

*However, we have rephrased this to read (lines 54-55):*

*"The results demonstrate that combining qtz-in-ep barometry with careful structural and microstructural observations allows us to delineate a retrograde P-T-D path that is contextually constrained, and provide new insights into the exhumation history of the CBU on Syros, Greece."*

Lines 77-79: Please, be more precise here. In this study they find that maximum P-T conditions of the Upper Cycladic Nappe is ~19.5 kbar at 550˚C.

*This has been changed to read ~2.0 GPa at 550 °C, for the Upper Cycladic Nappe (lines 77 – 79):*

*"The CBU has been separated into the "Upper Cycladic Blueschist Nappe" and the "Lower Cycladic Blueschist Nappe" on Milos Island; the Upper Nappe records peak pressure conditions above ~0.8 GPa (~2.0 GPa and 550 °C; Grasemann et al., 2018)."*

Lines 79 – 83: It is really important to consider works that have been done in the CBU of different Cycladic islands (and more specifically in the Upper Cycladic Nappe as defined by Grasemann et al., 2018). And if you look at the scale of the CBU, there is now a clear consensus that maximum P-T conditions in the CBU is around 2.0 ± 0.2 GPa (see my main review report). Since this study has been resubmitted, at least 2 new studies found maximum P-T conditions around ~2 GPa (one on Syros: Gyomlai et al. 2021; one on Naxos: Peillod et al. 2021). The real question that should be treated in this paper is more: why is the elastic thermobarometry technique yielding to lower P-T conditions? Note that Peillod et al. (2021) used the Qtz-in-Grt technique and find pressures of ~ 2 GPa.

*We agree, it is important to consider all of the islands discussed above; however, the focus of this manuscript is the CBU on Syros. However, as discussed in the manuscript and the above comments, thermodynamic modeling has produced the highest P results in the Cyclades, and most other techniques provide lower P results. We discuss previous P-T constraints from the CBU on Syros in detail (6 pages of discussion) for this purpose. We remind the reviewer that our results are actually in fairly good agreement with 2.0 ± 0.2 GPa, our maximum P is ~1.8 ± 0.2 GPa (2 σ). Our results just don't agree with results that significantly exceed 2.0 GPa. Previous studies (e.g., Laurent et al., 2018), constrain a best-estimate of max P of 2.2 ± 0.2 GPa, but extend their max P estimate to ~ 2.4 GPa.*

[Figure]

*Figure 17d: Laurent et al. (2018)*

*We have also added a full section that discusses the limitations of elastic thermobarometry, and the pressure estimates from this study (section 6.4).*

Lines 86 – 87: Again, for me you can't say that. The CBU is one of the best studied HP-LT belt worldwide since 40-50 years now. In the literature, there are a multitude of research groups and studies that have combined structural geology, petrology and thermobarometry in the CBU at a level of integration that is better than in this study (e.g. Ring group; Grasemann group; Jolivet group; Lister group; and so many others...). There is another thing that concern me... The detailed

structural study that these authors are referring to in their response to my comments is a study that has not been published yet (Kotowski et al.).

*This is a very fair point. The CBU is very well studied. We didn't mean to suggest that other studies have not done a great job; there are many excellent previous studies. We just meant to state that there are few studies from the CBU on Syros that combine structures, petrology, P-T, and timing constraints, all in one study (the reviewer's own work being an exception). We have removed this sentence to avoid this confusion.*

Lines 93-94: One of the main goal of this study is to try to determine maximum P-T conditions in the CBU. I wonder why the authors have not analysed any non-deformed eclogitic sample with unretromorphosed HP-LT paragneses (from the northern part of Syros for example) which seems to be the ideal candidate to retrieve maximum P-T conditions.

*The focus of this manuscript was to primarily address the retrograde P-T path, hence we focused on retrograde rocks. For further elastic thermobarometry constraints from prograde-to-peak rocks from Syros, we refer to the data from Behr et al. (2018), which comes from prograde-to-peak blueschists and eclogites. There is no difference in max P. The garnets record the max P conditions; the amount of retrogression does not affect the pressures (as long as the garnet has survived retrogression). The prograde-to-peak data from Behr et al. (2018) is now explicitly mentioned (lines 321-329):*

*"Several observations support that the qtz-in-grt barometry results, record max P conditions of the CBU on Syros: 1) quartz inclusion measurements across core-to-rims of garnets that show prograde growth (decreasing Mn), show no systematic change in $P_{trap}$ (Fig. 6), 2) max pressures from this study are equivalent to qtz-in-grt barometry results from prograde-to-peak eclogites and blueschists (non-retrogressed) from the CBU on Syros (Behr et al., 2018), 3) retrograde ep1 pressures, do not exceed those recorded by qtz-in-grt barometry, and 4) several studies from the CBU have used garnets to constrain max pressures, suggesting that garnets are suitable for constraining maximum pressures (e.g., Laurent et al., 2018; Dragovic et al., 2012, 2015; Groppo et al., 2009). We herein discuss our qtz-in-grt barometry results as max pressures constraints, but acknowledge that we may have missed high-P rims that have been found in other studies from the CBU on Syros (e.g., Laurent et al., 2018)."*

Line 262: Section 3 should be considered in your Results section (see previous comment). Or at least rename your section 5. as 'Thermobarometry results'.

*We have renamed this section to "thermobarometry results".*

Lines 360: Effectively our sample SY-14-01 is a poorly deformed unretromorphosed eclogite while your sample collected in Kalamisia is described as a retrograde blueschist (Section 3). So clearly, you have not sampled the same rock.

*This is correct, our sample is a retrogressed eclogite (now blueschist), and sample SY-14-01 from Laurent et al. (2018) is a near pristine eclogite. Both eclogites could still be from the same eclogite*

*body (Kalamisia appears structurally coherent), but exhibited different degrees of retrogression. However, this retrogression has no effect on the quartz-in-garnet pressures.*

Lines 362-367: What is the point here? If it has been shown that kyanite is not expected in the CBU of Syros what is the interest of saying that?

*This has been shown for one eclogite composition from an outcrop that is not extensively discussed in this manuscript (Fabrikas). This does not exclude the possible presence of kyanite in other eclogites from the CBU on Syros. Eclogites from other outcrops (e.g., Kini, Kampos, Kalamisia), likely have very different compositions. The eclogites (e.g., Kini, Kampos, Kalamisia) are derived from metabasic protoliths that likely formed in an ocean basin, whereas the protoliths of eclogites from Fabrikas (extremely intermixed with sediments), may represent from shallow intrusion during Cretaceous rifting (or older mafic rocks related to Triassic rifting).*

Lines 409 – 412: There is now a new recent study on Syros that has find exactly the same reheating phase after looking to fluid-rock interactions and metasomatism in the northern block-in-matrix structures (Gyomlai et al. 2021). Please discuss. A difference is that they were able to constrain this reheating phase from rocks located at the top of the CBU on Syros, implying that all the CBU has undergone this phase.

*Different studies have proposed different cooling paths for the CBU on Syros. Most studies propose cooling during decompression (Schumacher et al., 2008; Hamelin et al., 2018; Marschall, 2006; Miller et al., 2009; Trotet et al., 2001) or isothermal decompression (Trotet et al., 2001; Breeding et al., 2004). Two studies now report a heating excursion at ~ 1.0 GPa (Gyomlai et al., 2021; Laurent et al., 2016). Results from Gyomlai et al. (2021) are now further discussed on lines 417 – 433:*

*"Gyomlai et al. (2021) estimate max and retrograde P-T conditions, but from metasomatic rocks from the Kampos belt in northern Syros. The authors estimated maximum T conditions of 561 ± 78 °C, and two retrograde pressure-temperature conditions: 1.02 ± 0.15 GPa and 505 ± 155 °C, and 1.03 ± 0.11 GPa and 653 ± 27 °C. The retrograde pressures are reasonable (~1.0 ± 0.1 - 0.2 GPa), but the max temperatures raise questions that the authors discuss. Specifically, temperatures above ~600 °C (at ~1.0 GPa) would lead to serpentine breakdown (Guillot et al., 2015; Wunder and Schreyer, 1997); however, serpentine is abundant across Syros. The authors used the 505 ± 155 °C temperature constraint, and a temperature below 600 °C, to suggest their studied rocks reached temperatures between 500 – 600 °C at ~ 1.0 GPa. Several other studies on retrograde metasomatic rocks from Kampos constrain P-T conditions: ~1.17 – 1.23 GPa and 500 – 550 °C (Breeding et al., 2004), ~ 0.60 – 0.75 GPa and 400 – 430 °C (Marschall et al., 2006), and ~ 1.20 GPa and 430 °C (Miller et al., 2009). Breeding et al. (2004) did not constrain a temperature, but used an estimated temperature from Trotet et al. (2001a), and constrained a pressure of ~1.17 – 1.23 GPa at the estimated T of ~500 – 550 °C) using Thermocalc V. 3.2. Marschall et al. (2006) used the garnet-clinopyroxene thermometer and Thermocalc V. 3.01 to calculate temperatures, and estimated a pressure based on jadeite + $SiO_2$ ☐ albite reaction. Miller et al. (2009) used Perple_X and the thermodynamic database of Holland and Powell (1998) to calculate P-T conditions from reaction zones. In general, most studies indicate cooling during*

*decompression for metasomatic rocks from Kampos, with the exception of interpretations by Gyomlai et al. (2021); however, the large uncertainty of their temperature estimate (505 ± 155 °C) makes it difficult to differentiate between cooling during decompression, isothermal decompression, or re-heating."*

*As discussed above, Gyomlai et al., (2021) constrain two pressure-temperature conditions for the proposed re-heating event at ~ 1.0 GPa: 1.02 ± 0.15 GPa and 505 ± 155 °C and 1.03 ± 0.11 GPa and 653 ± 27 °C. The retrograde pressures are reasonable (~1.0 GPa with small errors), but the temperatures have some issues that the authors point out in the manuscript. The main issue is the T constraint above ~ 600 °C, which would lead to serpentine breakdown across the island of Syros (Guillot et al., 2015; Wunder and Schreyer, 1997), so the authors did not use this temperature. Furthermore, retrograde lawsonite breakdown to kyanite + zoisite should be expected above ~ 450 °C at ~ 1.0 GPa (Schumacher et al., 2008; Hamelin et al., 2018), something we further highlight in the manuscript now (lines 419 – 433). Serpentine and lawsonite are prevalent on Syros. So the authors were left with a single temperature constraint of 505 ± 155 °C, which means the temperature can be as low as 350 °C, or as high as 660 °C. The authors used the absolute value of 505 °C, to suggest that heating occurred at ~1.0 GPa between 500 – 600 °C (below serpentine breakdown). The absolute temperatures of the data can be treated as being correct, but the error bars cannot simply be ignored, along with supplementary petrologic constraints. This data could also be interpreted as linear cooling during decompression down to ~350 C at 1.0 GPa, isothermal decompression, or re-heating. The key is, it is difficult to differentiate the end-members, based on the presented data.*

Lines 416-419: Again, this is not completely true as only the estimations calculated using a bulk composition and considering Mn suggests that (note that in this case the rim grt is expected to crystallise at ~10-12 kbar / 500-550KC). When Mn is not considered bulk rock compo also yields to results of ~ 10-12 bar 500-550°C.

*This is a good point for clarity, only the SY1407 core/mantle results that used Mn (bulk rock composition) plot at ~ 1.8 GPa [core (1) and mantle (2)]. The remaining results indicate that garnet grew at ~1.0 GPa. This has been clarified on lines 455 – 458:*

*"Furthermore, results from sample SY1407 of Laurent et al. (2018)* sometimes *disagree when using local vs. bulk compositions for modeling. Models that use bulk compositions* and consider Mn *suggest that the core and mantle of the garnet record P-T conditions of ~1.8 GPa and 475 °C, whereas models that use local compositions* or do not consider Mn *suggest that the garnets do not record conditions above ~1.0 GPa (model residuals are lower using local bulk composition models)."*

[Figure]

*Figure 15c (Laurent et al., 2018)*

Lines 420-421: This sentence indirectly suggests that results from Laurent et al. (2018) can be disregarded due to this 'issue'. However, it is very important to note that Laurent et al (2018) have looked to this 'issue' by using (and publishing) results obtained everytime with both a bulk rock and local composition. Clearly for this sample, it seems that there is quite clearly event of garnet crystallisation at 10-12kbar and 550-550˚C. This is now comforted by the recent study of Gyomlai et al. (2021).

*We did not intend to imply that the Laurent et al. (2018) should be disregarded. It is a very good dataset, and the authors do a nice job comparing multiple thermobarometry techniques. But we do think that many previous studies deserved some more "digging" into, to make sure P-T conditions are petrologically consistent. We have removed this sentence; it was not our intention to make this come across this way.*

*However, we do re-iterate that most results from the CBU on Syros (e.g., Schumacher et al., 2008; Miller et al., 2009; Hamelin et al., 2018; Marschall, 2006; Trotet et al., 2001), do not support this heating event.*

Lines 424-425: This was already known (and not subjected to debate) before your study. Syros is - with Sifnos - the Cycladic Island where HP-LT parageneses are the best preserved. So of course it is part of the Upper Cycladic Nappe as described by Grasemann et al. (2018). This is the sort of info that should appear in the Geological Setting not in discussion.

*Agreed, we had mentioned the Upper Cycladic Nappe in the Geological Setting (lines 77 – 79):*

*"The CBU has been separated into the "Upper Cycladic Blueschist Nappe" and the "Lower Cycladic Blueschist Nappe" on Milos Island; the Upper Nappe records peak pressure conditions above ~0.8 GPa (~2.0 GPa and 550 °C; Grasemann et al., 2018)."*

*We mentioned that the CBU rocks from Syros belong to the Upper Cycladic nappe here, since quartz-in-garnet barometry can be a "window" into the peak-to-prograde conditions of rocks from the CBU (for retrogressed rocks). We thought that this would be a good place to re-highlight that all of the retrogressed rocks (primarily greenschists) seem to have reach similar max P conditions, because the max P conditions of significantly retrogressed rocks from the CBU on Syros are not explicitly known from previous studies (there is no data that actually supports this). We've clarified this for readers (lines 460– 463):*

*"The similar peak pressures (> 0.8 GPa) between different Syros outcrops suggests that these rocks belong to the Upper Cycladic Blueschist Nappe (Grasemann et al., 2018), even though in some cases significant retrogression overprinted indicators that would suggest these rocks reached P conditions above ~0.8 GPa."*

Lines 436 – 438: Be more detailed here (when do you think exactly there is a rotation from N-S to E-W stretching lineations - after peak metam or later during exhumation?) and make it clearer how your data support this model (actually I don't see any measurements of N-S stretching lineations in your Fig. 1).

*This is addressed in the same sentence (lines 528-531):*

*"During this phase of exhumation, CBU rocks remained within a cold forearc until they reached the mid-crust (~1.0 GPa), and exhibit a progressive change in kinematics, from N-S stretching lineations during subduction (e.g., Behr et al., 2018; Laurent et al., 2016; Philippon et al., 2011), to lineations that swing towards the NE (this study, Roche et al., 2016: Sifnos) and E-W during exhumation (c.f., Kotowski and Behr, 2019; Laurent et al., 2016)."*

*We have added more information to the proposed tectonic model in Section 6.5 (previously 6.4).*

Lines 438-440: Not only. Stretching lineations are also oriented NE-SW with top-to-the NE sense of shear on Sifnos.

*Agreed. This has been added (lines 528 – 531):*

*"During this phase of exhumation, CBU rocks remained within a cold forearc until they reached the mid-crust (~1.0 GPa), and exhibit a progressive change in kinematics, from N-S stretching lineations during subduction (e.g., Behr et al., 2018; Laurent et al., 2016; Philippon et al., 2011), to lineations that swing towards the NE (this study, Roche et al., 2016: Sifnos) and E-W during exhumation (c.f., Kotowski and Behr, 2019; Laurent et al., 2016)."*

Lines 440 – 444: Can you be a bit more specific on what is exactly the input of your study on the exhumation model of the CBU? Should we understand that there is no record of deformation acquired during back-arc setting in Syros? You presented different exhumation models (e.g.

subduction channel vs. extrusion wedge models) earlier in the manuscript and it would be great to come back on this here. Why do you prefer the subduction channel model? Perhaps illustrating your exhumation model would help.

*We now further discuss our exhumation model in section 6.5. In short, we think deformation that is related to back-arc extension does occur, but back-arc extension is primarily recorded locally, directly adjacent to the Vari detachment on southern Syros.*

Figure 2: I am really not convinced by picture 2a. The fold is not clearly visible and it seems that the foliation in the greenschist cross-cuts the fold axial plane.

*We've updated figure 2a to better illustrate the broad-scale fold and core of the fold. The foliation ($S_s$), is subsequently folded during $D_{t2}$. The figure shows the axial plane of the $F_{t2}$ fold, "cutting" the $D_s$ foliation. The $S_s$ foliation is a bit tricky to envision, but we consider it an early foliation that is continually retrogressed during $D_{t1}$ (blueschist) and $D_{t2}$ (primarily greenschist) upright folding. The $S_{t2}$ axial planar cleavage that is formed during $D_{t2}$ is primarily non-penetrative, but only locally produces a penetrative cleavage in the cores of the $F_{t2}$ folds (Figure 2a,b,c).*

[Figure]

*Figure 2*

---

## Author Response (AR3)

*We thank the editor for further comments that help strengthen the clarity of this manuscript for future readers. We address this comment below. Replies are italicized, text added to the manuscript is in red.*

"Dear Authors,
many thanks for this further revision.

After having gone through the Author's response letter and the revised typescript, I acknowledge an overall implementation of the manuscript. I am therefore supportive of the manuscript publication.

I have a comment dealing with nomeclature of the deformation structures adopted in this study: which the meaning of the subscript "s" and "t", when indicating the deformation phases? Which the difference between Ds and Dt1 and Dt2 (and related structures: S, F...)?...I think this should be clarified and implemented in the text and figures.

Sincerely,
federico rossetti"

*We have added further clarification in the main text and in figure captions to clarify the structural progression and subscripts used in this manuscript. We hope that this additional text helps clarify the nomenclature for readers.*

***Main text (lines 96 – 105):***

*"3. Field and Microstructural Observations*

*We studied four localities on Syros (Kalamisia, Delfini, Lotos, Megas Gialos; Fig. 1). Each locality exhibits multiple stages of mineral growth, and the same deformation and P-T progression. The abbreviations D, F, and S refer to deformation, folds, and foliations, respectively. Subscripts are listed in alphabetical order to differentiate older and younger stages of deformation (i.e., $D_s$, $D_t$). $D_s$ is the oldest observed deformation in outcrop that is recorded by tight isoclinal folds ($F_s$) that define the primary foliation ($S_s$). $D_t$ refers to younger deformation that is defined by upright folds ($F_t$). We assign subscripts in numerical order to indicate older ($D_{t1}$) and younger ($D_{t2}$) upright folding. Kalamisia records blueschist facies metamorphism, and Delfini, Lotos, and Megas Gialos record blueschist-greenschist facies metamorphism. GPS coordinates of collected samples and their associated mineralogy are provided in the supplementary material (Supplementary Table S1). 1 – 4 samples from each locality were examined petrographically."*

***Figure 2, caption:***

*"Figure 2. Outcrop, micrograph, and electron images showing stages of retrograde deformation present in southern Delfini. $D_s$ and $D_{t2}$ represent older and younger stages of deformation,*

*respectively. a) Upright folds ($F_{t2}$) formed during $D_{t2}$, that refold the older primary $S_s$ foliation. b): Core of $F_{t2}$ folds (below Fig. 2a, KCS34)."*

**Figure 3, caption:**

*"Outcrop photos of epidote boudins sampled for oxygen isotope thermometry. a) SY1613 (Lotos), b) SY1617 (Delfini), c) SY1618 (Delfini), d) SY1623 (Delfini). Boudins formed during $D_{t2}$, parallel to $F_{t2}$ fold hinge lines."*

**Figure 7, caption:**

*"$D_s$ is the oldest stage of deformation in outcrop. Subsequent $D_t$ deformation is separated into $D_{t1}$ and $D_{t2}$ to differentiate older and younger stages of upright folding, respectively, that form NE-SW ($D_{t1}$) and E-W ($D_{t2}$) lineations under blueschist and greenschist facies conditions."*